# Sample-efficient Integration of New Modalities into Large Language Models

## Abstract

Multimodal foundation models can process several modalities. However, since the space of possible modalities is large and evolving over time, training a model from scratch to encompass all modalities is unfeasible. Moreover, integrating a modality into a pre-existing foundation model currently requires a significant amount of paired data, which is often not available for low-resource modalities. In this paper, we introduce a method for *sample-efficient modality integration* (SEMI) into Large Language Models (LLMs). To this end, we devise a hypernetwork that can adapt a shared projector—placed between modality-specific encoders and an LLM decoder—to any modality. The hypernetwork, trained on high-resource modalities (i.e., text, speech, audio, video), is conditioned on a few samples from any arbitrary modality at inference time to generate a suitable adapter. To increase the diversity of training modalities, we artificially augment the number of encoders through isometric transformations. We find that SEMI achieves a significant boost in sample efficiency during few-shot integration of new modalities (i.e., satellite images, astronomical images, inertial measurements, and molecules) with encoders of arbitrary embedding dimensionality. For instance, to reach the same accuracy as 32-shot SEMI, training the projector from scratch needs $64\times$ more data. As a result, SEMI holds promise to extend the modality coverage of foundation models.

## 1 Introduction

Multimodal Foundation Models (MFMs) can perceive multiple modalities in input. Despite recent attempts to train "omni-modal" models (Lu et al., 2024b; Shukor et al., 2023; Xu et al., 2025), these typically cover only a pre-defined and limited set of modalities. As AI-based solutions are introduced into new fields and problem settings, the set of relevant modalities grows. As a consequence, it has become crucial to develop strategies to integrate new modalities incrementally into existing models (Han et al., 2023a; Yu et al., 2024) without naïvely re-training them from scratch, which is extremely resource-intensive (Jiang et al., 2024). A widely established practice consists of training a projector between each modality-specific encoder and a shared Large Language Model (LLM) decoder (Dubey et al., 2024) in a modular fashion (Pfeiffer et al., 2023), thus recycling the pre-trained components. While being more compute-efficient than re-training from scratch, this often requires a large amount of paired data containing samples of the new modality and text. This crucial limitation makes integration unfeasible for low-resource modalities and burdensome for high-resource ones.

To this end, we propose a novel paradigm called *sample-efficient modality integration* (SEMI), which tackles the fundamental question: how to integrate new modalities into foundation models given only a minimal set of samples? Specifically, we partition modalities into two groups: a few high-resource training modalities (i.e., image, audio, and video) and low-resource test modalities spanning distinct domains (i.e., satellite and galaxy images) and even entirely novel sample spaces (i.e., inertial measurement unit data and molecules). We hold out these test modalities to simulate new modalities that may emerge in real-world scenarios, which could range from very similar to highly distinct compared to the training modalities. We posit that learning the fundamental structure of modality integration from a subset of resource-rich modalities is sufficient to extrapolate to most other modalities (see Figure 1).

We introduce a three-stage paradigm for SEMI. In the first stage, we pre-train a non-linear projector between encoders of training modalities and an LLM decoder (both frozen) on the task of modality-

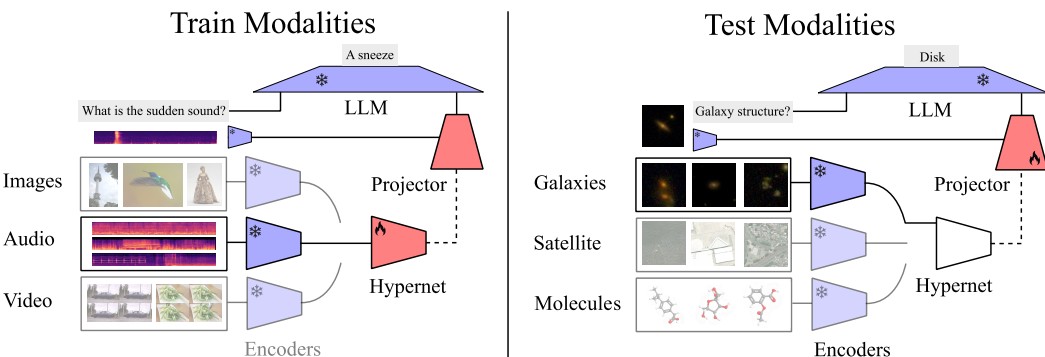

Figure 1: High-level framework of SEMI. **Left:** A hypernetwork is trained to generate an appropriate projector adaptation for high-resource training modalities. **Right:** A projector is generated by the hypernetwork for any unseen low-resource modality given only a few samples and is then fine-tuned on the same data. This enables the integration of new modalities with minimal training and paired data. Note that some modalities (audio, molecules) appear as images for visualisation purposes only.

to-text generation. In the second stage, we train a hypernetwork (Schmidhuber, 1992; Ha et al., 2017) on multimodal description data. The hypernetwork generates LoRAs (Hu et al., 2022), conditioned on samples from training modalities, to adapt the shared projector from the first stage. Importantly, this hypernetwork is task-agnostic, enabling a single trained model to generalise to any downstream task for a given modality.

In the third stage, we evaluate few-shot adaptation to the 4 unseen (i.e., held-out during training) low-resource modalities. At the start of the third stage, the hypernetwork generates an adapter for each unseen modality given a small set of samples. The adapter is then fine-tuned on this very same data. By measuring how the model performance varies with increasing sample sizes, we find a significant boost in sample efficiency by virtue of our method. Specifically, we show that SEMI creates effective projectors with as few as 32 samples while other baselines fail, and often remains the best approach even in comparably larger-scale data regimes.

In summary, we offer the following main contributions: **1)** enabling the integration of low-resource modalities that have small-scale paired modality–text data but large-scale modality-only data; **2)** providing a systematic comparison of different baseline approaches for the newly defined challenge of sample-efficient modality integration; **3)** curating a collection of benchmarks for this challenge, including the creation of a new dataset for astronomical imaging; **4)** proposing inexpensive strategies to augment the number of training 'modalities' through isometric transformations of encoder outputs and generalise to arbitrary-dimensionality encoders. As a consequence, our work opens new opportunities to apply AI-based solutions to modalities that are resource-poor due to privacy constraints or the difficulty of collecting large quantities of paired data.

## 2 RELATED WORK

**Integrating a New Modality into Language Models**  Modalities different from text can be integrated into LLMs through several approaches, usually requiring large amounts of paired modality–text data. The most common one involves learning an MLP projector to map modality-specific encoder outputs onto the LLM input space (Mokady et al., 2021; Koh et al., 2023; Gao et al., 2023; Liu et al., 2023a). Alternative methods include representing modality data as discrete tokens (Ge et al., 2023), incorporating trainable cross-attention layers within the LLM (Chen et al., 2021), employing more complex projectors like Q-Former (Li et al., 2023) or Perceiver (Jaegle et al., 2021; Guo et al., 2024), or combining cross-attention with a Perceiver resampler (Alayrac et al., 2022). In the present work, we empirically demonstrate that the common strategy of naïvely training an MLP projector from scratch for each new modality does not effectively integrate low-resource modalities into LLMs.

**Incremental Integration of Multiple Modalities**  MFMs often directly build upon the approaches listed above to integrate multiple modalities, thereby inheriting their data inefficiency. For instance, (Zhao et al., 2023b; Moon et al., 2024) train a separate projector for each modality, without any

parameter sharing. Others improve upon this paradigm, by incrementally aligning modalities during training: this is achieved through a shared encoder paired with modality-specific tokens, which are then routed to projector experts in OneLLM (Han et al., 2023a), or a combination of uni-modal and cross-modal adapters in PathWeave (Yu et al., 2024). We compare against baselines inspired by these incremental modality adaptation strategies, where we adapt shared projectors with parameter-efficient fine-tuning. Contrary to the setup in OneLLM, we assume modality-specific encoders to be given, instead of jointly trained. This modularity adds more flexibility and is compatible with integrating existing off-the-shelf encoders.

On the other hand, other strategies for integrating multiple modalities do not meet the desiderata for our setup, namely, sample efficiency and compatibility with generative text models. Incorporating new modalities by combining existing MFMs (Chen et al., 2024a) is unrealistic for our setup, as it assumes that MFMs for a low-resource modality exist in the first place. While Liu et al. (2024c) focused on "*enabling models to generalise to unseen modalities*", it did not demonstrate how to integrate new modalities into generative LLMs. Bind-style frameworks (Girdhar et al., 2023; Zhu et al., 2024a; Lyu et al., 2024) similarly do not meet our sample efficiency requirements, as they require thousands to millions of paired examples to align modalities into shared embedding spaces. Finally, other strategies—such as EE-MLLM (Ma et al., 2024), Macaw-LLM (Lyu et al., 2023), and ImageBind-LLM (Han et al., 2023b)—either insufficiently substantiate their sample efficiency or implicitly assume the availability of large-scale paired data for low-resource modalities during prior training stages. This prevents them from serving as baselines for our low-resource integration challenge.

## 3 SAMPLE-EFFICIENT MODALITY INTEGRATION

In this work, we pragmatically define any two 'modalities' to be distinct if they correspond to different distributions of encoder outputs. However, a finer distinction can also be made between a new encoder for the same data distribution, a change in probability function $p$ for the same sample space $\Omega$ (e.g., a domain shift), and a change in both $p$ and $\Omega$, for instance, RGB images $\{0, \ldots, 255\}^{H \times W \times 3}$ versus directed graphs $\{0, 1\}^{N \times N}$.

Our main goal is to devise an efficient and effective way of integrating low-resource data modalities into LLMs. We aim to develop a general framework that makes minimal assumptions about the distributions of data for unseen modalities or the encoder architectures they use. Although several approaches with these properties exist (see Section 2), we start from the most widespread and conceptually simple approach that only requires an MLP projector between the modality encoders and the LLM decoder.[1]

For SEMI, we propose a solution consisting of three stages: **1**) we train a shared MLP projector with coarse-grained data from resource-rich modalities (Section 3.1); **2**) we then train a hypernetwork (Schmidhuber, 1992; Ha et al., 2017) to generate projector adapters with instruction data from resource-rich modalities (Section 3.2). Intuitively, this aims to transfer the ability to adapt to a modality from high-resource modalities to low-resource ones; **3**) finally, we fine-tune the adapter generated for a new, resource-poor modality with only a few data points (Section 3.3). Additionally, we show how SEMI can generalise to new encoders of arbitrary dimensionality (Section 3.4). We present an overview of this process in Figure 1.

### 3.1 TRAINING THE SHARED PROJECTOR ON TRAIN MODALITIES

Assume we are given $M$ encoders $\{\text{enc}_m\}_{m=1}^M$ for high-resource modalities and a decoder LLM, whose parameters are all frozen. First, we train a projector $\text{proj}_\psi(\cdot) : \mathbb{R}^{h_e} \to \mathbb{R}^{h_d}$ on paired raw data from these modalities (i.e., image–text, video–text, and audio–text), where $h_e$ and $h_d$ are the encoder output and decoder input dimensions, respectively. This establishes a universal mapping between observed encoders and the LLM decoder.

---

[1] While our framework is compatible with other approaches such as cross-attention or Q-Former, we leave these extensions to future work.

## 3.2 TRAINING THE HYPERNETWORK ON TRAIN MODALITIES

Afterwards, we train a hypernetwork to generate modality-specific adapters to be composed with the shared projector given only a small data sample, effectively simulating the desired test-time few-shot adaptation to low-resource modalities. The first step in hypernetwork training is sampling a training modality encoder $\text{enc}_m$, an instruction $\mathbf{i}_m \in \mathcal{I}_m$ from the instruction pool belonging to that modality, and a sample of examples $\{\mathbf{x}_m, \mathbf{y}_m\}_1^S \in \mathcal{D}_m$: each example consists in a modality-specific input $\mathbf{x}_m$ and text $\mathbf{y}_m$ to better ground the modality. The hypernetwork receives the corresponding interleaved encodings $(\text{enc}_{\text{text}}(\mathbf{i}_m) \oplus [\text{enc}_m(\mathbf{x}_m) \oplus \text{enc}_{\text{text}}(\mathbf{y}_m)]_1^S)$ and generates LoRA adapters $\delta$.

After sampling a separate batch $\{\mathbf{x}_m, \mathbf{y}_m\}_1^B$, the LoRA adapter is plugged into the parameters of the shared projector (now frozen) to project the encoding of the modality-specific input into the LLM token space. This yields $\text{proj}_{\psi+\delta}(\mathbf{x}_m)$, which, combined with the instruction $\mathbf{i}$, is fed into the LLM decoder. The final loss is the cross-entropy of the LLM decoder prediction with respect to the target texts $\{\mathbf{y}_m\}_1^B$, whose gradient is back-propagated to the hypernetwork parameters for optimisation. Note that the sample size for the hypernetwork $S$ and the batch size of the LLM decoder $B$ may be different, and thus can be chosen arbitrarily. Nevertheless, $S$ is preferably chosen to be small due to the limited amount of text-paired data one might collect for test modalities. Additionally, we demonstrate that larger context lengths do not necessarily improve performance (see Table 3).

We employ several techniques to enhance performance and simplify the optimisation process. These techniques include factorising the hyper-network's generated parameters, using isometric transformations for encoder augmentation, and grounding modality inputs with text when feeding them to the hypernetwork, as expounded in the following paragraphs. The pseudocode for the shared projector training, as well as hypernetwork training and adaptation, is provided in Appendix J.

**Hypernetwork Optimisation** Hypernetworks often present optimisation challenges and high computational complexity. For instance, predicting all parameters of an $N \times M$ weight matrix from a $K$-dimensional embedding requires $N \times M \times K$ parameters in the hypernetwork's generating linear layer. In our setup, we reduce complexity by generating lower-rank adapter parameters $\delta$ representing the difference between the pre-trained projector $\text{proj}_{\psi}$ and the target modality projector $\text{proj}_{\psi+\delta}$ instead of the full projector weights. This change requires $(N + M) \times R \times K$ parameters, and depending on the LoRA rank $R \ll M$, it can remediate the parametric complexity significantly. This also alleviates hypernetwork initialisation challenges (Chang et al., 2020; Chauhan et al., 2024).

**Encoder Augmentation** Ideally, our hypernetwork should infer the statistical properties and characteristics of a new modality $m'$ through the lens of a modality encoder $\text{enc}_{m'}$ given only a small number of samples. Therefore, training the hypernetwork on numerous modality encoders so that it generalises better, rather than over-fitting to a few encoders, is preferred. However, if we simply scaled the number of modalities by sourcing more readily available, pre-trained encoders, we would soon encounter a barrier due to their scarcity.

Hence, we use random orthogonal matrices as encoder augmentation, sampling them from an $O(d_h)$ Haar distribution (Mezzadri, 2007; Virtanen et al., 2020), where $d_h$ denotes the hypernetwork dimension, and we transform modality-specific encodings through these matrices before feeding them as input to the hypernetwork or the adapted projector. Orthogonal matrices possess desirable properties like invertibility and isometry, preserving the Euclidean distance and the inner product of vectors. For instance, orthogonal matrices encompass rotation, reflection, and permutation, among other types of linear transformations. By using these random transformations, we can augment encoders by altering the general distribution of the data while preserving certain characteristics along with the local spatial relationships between instances.

**Grounding with Text** Increasingly more bimodal MFMs incorporate text conditioning into modality projections (Dai et al., 2023; Kar et al., 2024; Ghazanfari et al., 2024). As our hypernetwork must generalise across diverse unseen modalities and encoder distributions, the same embedding may hold different meanings in different encoder spaces. This is because each modality encoder $\text{enc}_m$ learns a unique mapping from the input data to its embedding space, influenced by its architecture and training data, among other factors. If the hypernetwork treats these embeddings as universally comparable without taking this variation into account, it can generate inaccurate adapters for new modalities. To overcome this issue, we provide the hypernetwork with both instruction embeddings $\text{enc}_{\text{text}}(\mathbf{i}_m)$ and text embeddings $\text{enc}_{\text{text}}(\mathbf{y}_m)$ alongside modality embeddings $\text{enc}_m(\mathbf{x}_m)$. Note that

Table 1: Training modalities, datasets, and encoders. We generally use captioning datasets during the pre-training of the projector and instruction datasets during the hypernetwork training.

| Modality | Stage 1 (Projector pre-training) | | Stage 2 (Hypernetwork training) | |
| --- | --- | --- | --- | --- |
| | Dataset | Encoder | Dataset | Encoder |
| Text & Image | COCO[(Lin et al., 2014)] | CLIP[(Radford et al., 2021)] | ShareGPT4V[(Chen et al., 2023)] | SigLIP 2[(Tschannen et al., 2025)] |
| Text & Audio | AudioCaps[(Kim et al., 2019)] | CLAP[(Elizalde et al., 2023)] | Clotho-Detail[(Drossos et al., 2019)] | Cacophony[(Zhu et al., 2024b)] |
| Text & Video | OpenVid[(Nan et al., 2024)] | VideoCLIP-XL[(Wang et al., 2024a)] | ShareGPT4Video[(Chen et al., 2024b)] | ViCLIP[(Wang et al., 2024c)] |

instruction and text embeddings are extracted from the same frozen text encoder $enc_{text}$ throughout training and inference, effectively grounding modality embeddings on a fixed representation space.

### 3.3 Adaptation of the Generated Adapters for Test Modalities

Finally, during adaptation to a new test modality, the hypernetwork (as well as the LLM decoder and modality encoders) remains frozen. We partition the training data into batches with a maximum sequence length determined by the context length of the hypernetwork, and generate adapters for each batch. For each batch, similar to Section 3.2, an instruction and interleaved modality–text data are encoded and then fed to the hypernetwork, which generates adapters for the new modality. Adapters generated across batches are averaged ($\bar{\delta}$), then merged with the pre-trained projector parameters $\psi$ to create an updated projector $proj_{\psi+\bar{\delta}}$. We provide additional details on adapter generation techniques in Appendix F. Finally, the merged projector is fine-tuned on the low-resource modality's few-shot samples, based on the cross-entropy loss of true and predicted output text. We illustrate the effects of this third stage on the representation similarity of text and each unseen modality in Appendix H.

### 3.4 Integrating Arbitrary Dimensionality Encoders

Although we enforce a fixed hypernetwork input dimension, we demonstrate that SEMI can effectively generalise to encoders with varying output dimensions, a capability previously unexplored in this context to the best of our knowledge. To handle smaller encoder dimensions, we prune the pre-trained projector by removing the final dimensions from its weights and biases to match the encoder output dimension. To handle larger encoder dimensions, we utilise an efficient unsupervised feature selection method, Infinite Feature Selection (Inf-FS) (Roffo et al., 2015), to reduce the encoder output dimensions. We hypothesise that Inf-FS is superior to alternative techniques such as PCA in settings (such as ours) where the number of training samples is much smaller than the number of dimensions. With $N$ samples, $d_e$-dimensional encoder, and $d_h$-dimensional hypernetwork, where $d_h < d_e$, $d_e \gg N$, and $d_h \gg N$, PCA's resulting embedding rank is limited to $N$. In contrast, as Inf-FS selects $d_h$ features, it can construct $d_h$-rank embeddings upper bounded by the rank of the original embeddings. We verify that Inf-FS is more stable than PCA with an ablation in Appendix E.

## 4 Experimental Setup

**Model Architecture** The pre-trained projector is a 2-layer MLP connecting modality-specific encoders to the LLM's input space. Our hypernetwork adopts an architecture where special tokens (one per generated layer) are concatenated with instruction–text–modality samples, whose encodings are combined with sinusoidal positional embeddings. An attention layer then contextualises the special token embeddings with respect to the samples. Finally, linear layers applied to each special token generate the corresponding adapter layers. In our setup, we generate adapters only for the first projector layer, leaving the second layer unchanged (see Appendix C for more details). We adopt Llama 3.1 8B Instruct (Dubey et al., 2024) and Llama 3.2 1B Instruct (Meta, 2024) as LLMs for our experiments. We report 8B LLM results in the main paper and 1B LM results in Appendix K.2. We conducted ablations and other exploratory experiments with the 1B LLM unless otherwise specified. We use GTE-ModernBERT-Base (Zhang et al., 2024; Warner et al., 2024) as our text encoder. Details on compute resources, runtime, and hyperparameters are provided in Appendix M.

**Modalities: Datasets and Encoders** We trained our hypernetwork on image, audio, and video data, then evaluated its few-shot adaptation capabilities across a spectrum of shifts from the training encoders: a new encoder for a seen modality (audio), two unseen domains for images (satellite and

Table 2: Evaluation modalities with their datasets and encoders. We distinguish among modalities based on whether they are seen, constitute an unseen distribution of a seen input space (i.e., a domain shift), or are completely unseen during training (see Section 3).

| Modality | Seen | Unseen Domain | | Unseen Modality | |
|---|---|---|---|---|---|
| | Audio | Satellite Images | Astronomical Images | IMU | Molecule |
| **Dataset** | SoundBible | SydneyCaptions & RSVQA | CAPDELS (ours) | SensorCaps & OpenSQA | ChEBI-20 |
| **Task** | Captioning | Captioning & VQA | Captioning | Captioning & Inst. Following | Captioning |
| **Encoder** | BLAT | RemoteCLIP | Zoobot ConvNeXt | LIMU-BERT | MolCA |
| **Variants** | - | B-32 / L-14 / RN-50 | Nano / Tiny / Base | - | - |
| **Emb. Dim** | 768 | 512 / 768 / 1024 | 640 / 768 / 1024 | 720 | 768 |

galaxies), and two entirely unseen modalities (IMU data and molecules). This allowed us to assess how the performance of SEMI changes on modalities from most to least similar to the training data in a systematic way. The chosen test modalities thus span a range of adaptation difficulties and diverse applied use cases of AI (in geolocation, astronomy, navigation, and biology/medicine).

Table 1 lists the modalities, datasets, and encoders employed during training. We followed the LLaVA framework (Liu et al., 2023a) and used coarse captioning datasets for projector pre-training (except for video data) and fine-grained description datasets for hypernetwork training. To mitigate potential overfitting, we used different encoders for the same modality during the two stages. This mimics our intended new modality adaptation scenario, in which the encoders for new modalities are unseen.

As for test modalities, their datasets, encoders, and additional information are detailed in Table 2. In addition to sourcing existing datasets, we also created CAPDELS, a pioneering novel astronomical imaging captioning dataset (see Appendix A for details), built on the Galaxy Zoo CANDELS multi-label galaxy morphological classification dataset (Simmons et al., 2016). For both satellite and astronomical imaging, we used a family of encoders varying in size and embedding dimension—RemoteCLIP (Liu et al., 2024a) and Zoobot ConvNeXt (Walmsley et al., 2023), respectively—but trained on the same dataset. This allowed us to make justifiable claims about the ability of SEMI to generalise to different encoder sizes. Note that the input spaces (images) for these two modalities are observed during training, while their distribution is shifted with respect to the domains of training images in COCO. In addition, we explore two entirely new input spaces: three-axis accelerometer and gyroscope numerical readings for IMU data and labelled graphs for molecules. For molecule data, we utilised the Q-Former of MolCA (Liu et al., 2023b). For IMU data, after extracting IMU embeddings via LIMU-BERT (Xu et al., 2021), we performed dimensionality reduction by averaging groups of consecutive tokens, effectively reducing the temporal resolution while preserving the feature space. The resulting embeddings are flattened into a single vector. Finally, we evaluated SEMI also on few-shot adaptation to an unseen encoder for one of the high-resource training modalities, namely audio. In this case, both the data domain and input space are similar to one of the training datasets (AudioCaps). This helps demonstrate the broad scope of our methods, which may benefit the integration of new encoders for seen, high-resource modalities, too. Additional dataset details, including dataset sizes and pre-processing steps, are provided in Appendix L.

**Baselines** To evaluate the impact of cross-modality transfer with our hypernetwork, we compared SEMI against three baselines that are representative of current state-of-the-art approaches. Our simplest baseline (*Projector*) consists in training a (randomly initialised) projector from scratch on the few-shot examples of each low-resource modality. A second baseline (*LoRA*) trains a LoRA adapter on few-shot examples and merges it with the pre-trained shared projector. Finally, *FT Projector* fully fine-tunes the pre-trained shared projector on each test modality, representing our strongest baseline. *LoRA* is reminiscent of PathWeave (Yu et al., 2024) and *FT Projector* of the OneLLM (Han et al., 2023a) framework; however, to the best of our knowledge, these baselines constitute the first attempt to streamline these setups and make them comparable in a controlled setting. As with our setup, we applied weight pruning or Inf-FS dimensionality reduction to *LoRA* and *FT Projector* when adapting them to smaller and larger encoder dimensionalities, respectively. Instead, the *Projector* baseline was directly created with the target dimensions as it does not rely on the pre-trained projector.

**Evaluation** We evaluate SEMI and the baselines using greedy decoding on the test set and calculating n-gram based metrics (BLEU-4 (BLEU) (Papineni et al., 2002), METEOR (Banerjee & Lavie, 2005), ROUGE-1, and ROUGE-2 (Lin, 2004)), a longest common subsequence based metric (ROUGE-L (Lin, 2004)), and CIDEr (Vedantam et al., 2015). Since CIDEr is designed for image description

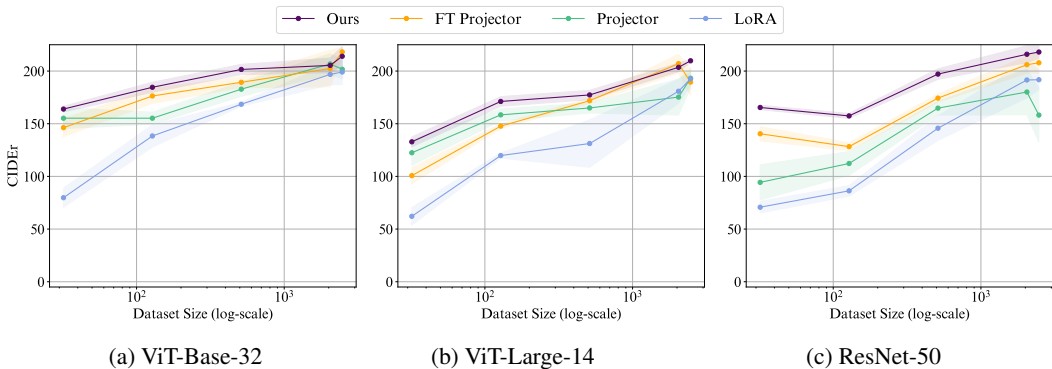

(a) ViT-Base-32      (b) ViT-Large-14      (c) ResNet-50

Figure 2: SydneyCaptions (Qu et al., 2016) satellite captioning results with three different encoders. The shaded areas around the lines indicate the standard error obtained from multiple seeds.

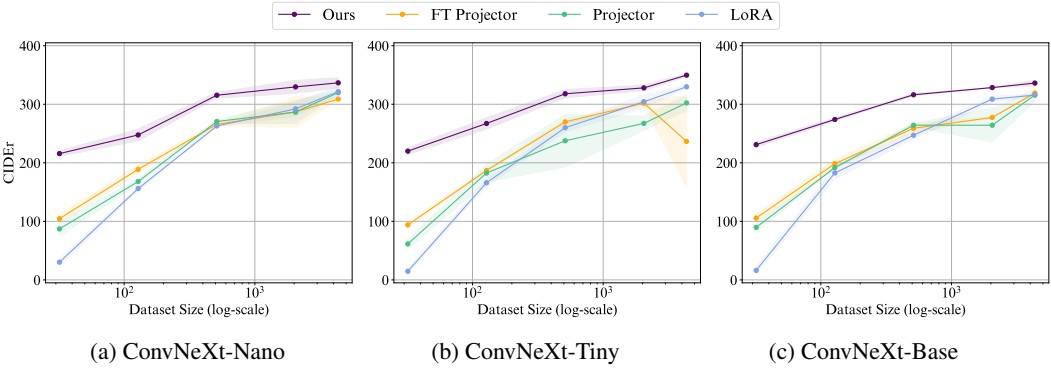

(a) ConvNeXt-Nano      (b) ConvNeXt-Tiny      (c) ConvNeXt-Base

Figure 3: CAPDELS astronomical image captioning results with three different encoders. The shaded areas around the lines indicate the standard error obtained from multiple seeds.

tasks, we excluded it from our IMU and molecule evaluation. We perform model selection through early stopping according to the model's CIDEr (or BLEU when unavailable) on the validation sets.

Given our focus on sample-efficient modality integration, we evaluated SEMI against the baselines using a range of dataset sizes for few-shot adaptation. To ensure a fair comparison, we randomly selected subsets of varying sizes from each unseen modality dataset and tested all methods on these identical splits. Specifically, the subsets range from 32 samples up to the full dataset size, increasing by a factor of four (e.g., 32, 128, 512, 2048, and 2485 for SydneyCaptions) (see Table 30 dataset sizes). We trained and evaluated each method with the same three random seeds to ensure identical training batches (with the exception of ChEBI-20, where a single seed was used due to its large size).

## 5 RESULTS

### 5.1 MAIN RESULTS

In this section, we report exact match for RSVQA, CIDEr scores for SydneyCaps and CAPDELS, BLEU scores for OpenSQA, SensorCaps, ChEBI-20, and SoundBible datasets; additional metrics and qualitative examples are available in Appendices K and I, respectively. To measure sample efficiency, we study how these metrics vary as a function of the sample size for each new modality.

**Satellite Images** The CIDEr scores for the SydneyCaptions are shown in Figure 2. Overall, our approach outperforms (or at worst matches) all baselines across all sample sizes and encoder dimensions. The methods generally rank in descending order of performance as follows: SEMI, *FT Projector*, *Projector*, and *LoRA*. However, note that *LoRA* partially bridges the gap in the higher data regimes. In fact, as the sample size grows, the impact of the inductive bias provided by each method diminishes.

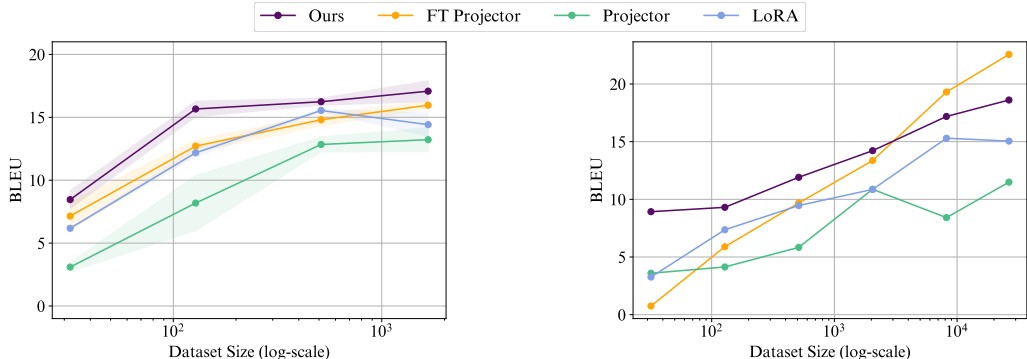

Figure 5: **Left**: SensorCaps (Imran et al., 2025) activity description dataset results. **Right**: ChEBI-20 (Edwards et al., 2021) molecule description dataset results.

When comparing encoders of varying sizes and dimensionality, our method consistently performed well, effectively generalising to encoders with a smaller or larger embedding dimension than what was observed during training (see Table 1 for the embedding dimensions of encoders). Importantly, our method exhibits the largest gains over baselines when integrating the encoder with the largest dimensionality, i.e. the ResNet-50 variant (Figure 2b). These results highlight the positive scaling behaviour of SEMI. As an additional finding, we observed that training projectors from scratch (*Projector* baseline) with smaller-dimensionality encoders (ViT-Base-32) improved performance in low-data regimes over larger encoders (ResNet-50) by preventing overfitting.

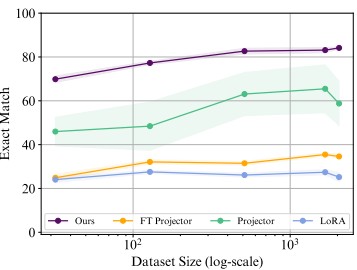

Figure 4: RSVQA satellite results with ViT-Large-14 encoder.

In RSVQA dataset, SEMI substantially outperforms all baselines across dataset sizes (see Figure 4). Notably, SEMI never observed visual reasoning tasks during training, yet demonstrates strong cross-task generalisation where the FT Projector baseline fails to adapt. For instance, at 32 samples, SEMI achieves 69.9% exact match compared to FT Projector's 24.9% – a performance gap that persists even at full dataset scale (83.1% vs 35.5%).

**Astronomical Images**  Although astronomical imaging results are similar to satellite imaging results, we note certain differences (Figure 3). Particularly, the gap between SEMI and all baseline is noticeably larger, especially in low- and mid-size encoders, reaching a difference of 200 CIDEr for ConvNeXt-Tiny and ConvNeXt-Base in 32-shot settings. On the other hand, the baselines perform comparably, and they all exhibit remarkable variance, showcasing their brittleness compared to SEMI.

**IMU Data**  Focusing now on entirely novel modalities, SEMI outperforms all baselines across all sample sizes for SensorCaps (see Figure 5 left). After our method, *FT Projector* and *LoRA* remain the second-best-performing methods, followed by the significantly weaker *Projector*. This highlights the positive contribution of cross-modal transfer even when the distance between train and test modalities increases. For instance, the best baseline, *FT Projector*, requires 16 times more examples (2048) to achieve comparable performance to 128-shot SEMI.

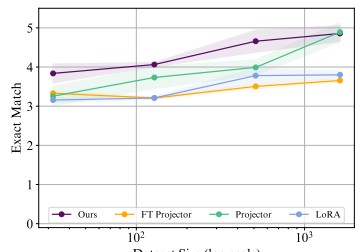

In OpenSQA, SEMI again outperforms or (at worst matches) all baselines (see Figure 6). This is particularly notable as SEMI was not trained on diverse instruction-following tasks, suggesting these results may represent a lower bound on SEMI's capabilities for this family of tasks.

Figure 6: OpenSQA IMU instruction following results.

**Molecules**  Overall, we observe that our method outperforms all baselines (see Figure 5 right) for molecules; however, differently from previous modalities, *FT Projector* eventually catches up and

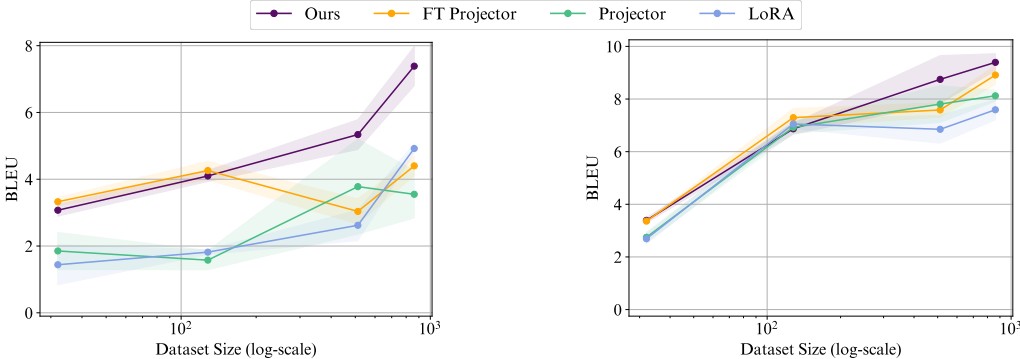

Figure 7: SoundBible (Mei et al., 2024) results with the BLAT (Xu et al., 2023) encoder. **Left:** Llama 3.2 1B Instruct. **Right:** Llama 3.1 8B Instruct.

surpasses SEMI at high-resource settings ($10^4$ examples). On the other hand, SEMI demonstrates particularly strong performance in low-resource settings. The best-performing baseline, *FT Projector*, requires 16 times more data (512 samples) to reach comparable results to 32-shot SEMI.

**Audio**   To present a more comprehensive view of our approach, we also evaluated our method on a new encoder for a seen modality, audio captioning. The results are available in Figure 7, where we additionally compare two LLM sizes (1B and 8B). For the smaller LM (left) and the larger LLM (right), we observe that our approach and *FT Projector* are comparable in extremely low data regimes ($< 10^2$); however, the performance discrepancy between SEMI and the baselines widens as sample size increases (more significantly so in the smaller LLM). This surprising finding suggests that hypernetwork-based SEMI may bring benefits to the integration of seen modalities, too.

## 5.2 Ablations

Finally, to justify the architecture of our hypernetwork, we conduct a series of ablations to demonstrate the impact of each of our design choices. Table 3 in Appendix D shows that the combination of text grounding and isometric transformations results in the most accurate modality integration overall, especially for the CAPDELS dataset. This stems from grounding all other modalities on text as an 'anchor' modality and from avoiding overfitting by artificially multiplying the number of encoders, respectively. On the other hand, a 2-layer transformer backbone instead of an attention backbone in the hypernetwork is detrimental to performance. We speculate that this occurs since a larger hypernetwork incurs overfitting to the training modalities by virtue of being more expressive. Moreover, increasing the context length from 128 to 192 is not beneficial, either, which contradicts our original expectation that larger samples should better approximate the underlying distribution of modality encodings.

## 6 Conclusions

We introduce a novel approach for sample-efficient integration of new modalities (SEMI) into large language models (LLMs). Given a projector, which maps between modality-specific encoders and a decoder LLM, we design a hypernetwork that can adapt it towards any modality. The hypernetwork is trained using data from a limited set of high-resource modalities (e.g., image, audio, and video) and learns to generalise to unseen modalities like satellite images, astronomical images, IMU data, and molecules. We curate a benchmark to measure sample efficiency in this diverse array of modalities, sourcing existing datasets and introducing a new one for galaxy captioning.

We employ isometric transformations to diversify the encoder distributions encountered during training, thereby preventing overfitting. On top of this, grounding modality-specific embeddings on text further enables sample-efficient integration. Overall, SEMI achieves an accuracy comparable with the strongest baseline, namely fine-tuning the shared projector on the new modality, while usually requiring $16\times$ less labelled data. Finally, we demonstrate the ability of SEMI to generalise to new encoders with arbitrary dimensionality and to tasks beyond those observed during training. By reducing the reliance on large-scale labelled data, our framework facilitates the integration of

a diverse set of new modalities into LLMs, expanding the potential applications of multimodal AI models to new areas of geo-location, astronomy, navigation, and chemistry. Our approach takes a step forward in the development of truly omni-modal foundation models by extending their coverage to low-resource modalities.

# 7 LIMITATIONS

Our work assumes that enough modality-specific raw data exists to train an encoder, as it relies on off-the-shelf encoders. In practice, we surveyed a wide array of additional low-resource modalities and found encoders to be widely available for all of them, as illustrated in Appendix B. This makes our assumption entirely realistic. Secondly, even though our projector architecture (MLP) is adopted by the vast majority of available multimodal LLMs (Liu et al., 2024b; Wang et al., 2024b; Lu et al., 2024a; Wu et al., 2025, and more), we did not experiment with alternative projector architectures. Finally, we focused only on integrating new modalities in input rather than generating new modalities in output: we leave this endeavour to future work. While testing SEMI on an even broader range of modalities is left for future work, we foresee no theoretical or practical barriers that would prevent SEMI from generalising beyond the modalities we studied.

ETHICS STATEMENT

The ethos of this work is democratising modality integration to LLMs in an efficient, private, and environmentally responsible way. However, similar to text-only adversarial prompts jailbreaking LLMs (Zeng et al., 2024), modality inputs to multimodal LLMs could be used for jailbreaking (Gu et al., 2024). Nonetheless, we do not see that our work opens more vulnerabilities than existing training paradigms. Moreover, we use every dataset, encoder, and LLM (see Tables 29 and 31) with their respective licences.

REPRODUCIBILITY STATEMENT

To promote reproducibility, we have provided detailed pseudocodes in Appendix J, dataset information and preprocessing details in Appendix L, and comprehensive training details—including encoder specifications, computing resources, run times, and hyperparameters—in Appendix M. The main experimental setup is described in Section 4. An anonymous code repository supporting our implementation is also available as supplementary material. We believe these resources collectively enable full reproduction of our results.

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

# APPENDIX

**LLM Use**    The authors acknowledge the use of Gemma 3 27B (Kamath et al., 2025) LLM to suggest improvements on fluency and the mitigation of grammatical errors.

## A    CAPDELS DATASET

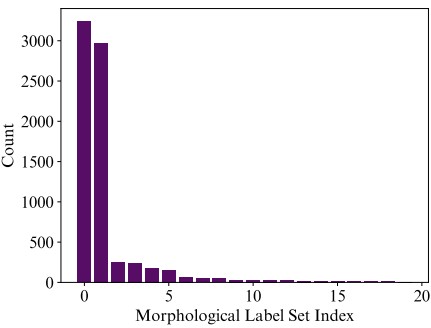 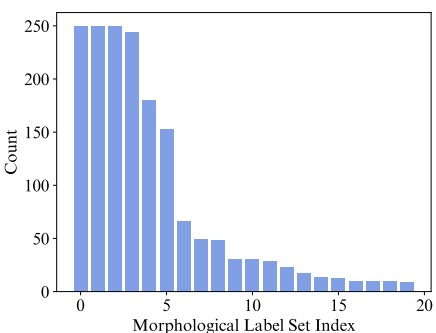

Figure 8: The frequency of most frequent label sets before (**left**) and after (**right**) balancing.

We introduce CAPDELS, a novel astronomical imaging captioning dataset constructed from the galaxy morphological multi-label classification dataset CANDELS (Simmons et al., 2016). While CANDELS contains approximately 50000 examples, only 8000 are 'clean' according to author-crafted thresholds. This limited number of clean labels makes it a low-resource dataset, making it a suitable use case for our work. Furthermore, the distribution of morphological label sets is highly imbalanced; only 170 distinct sets exist within these 8000 examples, with approximately 6000 examples belonging to just two dominant sets (see Figure 8). To address this imbalance, we pruned the most frequent two label sets to match the size of the third-largest set, leaving only 2045 galaxy images in total.

The CANDELS dataset employs a classification tree for categorising galaxy morphology. We leveraged this structure by using the probabilities assigned by annotators for each galaxy to determine its corresponding label sets. We then used 4-bit quantised Qwen-2.5-32B Instruct (Yang et al., 2024) to generate captions, providing only the morphological label sets as input – the LLM does not have access to the images themselves. Captions were generated using a system prompt inspired by Liu et al. (2023a), along with a JSON dictionary containing each sample's morphological information, producing three captions per image. We used sampling hyperparameters of temperature = 0.4, top-k = 30, and top-p = 0.8 to encourage consistent outputs.

CAPDELS will be released under the same CC BY-NC-SA 4.0 licence as the CANDELS dataset.

---

**System prompt to Qwen-2.5-32B Instruct for caption generation**

You are an AI assistant tasked with generating a caption for a galaxy image based on its morphological structure. The information about the galaxy will be provided to you as a JSON dictionary, which includes details about its morphological properties.

Your job is to create a caption using all the given morphological details from the JSON dictionary. Ensure your caption is:

- Simple and easy to understand
- Concise but specific
- Complete (do not omit any details except probabilities)
- Using the exact astronomical terminology found in the JSON dictionary

---

---

**An input example to Qwen-2.5-32B Instruct**

```
 "Is the galaxy simply smooth and rounded, with no sign of a disk?":
  "Answer: features or disk. Probabilities: smooth: 15%, features: 82%,
   artifact: 0%",
 "Does the galaxy have a mostly clumpy appearance?":
  "Answer: no. Probabilities: yes: 23%, no: 77%",
 "Could this be a disk viewed edge-on?":
  "Answer: no. Probabilities: yes: 7%, no: 93%",
 "Is there a sign of a bar feature through the centre of the galaxy?":
  "Answer: no. Probabilities: yes: 10%, no: 90%",
 "Is there any sign of spiral arm pattern?":
  "Answer: yes. Probabilities: yes: 89%, no: 11%",
 "How tightly wound do the spiral arms appear?":
  "Answer: loose. Probabilities: tight: 35%, medium: 18%, loose: 47%",
 "How many spiral arms are there?":
  "Answer: 1. Probabilities: 1: 82%, 2: 6%, 3: 2%, 4: 0%, 5+: 0%,
   can't tell: 9%",
 "How prominent is the central bulge, compared with the rest of the galaxy?":
  "Answer: obvious. Probabilities: no bulge: 16%, obvious: 61%,
   dominant: 23%",
 "Is the galaxy currently merging or is there any sign of tidal debris?":
  "Answer: tidal debris. Probabilities: merging: 1%, debris: 50%,
  both: 2%, neither: 47%"
}
```

---

**Qwen-2.5-32B Instruct generated caption**

The galaxy has a distinct disk structure with no signs of being edge-on, featuring an obvious central bulge and one loose spiral arm. There is no bar feature through the centre, but there are signs of tidal debris present. The appearance is not clumpy.

---

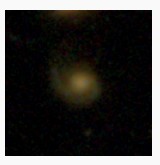 The image for reference (not used by LLM).

## B  AVAILABILITY OF LOW-RESOURCE MODALITY ENCODERS

We analysed six additional low-resource modalities and found that all of them have available encoders. Specifically, we found the following encoders for each field: for seismology EQTransformer (Mousavi et al., 2020), for low-resource speech MMS (Pratap et al., 2023), for meteorology LightWeather (Fu et al., 2024), for bioacoustics acoupi (Vuilliomenet et al., 2025), for ecology the encoder from Pantazis et al. (2021), and for tactile T3 (Zhao et al., 2024).

## C  PLUGGING ADAPTERS TO ALL LAYERS

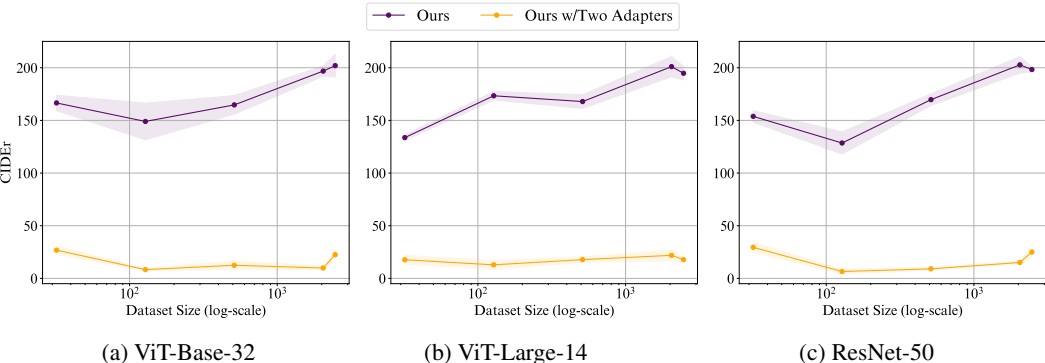

(a) ViT-Base-32      (b) ViT-Large-14      (c) ResNet-50

Figure 9: Comparison of different adapter integration techniques for the SydneyCaptions dataset.

As Figure 9 shows, we encountered a significant performance decrease when training the hypernetwork to add adapters to both projector layers during training. The qualitative results indicate that the projector built using the two-layer setup overfits to the hypernetwork training data (e.g., a drastic increase in usage of training data vocabulary). We hypothesise that bypassing the second layer and adapter during training mitigates this overfitting issue. Nonetheless, we speculate that this effect is due to the relative lack of variation in the hypernetwork and projector training data, and can be overcome by diversifying the training data and increasing its scale.

## D  ABLATIONS FOR THE HYPERNETWORK ARCHITECTURE

An ablation comparing different design choices for the hypernetwork architecture is shown in Table 3. Specifically, we consider removing text grounding (*w/o Text*), removing isometric transformations (*w/o IsoTransf*), or both (*w/o Text & IsoTransf*). We also evaluated a 2-layer Transformer hypernetwork instead of a single attention layer (*Larger Hypernet*) and expanding the hypernetwork context (*w/ Larger Ctx Len*).

Table 3: **Ablations.** The best result for each dataset size is bolded. Multiple methods are bolded if they fall within each other's standard error range. ViT-L-14 and Tiny encoder variants are used for SydneyCaptions and CAPDELS, respectively.

| | **CAPDELS** | | | | | | **SydneyCaptions** | | | | | |
|---|---|---|---|---|---|---|---|---|---|---|---|---|
| Dataset Size | 32 | 128 | 512 | 2048 | 4344 | Avg. | 32 | 128 | 512 | 2048 | 2485 | Avg. |
| Ours | 136.8 | **178.4** | 223.5 | 273.3 | 255.0 | 213.4 | 133.7 | **173.4** | **167.9** | **201.0** | **194.8** | 174.1 |
| w/o Text | **165.4** | **183.2** | **278.0** | **295.6** | **303.6** | 245.2 | **145.2** | 143.8 | 139.0 | 147.0 | 175.9 | 150.2 |
| w/o IsoTransf | 24.7 | 108.2 | 201.4 | 243.6 | 280.3 | 171.6 | **146.5** | 158.4 | **174.4** | **201.2** | 181.6 | 172.4 |
| w/o Text & IsoTransf | 16.2 | 123.2 | 190.1 | 221.8 | 276.1 | 165.5 | **141.5** | 158.7 | 157.0 | 167.8 | 155.2 | 156.0 |
| w/ Larger Hypernet | 111.5 | 150.7 | 197.3 | 210.3 | 266.0 | 187.2 | 134.9 | 153.1 | 124.8 | 169.2 | 158.5 | 148.1 |
| w/ Larger Ctx Len | 143.2 | **183.7** | 236.0 | 237.5 | 245.2 | 209.1 | 117.5 | 135.7 | 149.7 | 150.8 | 170.9 | 145.0 |

## E    COMPARING DIMENSIONALITY REDUCTION METHODS

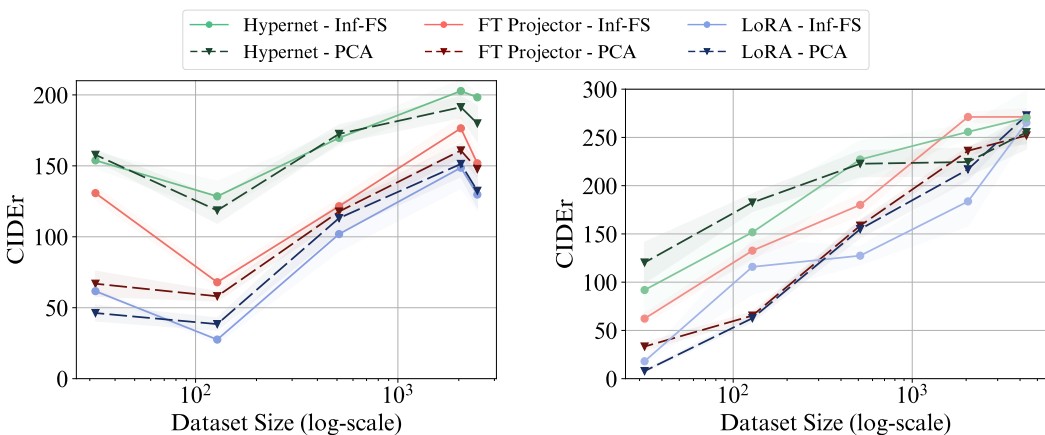

Figure 10: Results with PCA and Inf-FS dimensionality reduction techniques for SydneyCaptions (**left**) and CAPDELS (**right**) datasets.

When comparing dimensionality reduction techniques (see Figure 10), we found that while PCA performed similarly to, and sometimes slightly better than, Inf-FS in hypernetwork and LoRA experiments, its performance significantly degraded, especially on the *FT Projector* baseline. Because PCA offered no consistent advantage over Inf-FS across datasets when used with the hypernetwork, and negatively impacted the performance of *FT Projector*, we report results only for Inf-FS in the main paper.

## F    COMPARING DIFFERENT ADAPTER GENERATION TECHNIQUES

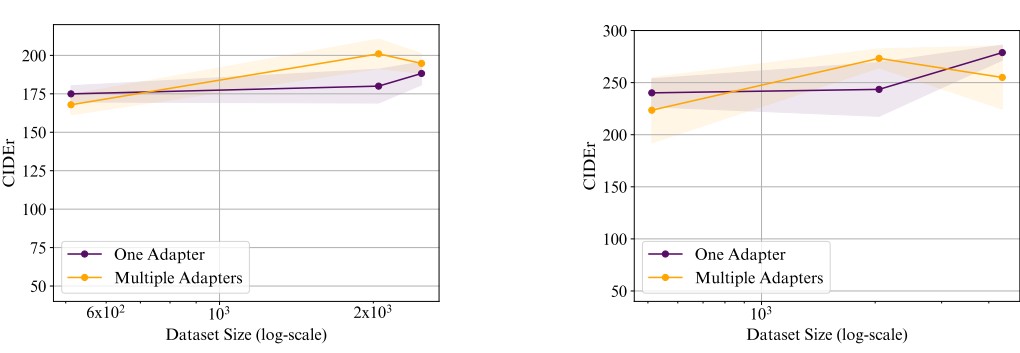

Figure 11: Comparison of single adapter versus averaging multiple adapters for SydneyCaptions - ViT-L-14 (**left**) and CAPDELS - ConvNeXt-Nano (**right**) setups.

In Figure 11, we compare the single adapter approach with the multiple adapter averaging approach. We observe that generating multiple adapters achieves comparable performance to the single adapter method, while incurring only negligible computational overhead (at most 16 seconds for 206 adapters on the full ChEBI-20 dataset split).

## G  SEMI WITHOUT FINE-TUNING (ZERO-SHOT ADAPTATION)

Table 4: **Zero-shot vs. Few-shot Adaptation.** The best result for each dataset size is bolded. CAPDELS Base and SydneyCaptions RN-50 encoder variants are used for CAPDELS and Sydney-Captions, respectively. BLEU-4 scores are reported.

| | CAPDELS | | | | | SydneyCaptions | | | | | |
|---|---|---|---|---|---|---|---|---|---|---|---|
| Method | 0-shot | 32 | 128 | 512 | 2048 | Full | 0-shot | 32 | 128 | 512 | 2048 | Full |
| SEMI | 0.41 | **46.52** | **53.57** | **56.38** | **59.31** | **61.27** | 0.45 | **43.88** | **41.87** | **47.38** | **49.18** | 48.03 |
| FT Projector | 0.02 | 31.26 | 41.58 | 49.71 | 53.93 | 58.11 | 0.01 | 39.18 | 37.55 | 44.64 | 47.26 | **49.71** |

SEMI is explicitly designed to require fine-tuning due to several key challenges. First, test modalities exhibit extreme distribution shifts from training modalities, spanning different encoders (audio), different domains (satellite and astronomical images), and entirely novel input spaces (molecules and IMU signals). Due to compounding challenges of modality-specific text distributions and complex patterns that might require iterative refinement beyond a single adapter generation step, fine-tuning serves as an essential step for integration.

Nonetheless, for completeness, we evaluate SEMI's zero-shot adaptation, that is, using only the generated LoRA adapters without any subsequent fine-tuning on the target modality.

As expected, zero-shot adaptation results are significantly worse than few-shot adaptations for both SEMI and FT Projector (Table 4). Nonetheless, SEMI gets a huge boost with fine-tuning on as few as 32 examples (e.g. $0.39 \rightarrow 46.01$), while FT Projector fails to show a similar gain ($0.02 \rightarrow 31.26$).

## H  CROSS-MODAL SIMILARITY OF EMBEDDINGS

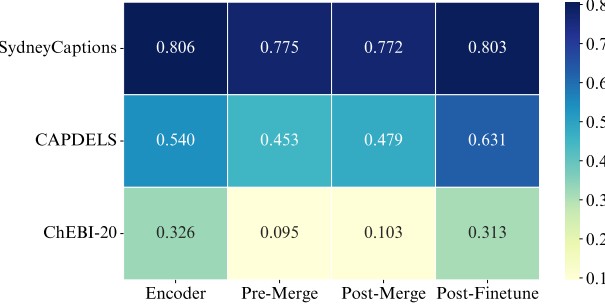

Figure 12: Linear CKA scores between modality embeddings at different stages and text embeddings. 'Encoder' embeddings are unprocessed. Embeddings projected with the pre-trained projector before and after merging with the adapters are labelled 'Pre-Merge' and 'Post-Merge', respectively. 'Post-Finetune' embeddings are extracted after fine-tuning of the merged projector. 2048 samples are used for ChEBI-20 and 128 for others.

We analyse the effect of different stages of our pipeline on the similarity between the embeddings of each unseen modality and the corresponding text. The similarity scores are obtained with Linear CKA (Kornblith et al., 2019) and shown in Figure 12 in the form of a heatmap. We find that the embeddings of text and each unseen modality are originally well aligned, but the modality embeddings are distorted during mapping into the LLM input space. However, the amount of distortion is reduced progressively as we incorporate adapters and further fine-tune the projector. Moreover, the effect of these two steps is more pronounced in the galaxy modality, as galaxy encoders lack text conditioning. The satellite encoders are already well aligned, which might explain the comparatively higher performance of training a projector from scratch. The molecule modality is arguably the most unique and challenging modality, which might explain the smaller effect of adapter merging on embedding alignment.

Table 5: Qualitative examples for SydneyCaptions dataset and ViT-L-14 encoder for methods trained with 128 samples

| Image | Ground Truth | Predictions |
|---|---|---|
| 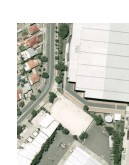 | This is an industrial area with many white buildings densely arranged while a residential area beside | **Ours:** This is an industrial area with some roads and many buildings

**FT Proj:** This image shows a residential area with some buildings and some roads

**Proj:** Some buildings there

**LoRA:** There is a residential area with some houses on a go road |
| 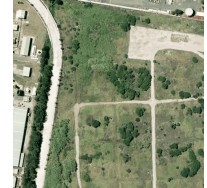 | This is a meadow with some green bushes on it while some roads passed by | **Ours:** This is a big meadow with some roads on it

**FT Proj:** There are some white flowers and some white sidewalks on the green field

**Proj:** Some green bushes and white sand on the beach

**LoRA:** Some sections of a farm are covered with green grass while others are divided by a straight road |

# I  QUALITATIVE EXAMPLES

In Table 5, we observe that the hypernetwork demonstrates a stronger ability to ground tasks compared to the fine-tuned projector, consistently generating correct outputs even if some details are omitted. While the fine-tuned projector often produces answers close to the ground truth, it occasionally makes mistakes and visibly retains remnants of its pre-trained state in the generated words—suggesting it has not fully adapted to the new task. *LoRA* frequently follows the *FT Projector*, omitting the industrial aspect of the image as well as hallucinating a "go road". This improved task grounding makes it applicable across diverse modalities, rather than simply memorising pre-training data.

Qualitative results demonstrate that the hypernetwork excels at task adaptation compared to other methods. Specifically, the hypernetwork consistently generates correct answers verbatim, while alternatives often introduce word choices absent from the training data, a clear sign of insufficient detachment from pre-trained knowledge. Furthermore, all of the competing approaches except *LoRA* failed to identify the merger occurring in the first image (as shown in Table 6), highlighting a limitation in their ability to perceive key details. Even when methods like *FT Projector* produce seemingly reasonable outputs, they tend to go beyond the specified task description, and the *Projector* and *LoRA* generations, while reflecting some truth, frequently hallucinate by adding irrelevant information such as incorrect details about "comfort", "presence", or colour.

In Table 7, we observe that the hypernetwork correctly answers the questions with a similar format to the ground truth while other baselines struggle to do so. In the first example, *FT Projector* incorrectly identifies the rotation as minimal, while other baselines produce nonsensical and unrelated outputs. In the second example, our approach is completely correct while the second-best *LoRA* baseline incorrectly hallucinates that the accelerometer data ranges from "-1 to 1 g" even though it stays the same – failing to finish the description within the maximum token limit.

While hypernetworks offer the most promising approach for integrating new modalities into LLMs, their generated responses are not without flaws; we observed errors such as misidentifying proanthocyanidins as cinnamaldehydes and incorrectly classifying GABA as a glycerophospholipid (see Table 8). The *FT Projector* baseline performs similarly – correctly identifying proanthocyanidin as a polyphenol, for example – but struggles with accurately listing the corresponding tree species. *LoRA* baseline captures salient patterns of the molecule, although it eventually provides non-factual information. Despite these limitations, the *Projector* baseline frequently exhibits significant inaccuracies. Importantly, our findings suggest that integrating more complex, knowledge-intensive modalities

requires larger training datasets; however, we demonstrate that leveraging information from other modalities can significantly ease this integration process.

Table 6: Qualitative examples for CAPDELS dataset and ConvNeXt-Tiny encoder for methods trained with 128 samples

| Image | Ground Truth | Predictions |
|---|---|---|
|  | The galaxy is smooth and rounded, with a shape that is in between completely round and cigar-shaped. There are signs that the galaxy is currently merging with another galaxy. | **Ours:** The galaxy is smooth and rounded, with a shape that is in between completely round and cigar-shaped. There are signs that the galaxy is currently merging with another galaxy.

**FT Proj:** The galaxy is smooth and rounded with no signs of a disk, having an immediate and sharp appearance.

**Proj:** The galaxy is smooth and rounded, with no signs of a disk. The galaxy is a small, showing no signs of merging or being in a state of merger. There are no signs that it has undergone any histological changes.

**LoRA:** The galaxy is smooth and rounded with no signs of a disk, having an in-between roundness, and shows signs of being present. There are indications that the galaxy is currently undergoing the process. |
|  | The galaxy is smooth and rounded, with a cigar-shaped appearance, and shows no signs of merging or tidal debris. | **Ours:** The galaxy is smooth and rounded, with a cigar-shaped appearance and shows no signs of merging or tidal debris.

**FT Proj:** The galaxy is smooth and evenly distributed, showing no signs of merging or tidal debris. The galaxy appears to be completely unrelated to others in the image.

**Proj:** The galaxy appears smooth and rounded, with a cigar-shaped appearance and has a peach-colored hue.

**LoRA:** The galaxy is smooth and rounded, with a cigar-shaped appearance, and shows no signs of merging or tidal debris. It appears more comfortable and has no indication of merging or tidal debris. |

Table 7: Qualitative examples for SensorCaps dataset and LIMU-BERT for methods trained with 128 samples

| Ground Truth | Predictions |
|---|---|
| Features: Walking, moderate intensity Narration: 1. The accelerometer data shows varying values along the x, y, and z axes, suggesting movement in multiple directions. 2. In the gyroscope data, there are consistent fluctuating values on all axes, indicating rotational movements from walking. 3. The positive and negative values in both sensor data indicate changes in acceleration and angular velocity during the walking activity. 4. The peaks and troughs in the data suggest steps taken in a rhythmic pattern. | **Ours:** In the gyroscope data, there are varying values of x, y, and z axes. The gyroscope data also shows fluctuating values of z, x, and y axes. The user is likely rotating their body while the gyroscope and accelerometer data capture the changes in orientation and movement. The accelerometer data may indicate changes in body position and movement. The fluctuations in gyroscope data suggest the user is moving or rotating in different directions.

**FT Proj:** Features in the description of the gyroscope readings suggest that the device is moving in a consistent pattern. The magnitude of the gyroscope data is not significant, indicating minimal rotation. The axis values are relatively close in value, indicating a stable movement in all axes. The accelerometer data shows minimal movement in all axes, consistent with a stationary device.

**Proj:** None, None, None, None.

**LoRA:** However, the accuracy is low. |
| Features: Stationary, subtle movements. Narration: The gyroscope data shows minimal variations around 0, indicating little rotational movement. Accelerometer data remains around 9.8 m/s², denoting no significant changes in acceleration. The consistent values suggest the user is standing still. Slight fluctuations in gyroscope and accelerometer readings imply subtle swaying or adjustments in posture while standing. | **Ours:** The gyroscope data shows minimal fluctuations, indicating minimal movement. The accelerometer data indicates consistent values, suggesting the user is standing still. The user's posture is likely stationary, as the gyroscope readings are close to zero and the accelerometer values are near the average weight. The slight variations in the accelerometer data may be attributed to minor adjustments in posture or slight movements while standing.

**FT Proj:** Note: No additional context is provided.

**Proj:** None, it's a non-physical activity.

**LoRA:** The symptoms match the characteristics of a person standing still with minimal physical activity. Here's a description of the gyroscope and accelerometer data:
1. Gyroscope data: - The x, y, and z axes show minimal changes, indicating no significant rotation or movement in these planes. The data is relatively stable, with slight variations, suggesting a stationary position.
2. Accelerometer data: - The x, y, and z axes show consistent changes, typically ranging from -1 to 1 g. This |

Table 8: Qualitative examples for ChEBI-20 dataset and MolCA for methods trained with 2048 samples

| Ground Truth | Predictions |
|---|---|
| **2D structure image:** 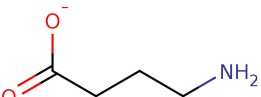 **Description:** The molecule is a gamma-amino acid anion resulting from the deprotonation of the carboxy group of gamma-aminobutyric acid. It has a role as a human metabolite. It derives from a butyrate. It is a conjugate base of a gamma-aminobutyric acid. | **Ours:** She is a species to which a molecule is a gamma-aminobutyric acid (GABA) conjugate that has been metabolized to a glycerol-3-phosphate conjugate. It is a glycerophospholipid and a member of the gamma-aminobutyric acid (GABA) conjugate family. It has a role as a human metabolite and a human metabolite. It is a major player in the metabolism of GABA. It is a conjugate acid of a gamma-aminobutyric acid (GABA) molecule. **FT Proj:** The molecule is a tetracyclic compound that consists of a 2,3,5-trisubstituted cyclohexene with a 2,3,5-trisubstituted 2,3,5-trimethylcyclohexene-1,4,5-diol and 2,3,5-trisubstituted 2,3,5-trimethylcyclohexene-1,4,5-dione. It has a role as a plant metabolite and a polyphenol. It derives from a 2,3,5-trisubstituted cyclohexene and a 2,3,5-trisubstituted 2,3,5-trimethylcyclohexene-1,4,5-diol. **Proj:** The molecule is a 2,3-diphenyl-2-pyridine-5-oxide derivative that is similar to 2,3-diphenyl-2-pyridine-5-oxide. **LoRA:** The molecule is a glucuronic acid derivative; it is a conjugate acid of 3-deoxy-D-glucuronic acid. It has a role as a human metabolite and a metabolite. It derives from a 3-deoxy-D-glucuronic acid. It is a conjugate acid of a 3-deoxy-D-glucuronic acid. |
| **2D structure image:** 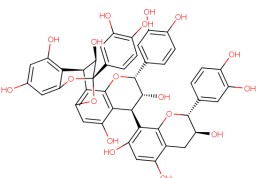 **Description:** The molecule is a proanthocyanidin found in Cinnamomum cassia. It has a role as a cyclooxygenase 2 inhibitor and a plant metabolite. | **Ours:** The molecule is a type of cinnamaldehyde found in the leaves of Cinnamomum aromaticum (Cinnamomum zeylanicum). It has a role as a plant metabolite and a cinnamaldehyde. **FT Proj:** The molecule is a polyphenol that is a secondary metabolite of the tree species Pinus spp. It is found in the bark of Pinus spp. and is composed of a mixture of various flavonoids and phenolic compounds. **Proj:** The molecule is a prostanoid derived from 6-deoxy-5,6,7,8-tetrahydro-2-methoxy-5,6,7,8-(2-methyl-9,10-phenyl)-2H-cyclopropeno-1,4,7,8(3,5)-tetraones. The molecule is a prostanoid and a flavonoid present in the plant Pterocarpus spinosus. It has a role as a flavonoid and a plant extract used in traditional medicine. **LoRA:** Alicin is a metabolite of aloe vera, a plant-based compound that is used in the treatment of hemorrhoids and other hemorrhagic conditions. It has a role as a metabolite and an ethyl ester. |

# J PSEUDOCODE

---

**Algorithm 1** Projector pre-training

---

**Input:** Modality encoders $\{\text{enc}_m\}_{m=1}^M$, LLM, training datasets $\{\mathcal{D}_m\}_{m=1}^M$, dataset instructions $\{\mathbf{i}_m\}_{m=1}^M$, projector proj with parameters $\psi$, initial projector parameters $\psi_{\text{init}}$, cross-entropy $H(\cdot)$

**Output:** Pre-trained projector parameters $\psi^\star$

1: $\psi \leftarrow \psi_{\text{init}}$
2: **while** not converged **do**
3:      $m \sim \{1, \ldots, M\}$                    ▷ Randomly sample a modality
4:      $\mathbf{x}, \mathbf{y} \sim \mathcal{D}_m$                 ▷ Get modality input and text output
5:      $\mathbf{z} \leftarrow \text{proj}_\psi(\text{enc}_m(\mathbf{x})) \oplus \mathbf{i}_m$
6:      $\ell \leftarrow H(\text{LLM}(\mathbf{z}), \mathbf{y})$              ▷ Calculate cross-entropy loss
7:      Update $\psi$ using $\nabla_\psi$ w.r.t. $\ell$
8: **end while**
9: $\psi^\star \leftarrow \psi$

---

**Algorithm 2** Hypernetwork training

---

**Input:** Modality encoders $\{\text{enc}_m\}_{m=1}^M$, text encoder $\text{enc}_{\text{text}}$, LLM, training datasets $\{\mathcal{D}_m\}_{m=1}^M$, hypernet hyp with parameters $\theta$, pre-trained projector proj with parameters $\psi^\star$, dataset instruction sets $\{\mathcal{I}_m\}_{m=1}^M$, initial hypernet parameters $\theta_{\text{init}}$, cross-entropy $H(\cdot)$

**Output:** Trained hypernetwork parameters $\theta^\star$

1: $\theta \leftarrow \theta_{\text{init}}$
2: **while** not converged **do**
3:      $m \sim \{1, \ldots, M\}$                    ▷ Randomly sample a modality
4:      $\mathbf{i}_m \sim \mathcal{I}_m$
5:      $\{\mathbf{x}_{\text{hyp}}, \mathbf{y}_{\text{hyp}}\}_1^S \sim \mathcal{D}_m$
6:      $\mathbf{x}_{\text{LLM}}, \mathbf{y}_{\text{LLM}} \sim \mathcal{D}_m$
7:      $\boldsymbol{Q} \sim \text{Haar}(\text{O}(h_{\text{enc}_m}))$            ▷ Sample orthogonal matrix
8:      $\mathbf{m}_{\text{LLM}} \leftarrow \boldsymbol{Q}\,\text{enc}_m(\mathbf{x}_{\text{LLM}})$
9:      $\mathbf{m}_{\text{hyp}} \leftarrow \boldsymbol{Q}\,\text{enc}_m(\mathbf{x}_{\text{hyp}})$       ▷ Extract embeddings with orthogonal transform
10:     $\delta \leftarrow \text{hyp}_\theta\left(\text{enc}_{\text{text}}(\mathbf{i}_m) \oplus [\mathbf{m}_{\text{hyp}} \oplus \text{enc}_{\text{text}}(\mathbf{y}_{\text{hyp}})]_1^S\right)$    ▷ Generate LoRA with hypernetwork
11:     $\psi' \leftarrow \psi^\star + \frac{\alpha}{r}\delta$               ▷ Combine projector and LoRA parameters
12:     $\mathbf{z} \leftarrow \text{proj}_{\psi'}(\mathbf{m}_{\text{LLM}})) \oplus \mathbf{i}_m$
13:     $\ell \leftarrow H(\text{LLM}(\mathbf{z}), \mathbf{y}_{\text{LLM}})$           ▷ Calculate loss
14:     Update $\theta$ using $\nabla_\theta$ w.r.t. $\ell$
15: **end while**
16: $\theta^\star \leftarrow \theta$

---

---

**Algorithm 3** Few-shot adaptation

---

**Input:** Test modality encoder $\text{enc}_m$, text encoder $\text{enc}_{\text{text}}$, LLM, training dataset $\mathcal{D}_m$, trained hypernet hyp with parameters $\theta^\star$, pre-trained projector proj with parameters $\psi^\star$, instruction $\mathbf{i}_m$, cross-entropy $H(\cdot)$

**Output:** Projector parameters $\psi'$

1: $\Delta \leftarrow \varnothing$
2: **for** $\{\mathbf{x}_{\text{hyp}}, \mathbf{y}_{\text{hyp}}\}_1^S \in \mathcal{D}_m$ **do**
3:      $\delta \leftarrow \text{hyp}_{\theta^\star}\left(\text{enc}_{\text{text}}(\mathbf{i}_m) \oplus [\text{enc}_m(\mathbf{x}_{\text{hyp}}) \oplus \text{enc}_{\text{text}}(\mathbf{y}_{\text{hyp}})]_1^S\right)$          ▷ Generate LoRA
4:      $\Delta \leftarrow \Delta \cup \delta$
5: **end for**
6: $\psi' \leftarrow \psi^\star + \frac{\alpha}{r}\bar{\Delta}$          ▷ Combine projector parameters and averaged LoRAs
7: **while** not converged **do**
8:      $\mathbf{x}_{\text{LLM}}, \mathbf{y}_{\text{LLM}} \sim \mathcal{D}_m$
9:      $\mathbf{z} \leftarrow \text{proj}_{\psi'}(\text{enc}_m(\mathbf{x}_{\text{LLM}})) \oplus \mathbf{i}_m$
10:     $\ell \leftarrow H(\text{LLM}(\mathbf{z}), \mathbf{y}_{\text{LLM}})$          ▷ Calculate loss
11:     Update $\psi'$ using $\nabla_{\psi'}$ w.r.t. $\ell$
12: **end while**

---

# K  RESULTS WITH ALL METRICS

We used the `allenai/scibert_scivocab_uncased` (Beltagy et al., 2019; Wolf et al., 2020) tokeniser for the ChEBI-20 dataset following Edwards et al. (2022), whereas we used a whitespace tokeniser (or metric-specific tokenisers) for calculating the metrics of the remaining datasets.

## K.1  LLAMA 3.1 8B INSTRUCT RESULTS

### K.1.1  SYDNEYCAPTIONS DATASET

Table 9: All results and metrics for the ViT-B-32 encoder on the SydneyCaptions dataset. We show the mean ± standard error calculated over three random seeds.

| Sample Size | Setup \ Metric | ROUGE-1 | ROUGE-2 | ROUGE-L | BLEU | METEOR | CIDEr |
|---|---|---|---|---|---|---|---|
| 32 | Ours | $59.23 \pm 0.41$ | $44.73 \pm 0.81$ | $56.84 \pm 0.55$ | $42.99 \pm 1.29$ | $56.42 \pm 0.71$ | $163.93 \pm 2.66$ |
| | FT Projector | $54.76 \pm 1.83$ | $37.82 \pm 2.32$ | $51.92 \pm 2.07$ | $36.24 \pm 2.10$ | $50.01 \pm 2.01$ | $146.25 \pm 8.50$ |
| | Projector | $58.25 \pm 1.24$ | $42.11 \pm 1.28$ | $55.39 \pm 1.43$ | $40.24 \pm 2.91$ | $52.72 \pm 0.65$ | $155.26 \pm 9.86$ |
| | LoRA | $45.68 \pm 0.27$ | $26.39 \pm 1.22$ | $42.43 \pm 0.67$ | $23.85 \pm 1.98$ | $40.51 \pm 1.10$ | $79.76 \pm 9.10$ |
| 128 | Ours | $63.67 \pm 0.39$ | $47.48 \pm 0.67$ | $60.95 \pm 0.51$ | $42.95 \pm 0.41$ | $61.97 \pm 0.67$ | $184.61 \pm 4.76$ |
| | FT Projector | $63.12 \pm 1.13$ | $46.68 \pm 1.36$ | $60.28 \pm 1.15$ | $43.43 \pm 1.69$ | $60.29 \pm 1.09$ | $176.34 \pm 6.99$ |
| | Projector | $60.91 \pm 1.04$ | $44.62 \pm 0.96$ | $57.85 \pm 1.13$ | $39.68 \pm 1.79$ | $58.90 \pm 0.87$ | $155.26 \pm 5.73$ |
| | LoRA | $58.98 \pm 0.89$ | $40.93 \pm 0.91$ | $55.58 \pm 0.77$ | $37.93 \pm 1.22$ | $56.46 \pm 0.60$ | $138.37 \pm 10.59$ |
| 512 | Ours | $64.81 \pm 0.59$ | $49.58 \pm 0.95$ | $61.95 \pm 0.71$ | $46.43 \pm 1.19$ | $63.58 \pm 0.66$ | $201.54 \pm 4.87$ |
| | FT Projector | $63.14 \pm 0.29$ | $47.07 \pm 0.53$ | $59.77 \pm 0.29$ | $44.05 \pm 0.83$ | $61.28 \pm 0.38$ | $189.34 \pm 5.10$ |
| | Projector | $63.86 \pm 1.17$ | $48.01 \pm 1.52$ | $60.73 \pm 1.31$ | $44.79 \pm 1.96$ | $62.00 \pm 1.56$ | $182.80 \pm 6.78$ |
| | LoRA | $61.19 \pm 0.39$ | $43.41 \pm 0.70$ | $57.53 \pm 0.40$ | $39.35 \pm 1.00$ | $58.33 \pm 0.26$ | $168.52 \pm 1.82$ |
| 2048 | Ours | $67.27 \pm 0.40$ | $52.03 \pm 0.46$ | $64.46 \pm 0.45$ | $49.53 \pm 0.93$ | $65.65 \pm 0.28$ | $205.38 \pm 6.94$ |
| | FT Projector | $68.14 \pm 0.31$ | $51.90 \pm 0.55$ | $64.69 \pm 0.43$ | $50.55 \pm 1.35$ | $65.03 \pm 0.40$ | $202.41 \pm 17.35$ |
| | Projector | $66.55 \pm 0.93$ | $50.86 \pm 1.17$ | $63.01 \pm 0.92$ | $48.45 \pm 1.18$ | $64.62 \pm 1.10$ | $206.76 \pm 9.66$ |
| | LoRA | $65.76 \pm 1.20$ | $49.75 \pm 1.38$ | $62.60 \pm 1.22$ | $46.24 \pm 1.88$ | $63.81 \pm 0.89$ | $196.85 \pm 9.91$ |
| 2470 | Ours | $67.39 \pm 0.78$ | $51.30 \pm 0.94$ | $64.47 \pm 0.92$ | $48.31 \pm 1.41$ | $65.52 \pm 0.60$ | $214.05 \pm 9.05$ |
| | FT Projector | $68.26 \pm 0.80$ | $52.04 \pm 0.92$ | $64.34 \pm 0.78$ | $49.77 \pm 1.66$ | $66.14 \pm 0.81$ | $218.18 \pm 3.95$ |
| | Projector | $66.19 \pm 0.65$ | $49.43 \pm 0.77$ | $62.55 \pm 0.53$ | $45.53 \pm 1.73$ | $64.82 \pm 0.79$ | $201.73 \pm 8.99$ |
| | LoRA | $66.70 \pm 0.96$ | $50.22 \pm 1.27$ | $63.52 \pm 1.08$ | $47.33 \pm 0.78$ | $63.74 \pm 1.53$ | $199.11 \pm 11.84$ |

Table 10: All results and metrics for the ViT-L-14 encoder on the SydneyCaptions dataset. We show the mean ± standard error calculated over three random seeds.

| Sample Size | Setup \ Metric | ROUGE-1 | ROUGE-2 | ROUGE-L | BLEU | METEOR | CIDEr |
|---|---|---|---|---|---|---|---|
| 32 | Ours | 57.30 ± 0.84 | 40.73 ± 1.12 | 52.79 ± 1.02 | 35.51 ± 1.16 | 53.88 ± 1.10 | 132.83 ± 4.72 |
| | FT Projector | 52.60 ± 0.57 | 36.07 ± 0.67 | 49.02 ± 0.82 | 35.26 ± 1.61 | 49.54 ± 0.71 | 100.72 ± 7.59 |
| | Projector | 53.81 ± 1.72 | 36.89 ± 2.49 | 50.37 ± 1.98 | 35.47 ± 3.52 | 49.73 ± 1.94 | 122.45 ± 12.01 |
| | LoRA | 48.46 ± 0.37 | 29.15 ± 0.58 | 44.91 ± 0.42 | 24.71 ± 1.72 | 44.52 ± 0.87 | 62.11 ± 8.35 |
| 128 | Ours | 62.18 ± 0.54 | 46.00 ± 0.75 | 59.42 ± 0.76 | 44.13 ± 0.92 | 59.04 ± 0.65 | 171.15 ± 4.62 |
| | FT Projector | 57.77 ± 0.66 | 41.45 ± 0.95 | 54.31 ± 0.73 | 40.07 ± 0.57 | 54.35 ± 0.57 | 147.71 ± 2.34 |
| | Projector | 60.67 ± 0.97 | 44.44 ± 0.97 | 57.23 ± 0.84 | 41.02 ± 1.12 | 57.91 ± 1.29 | 158.41 ± 5.59 |
| | LoRA | 54.92 ± 0.61 | 36.48 ± 1.01 | 51.01 ± 0.68 | 31.33 ± 0.65 | 51.90 ± 1.00 | 119.79 ± 2.41 |
| 512 | Ours | 63.70 ± 0.84 | 46.82 ± 1.21 | 60.86 ± 0.97 | 44.04 ± 1.59 | 60.19 ± 0.93 | 177.33 ± 4.40 |
| | FT Projector | 63.89 ± 0.91 | 46.60 ± 1.10 | 60.31 ± 1.05 | 43.26 ± 1.29 | 60.31 ± 0.91 | 171.88 ± 2.53 |
| | Projector | 60.98 ± 1.34 | 43.35 ± 1.41 | 57.54 ± 1.57 | 41.00 ± 1.45 | 57.38 ± 1.76 | 164.98 ± 4.80 |
| | LoRA | 59.20 ± 2.48 | 38.79 ± 3.60 | 54.97 ± 2.76 | 34.01 ± 3.85 | 55.01 ± 2.24 | 131.20 ± 22.58 |
| 2048 | Ours | 66.12 ± 1.00 | 50.32 ± 1.12 | 62.80 ± 1.15 | 47.05 ± 1.41 | 64.05 ± 1.27 | 203.55 ± 4.00 |
| | FT Projector | 67.64 ± 0.99 | 51.27 ± 1.18 | 64.08 ± 1.10 | 48.58 ± 1.77 | 65.45 ± 0.75 | 207.01 ± 8.90 |
| | Projector | 63.83 ± 2.12 | 47.01 ± 2.38 | 60.38 ± 2.37 | 43.97 ± 3.63 | 62.04 ± 1.60 | 175.30 ± 17.07 |
| | LoRA | 64.13 ± 2.05 | 48.24 ± 2.88 | 61.09 ± 2.31 | 44.00 ± 2.43 | 62.47 ± 2.28 | 180.86 ± 11.29 |
| 2470 | Ours | 67.80 ± 0.88 | 51.88 ± 1.38 | 64.67 ± 1.02 | 49.86 ± 1.92 | 65.36 ± 1.36 | 209.76 ± 1.67 |
| | FT Projector | 66.76 ± 0.55 | 50.54 ± 1.03 | 63.15 ± 0.81 | 48.19 ± 1.34 | 64.99 ± 0.51 | 189.53 ± 12.49 |
| | Projector | 65.10 ± 1.07 | 48.68 ± 1.31 | 62.35 ± 1.06 | 46.04 ± 1.11 | 61.92 ± 2.01 | 193.24 ± 9.08 |
| | LoRA | 65.55 ± 0.95 | 48.32 ± 0.95 | 61.98 ± 0.92 | 44.82 ± 1.57 | 63.34 ± 0.81 | 192.59 ± 6.11 |

Table 11: All results and metrics for the RN-50 encoder on the SydneyCaptions dataset. We show the mean ± standard error calculated over three random seeds.

| Sample Size | Setup \ Metric | ROUGE-1 | ROUGE-2 | ROUGE-L | BLEU | METEOR | CIDEr |
|---|---|---|---|---|---|---|---|
| 32 | Ours | 59.49 ± 0.85 | 44.70 ± 1.23 | 56.91 ± 1.03 | 43.88 ± 1.27 | 55.15 ± 0.87 | 165.44 ± 2.06 |
| | FT Projector | 55.53 ± 0.61 | 39.41 ± 1.14 | 52.75 ± 0.68 | 39.18 ± 1.31 | 51.58 ± 0.79 | 140.46 ± 6.76 |
| | Projector | 50.68 ± 1.95 | 33.06 ± 2.52 | 47.45 ± 2.16 | 31.66 ± 2.99 | 46.20 ± 1.71 | 94.34 ± 16.68 |
| | LoRA | 45.65 ± 0.43 | 27.34 ± 0.54 | 42.76 ± 0.50 | 24.72 ± 0.89 | 41.30 ± 0.56 | 70.76 ± 5.48 |
| 128 | Ours | 60.99 ± 0.16 | 44.20 ± 0.21 | 58.02 ± 0.18 | 41.87 ± 1.00 | 58.93 ± 0.35 | 157.43 ± 2.32 |
| | FT Projector | 58.23 ± 0.53 | 39.79 ± 0.49 | 54.60 ± 0.57 | 37.55 ± 0.45 | 54.38 ± 0.56 | 128.26 ± 3.65 |
| | Projector | 56.04 ± 1.74 | 36.95 ± 1.72 | 52.05 ± 1.67 | 33.55 ± 2.67 | 53.18 ± 2.06 | 112.26 ± 11.31 |
| | LoRA | 52.98 ± 1.06 | 32.94 ± 1.45 | 49.64 ± 1.06 | 30.37 ± 1.10 | 49.10 ± 1.80 | 86.30 ± 5.03 |
| 512 | Ours | 66.08 ± 0.86 | 49.81 ± 0.97 | 63.09 ± 0.89 | 47.38 ± 0.96 | 64.25 ± 1.23 | 197.13 ± 5.00 |
| | FT Projector | 63.57 ± 0.47 | 46.51 ± 0.72 | 59.88 ± 0.72 | 44.64 ± 0.89 | 60.91 ± 0.71 | 174.32 ± 3.81 |
| | Projector | 62.79 ± 0.98 | 45.50 ± 1.55 | 58.92 ± 1.37 | 43.61 ± 1.86 | 60.26 ± 1.01 | 164.82 ± 6.49 |
| | LoRA | 61.98 ± 1.19 | 44.33 ± 1.27 | 58.22 ± 1.06 | 40.74 ± 1.44 | 60.70 ± 1.05 | 145.76 ± 11.63 |
| 2048 | Ours | 67.33 ± 0.92 | 51.54 ± 1.32 | 64.36 ± 1.14 | 49.18 ± 1.53 | 66.03 ± 1.00 | 215.96 ± 8.06 |
| | FT Projector | 65.16 ± 0.49 | 49.04 ± 0.61 | 62.13 ± 0.40 | 47.26 ± 0.35 | 63.28 ± 0.75 | 206.04 ± 8.13 |
| | Projector | 62.83 ± 2.31 | 45.63 ± 3.12 | 59.17 ± 2.35 | 41.44 ± 4.65 | 61.08 ± 2.27 | 180.11 ± 21.68 |
| | LoRA | 63.40 ± 0.50 | 46.12 ± 0.22 | 59.80 ± 0.41 | 43.79 ± 0.57 | 61.05 ± 0.79 | 191.53 ± 16.98 |
| 2470 | Ours | 66.27 ± 0.78 | 50.46 ± 0.83 | 62.89 ± 0.85 | 48.03 ± 1.02 | 64.41 ± 0.93 | 218.06 ± 5.87 |
| | FT Projector | 66.31 ± 0.51 | 50.79 ± 0.57 | 63.33 ± 0.59 | 49.71 ± 0.64 | 63.69 ± 0.84 | 207.86 ± 10.02 |
| | Projector | 62.48 ± 2.98 | 45.49 ± 3.76 | 59.18 ± 3.17 | 41.92 ± 4.27 | 60.52 ± 2.73 | 158.20 ± 25.58 |
| | LoRA | 66.74 ± 1.24 | 50.86 ± 1.45 | 63.95 ± 1.14 | 49.36 ± 1.83 | 64.65 ± 1.74 | 191.78 ± 9.51 |

### K.1.2 RSVQA DATASET

Table 12: All results and metrics for the ViT-L-14 encoder on the RSVQA dataset. We show the mean ± standard error calculated over three random seeds.

| Sample Size | Setup \ Metric | Exact Match | F1 |
|---|---|---|---|
| 32 | Ours | 69.87 ± 1.46 | 69.93 ± 1.47 |
| | FT Projector | 24.88 ± 1.55 | 34.47 ± 2.97 |
| | Projector | 45.96 ± 6.54 | 53.05 ± 5.36 |
| | LoRA | 24.04 ± 1.31 | 31.15 ± 2.00 |
| 128 | Ours | 77.27 ± 0.79 | 77.27 ± 0.79 |
| | FT Projector | 32.12 ± 0.88 | 44.41 ± 1.04 |
| | Projector | 48.44 ± 11.02 | 55.53 ± 9.69 |
| | LoRA | 27.56 ± 0.72 | 36.87 ± 1.22 |
| 512 | Ours | 82.67 ± 1.30 | 82.67 ± 1.30 |
| | FT Projector | 31.48 ± 0.86 | 45.13 ± 1.39 |
| | Projector | 63.08 ± 9.84 | 67.32 ± 7.45 |
| | LoRA | 26.12 ± 0.90 | 34.76 ± 1.28 |
| 2048 | Ours | 84.13 ± 0.47 | 84.49 ± 0.66 |
| | FT Projector | 34.60 ± 0.94 | 50.37 ± 1.51 |
| | Projector | 58.76 ± 10.08 | 65.21 ± 7.56 |
| | LoRA | 25.20 ± 2.06 | 34.57 ± 3.11 |
| 2860 | Ours | 83.13 ± 1.20 | 83.44 ± 1.33 |
| | FT Projector | 35.48 ± 0.78 | 50.48 ± 1.20 |
| | Projector | 65.48 ± 10.94 | 69.33 ± 9.18 |
| | LoRA | 27.40 ± 1.22 | 38.55 ± 1.73 |

### K.1.3 CAPDELS DATASET

Table 13: All results and metrics for the ConvNeXt-Nano encoder on the CAPDELS dataset. We show the mean ± standard error calculated over three random seeds.

| Sample Size | Setup \ Metric | ROUGE-1 | ROUGE-2 | ROUGE-L | BLEU | METEOR | CIDEr |
|---|---|---|---|---|---|---|---|
| 32 | Ours | 66.19 ± 0.41 | 49.52 ± 0.66 | 58.98 ± 0.66 | 46.03 ± 0.75 | 62.10 ± 0.54 | 215.76 ± 4.61 |
| | FT Projector | 56.45 ± 0.51 | 36.95 ± 0.86 | 48.19 ± 0.63 | 31.84 ± 1.08 | 49.90 ± 0.43 | 104.63 ± 7.48 |
| | Projector | 49.71 ± 1.96 | 31.01 ± 2.23 | 42.77 ± 2.04 | 26.51 ± 2.22 | 43.23 ± 1.53 | 87.04 ± 12.03 |
| | LoRA | 46.75 ± 1.11 | 24.89 ± 2.02 | 38.18 ± 1.50 | 19.73 ± 1.57 | 38.61 ± 1.46 | 30.26 ± 5.83 |
| 128 | Ours | 69.71 ± 0.24 | 53.28 ± 0.54 | 62.26 ± 0.36 | 48.45 ± 0.48 | 66.57 ± 0.20 | 247.60 ± 10.07 |
| | FT Projector | 63.37 ± 0.58 | 45.29 ± 0.89 | 55.28 ± 0.78 | 39.17 ± 1.02 | 59.16 ± 0.29 | 188.85 ± 6.96 |
| | Projector | 60.76 ± 0.68 | 42.76 ± 0.74 | 53.31 ± 0.77 | 38.46 ± 1.25 | 55.15 ± 0.64 | 168.02 ± 5.11 |
| | LoRA | 60.66 ± 0.21 | 41.38 ± 0.04 | 52.16 ± 0.13 | 34.51 ± 0.23 | 56.74 ± 0.32 | 155.97 ± 3.71 |
| 512 | Ours | 74.91 ± 0.40 | 60.35 ± 0.53 | 67.91 ± 0.43 | 54.74 ± 0.83 | 72.65 ± 0.51 | 315.42 ± 4.78 |
| | FT Projector | 70.26 ± 0.03 | 54.64 ± 0.19 | 63.32 ± 0.10 | 49.41 ± 0.31 | 66.73 ± 0.20 | 265.62 ± 2.68 |
| | Projector | 69.85 ± 0.67 | 54.27 ± 0.49 | 62.87 ± 0.48 | 48.62 ± 0.02 | 66.50 ± 1.15 | 270.58 ± 3.18 |
| | LoRA | 70.25 ± 0.12 | 54.35 ± 0.46 | 63.10 ± 0.32 | 50.56 ± 0.81 | 66.08 ± 0.15 | 262.91 ± 1.51 |
| 2048 | Ours | 76.13 ± 0.78 | 62.17 ± 1.11 | 69.44 ± 1.14 | 57.83 ± 2.78 | 74.16 ± 0.69 | 329.64 ± 11.05 |
| | FT Projector | 73.91 ± 1.49 | 58.92 ± 2.26 | 66.98 ± 1.69 | 55.35 ± 1.68 | 70.90 ± 1.61 | 286.94 ± 25.94 |
| | Projector | 74.05 ± 1.48 | 59.12 ± 2.30 | 67.27 ± 1.84 | 55.08 ± 3.25 | 71.68 ± 1.05 | 286.71 ± 19.48 |
| | LoRA | 74.28 ± 0.81 | 59.09 ± 1.41 | 66.98 ± 1.12 | 55.28 ± 1.59 | 71.42 ± 0.94 | 292.11 ± 12.21 |
| 4344 | Ours | 77.11 ± 0.36 | 63.44 ± 0.55 | 70.75 ± 0.47 | 59.95 ± 0.45 | 75.04 ± 0.42 | 336.61 ± 8.49 |
| | FT Projector | 75.52 ± 0.03 | 61.17 ± 0.24 | 68.88 ± 0.15 | 58.70 ± 0.20 | 72.99 ± 0.21 | 308.84 ± 2.38 |
| | Projector | 75.69 ± 0.10 | 61.43 ± 0.31 | 69.15 ± 0.12 | 58.51 ± 0.39 | 73.40 ± 0.14 | 319.89 ± 4.13 |
| | LoRA | 75.69 ± 0.35 | 61.51 ± 0.51 | 68.99 ± 0.41 | 58.16 ± 0.59 | 73.01 ± 0.49 | 321.52 ± 8.95 |

Table 14: All results and metrics for the ConvNeXt-Tiny encoder on the CAPDELS dataset. We show the mean ± standard error calculated over three random seeds.

| Sample Size | Setup \ Metric | ROUGE-1 | ROUGE-2 | ROUGE-L | BLEU | METEOR | CIDEr |
|---|---|---|---|---|---|---|---|
| 32 | Ours | 66.79 ± 0.46 | 50.40 ± 0.96 | 59.80 ± 0.75 | 46.01 ± 0.58 | 62.54 ± 0.54 | 219.97 ± 5.52 |
| | FT Projector | 56.39 ± 0.53 | 37.45 ± 0.37 | 48.65 ± 0.41 | 32.92 ± 0.56 | 50.26 ± 0.74 | 94.00 ± 5.16 |
| | Projector | 47.00 ± 2.81 | 28.43 ± 3.11 | 40.38 ± 2.78 | 23.95 ± 2.64 | 39.34 ± 2.90 | 61.45 ± 8.38 |
| | LoRA | 43.96 ± 0.72 | 22.07 ± 1.07 | 35.35 ± 0.90 | 17.26 ± 1.10 | 36.61 ± 0.46 | 14.74 ± 3.98 |
| 128 | Ours | 71.48 ± 0.36 | 56.29 ± 0.44 | 64.86 ± 0.36 | 52.72 ± 0.69 | 68.36 ± 0.35 | 267.19 ± 9.72 |
| | FT Projector | 63.87 ± 0.58 | 46.16 ± 0.65 | 56.08 ± 0.62 | 39.04 ± 1.12 | 59.78 ± 0.77 | 186.83 ± 1.87 |
| | Projector | 61.92 ± 1.18 | 44.59 ± 1.61 | 54.61 ± 1.31 | 39.86 ± 1.83 | 56.58 ± 1.24 | 182.55 ± 15.27 |
| | LoRA | 60.90 ± 0.38 | 43.33 ± 0.61 | 53.44 ± 0.39 | 37.48 ± 0.69 | 55.76 ± 0.82 | 165.87 ± 5.92 |
| 512 | Ours | 75.38 ± 0.49 | 61.29 ± 0.42 | 68.77 ± 0.32 | 58.05 ± 0.54 | 72.41 ± 0.50 | 317.93 ± 7.09 |
| | FT Projector | 71.20 ± 0.36 | 55.64 ± 0.44 | 64.19 ± 0.38 | 50.75 ± 0.51 | 67.90 ± 0.28 | 269.92 ± 3.72 |
| | Projector | 68.18 ± 2.68 | 51.96 ± 3.80 | 60.78 ± 2.84 | 46.85 ± 2.91 | 64.63 ± 3.31 | 237.67 ± 44.88 |
| | LoRA | 68.92 ± 0.55 | 52.41 ± 0.56 | 61.46 ± 0.43 | 47.00 ± 1.03 | 65.40 ± 0.52 | 260.03 ± 7.12 |
| 2048 | Ours | 76.32 ± 0.29 | 62.16 ± 0.34 | 69.73 ± 0.48 | 58.17 ± 0.64 | 74.36 ± 0.48 | 327.99 ± 4.57 |
| | FT Projector | 73.94 ± 0.44 | 59.24 ± 0.74 | 67.18 ± 0.66 | 56.97 ± 1.80 | 70.42 ± 0.52 | 301.96 ± 4.35 |
| | Projector | 72.12 ± 0.03 | 56.57 ± 0.18 | 65.15 ± 0.26 | 53.39 ± 1.12 | 68.78 ± 0.26 | 267.34 ± 12.28 |
| | LoRA | 74.41 ± 0.38 | 59.72 ± 0.45 | 67.65 ± 0.37 | 55.49 ± 0.62 | 72.08 ± 0.43 | 304.34 ± 11.17 |
| 4344 | Ours | 77.46 ± 0.20 | 64.13 ± 0.29 | 71.09 ± 0.27 | 60.84 ± 1.02 | 75.16 ± 0.16 | 349.75 ± 2.44 |
| | FT Projector | 75.45 ± 0.44 | 60.96 ± 0.49 | 68.61 ± 0.55 | 56.60 ± 1.43 | 73.12 ± 0.18 | 236.56 ± 76.98 |
| | Projector | 74.96 ± 0.59 | 60.24 ± 0.92 | 68.21 ± 0.86 | 57.05 ± 1.56 | 72.00 ± 0.53 | 302.32 ± 13.56 |
| | LoRA | 76.28 ± 0.17 | 62.23 ± 0.18 | 69.79 ± 0.23 | 58.95 ± 0.53 | 73.91 ± 0.43 | 329.78 ± 3.19 |

Table 15: All results and metrics for the ConvNeXt-Base encoder on the CAPDELS dataset. We show the mean ± standard error calculated over three random seeds.

| Sample Size | Setup \ Metric | ROUGE-1 | ROUGE-2 | ROUGE-L | BLEU | METEOR | CIDEr |
|---|---|---|---|---|---|---|---|
| 32 | Ours | 66.98 ± 0.34 | 51.12 ± 0.19 | 60.36 ± 0.34 | 46.52 ± 0.26 | 62.25 ± 0.55 | 231.03 ± 4.55 |
| | FT Projector | 55.77 ± 0.33 | 36.49 ± 0.49 | 47.63 ± 0.38 | 31.26 ± 0.75 | 49.68 ± 0.38 | 105.70 ± 7.56 |
| | Projector | 51.34 ± 1.49 | 32.48 ± 1.89 | 44.20 ± 1.54 | 26.80 ± 1.47 | 43.92 ± 1.70 | 89.78 ± 7.57 |
| | LoRA | 43.32 ± 0.28 | 22.49 ± 0.56 | 34.69 ± 0.49 | 18.48 ± 0.47 | 36.59 ± 0.21 | 16.30 ± 6.21 |
| 128 | Ours | 72.14 ± 0.32 | 57.04 ± 0.44 | 65.55 ± 0.37 | 53.57 ± 0.79 | 68.84 ± 0.36 | 274.00 ± 2.80 |
| | FT Projector | 65.35 ± 0.36 | 48.13 ± 0.50 | 57.67 ± 0.47 | 41.58 ± 0.63 | 61.45 ± 0.25 | 198.49 ± 7.28 |
| | Projector | 61.94 ± 0.21 | 44.63 ± 0.23 | 54.61 ± 0.18 | 39.35 ± 0.66 | 57.39 ± 0.57 | 191.78 ± 1.90 |
| | LoRA | 62.77 ± 0.76 | 44.51 ± 0.91 | 54.54 ± 0.92 | 37.17 ± 1.33 | 59.08 ± 0.40 | 182.73 ± 10.37 |
| 512 | Ours | 74.84 ± 0.29 | 60.34 ± 0.36 | 68.20 ± 0.38 | 56.38 ± 1.28 | 72.16 ± 0.18 | 316.24 ± 2.04 |
| | FT Projector | 69.86 ± 0.59 | 53.96 ± 0.53 | 62.86 ± 0.52 | 49.71 ± 0.10 | 66.31 ± 0.68 | 258.96 ± 8.31 |
| | Projector | 69.53 ± 0.41 | 53.79 ± 0.64 | 62.51 ± 0.58 | 49.48 ± 1.20 | 65.88 ± 0.22 | 264.27 ± 2.98 |
| | LoRA | 69.23 ± 0.42 | 53.08 ± 0.35 | 61.97 ± 0.29 | 48.06 ± 0.71 | 65.67 ± 0.35 | 247.03 ± 6.10 |
| 2048 | Ours | 76.45 ± 0.35 | 62.49 ± 0.39 | 70.16 ± 0.18 | 59.31 ± 0.32 | 74.07 ± 0.45 | 328.52 ± 0.39 |
| | FT Projector | 73.16 ± 0.26 | 57.74 ± 0.59 | 66.19 ± 0.55 | 53.93 ± 0.91 | 70.49 ± 0.27 | 277.44 ± 1.02 |
| | Projector | 72.21 ± 1.34 | 56.12 ± 2.25 | 64.81 ± 1.49 | 51.31 ± 2.91 | 69.85 ± 1.00 | 264.20 ± 28.54 |
| | LoRA | 75.07 ± 0.70 | 60.54 ± 0.85 | 68.27 ± 0.68 | 57.60 ± 1.19 | 72.07 ± 0.75 | 308.78 ± 5.69 |
| 4344 | Ours | 76.87 ± 0.63 | 63.28 ± 1.00 | 70.75 ± 0.78 | 61.27 ± 1.04 | 74.10 ± 0.87 | 336.19 ± 4.67 |
| | FT Projector | 75.90 ± 0.26 | 61.78 ± 0.62 | 69.43 ± 0.57 | 58.11 ± 0.87 | 73.59 ± 0.19 | 319.13 ± 1.52 |
| | Projector | 75.23 ± 0.96 | 60.76 ± 1.17 | 68.17 ± 1.21 | 56.88 ± 0.33 | 72.44 ± 1.64 | 315.71 ± 10.34 |
| | LoRA | 75.66 ± 0.23 | 61.44 ± 0.21 | 69.19 ± 0.16 | 59.07 ± 0.37 | 72.74 ± 0.35 | 315.63 ± 1.14 |

### K.1.4 SENSORCAPS DATASET

Table 16: All results and metrics for the LIMU-BERT encoder on the SensorCaps dataset. We show the mean ± standard error calculated over three random seeds.

| Sample Size | Setup \ Metric | ROUGE-1 | ROUGE-2 | ROUGE-L | BLEU | METEOR |
|---|---|---|---|---|---|---|
| 32 | Ours | 38.92 ± 0.55 | 11.61 ± 0.50 | 23.11 ± 0.80 | 8.46 ± 0.71 | 29.78 ± 0.82 |
| | FT Projector | 34.80 ± 3.12 | 9.95 ± 1.13 | 21.35 ± 1.65 | 7.15 ± 1.11 | 26.91 ± 2.88 |
| | Projector | 23.39 ± 1.95 | 4.69 ± 1.35 | 14.99 ± 1.03 | 3.09 ± 0.37 | 17.16 ± 0.68 |
| | LoRA | 32.92 ± 0.33 | 8.97 ± 0.26 | 19.71 ± 0.26 | 6.18 ± 0.04 | 25.02 ± 0.35 |
| 128 | Ours | 48.36 ± 0.83 | 18.54 ± 0.71 | 30.63 ± 0.61 | 15.66 ± 0.62 | 39.49 ± 0.83 |
| | FT Projector | 44.98 ± 0.78 | 15.48 ± 0.35 | 27.56 ± 0.43 | 12.71 ± 0.48 | 35.80 ± 0.72 |
| | Projector | 35.31 ± 4.27 | 10.85 ± 2.55 | 21.68 ± 2.97 | 8.18 ± 2.19 | 26.78 ± 3.90 |
| | LoRA | 43.74 ± 0.75 | 15.09 ± 0.19 | 27.14 ± 0.47 | 12.18 ± 0.31 | 34.21 ± 0.69 |
| 512 | Ours | 49.46 ± 0.16 | 19.61 ± 0.22 | 31.17 ± 0.27 | 16.24 ± 0.27 | 40.31 ± 0.41 |
| | FT Projector | 47.83 ± 0.51 | 18.21 ± 0.58 | 30.57 ± 0.53 | 14.80 ± 0.48 | 38.27 ± 0.30 |
| | Projector | 45.41 ± 0.55 | 17.10 ± 0.29 | 28.54 ± 0.02 | 12.84 ± 0.59 | 34.77 ± 1.23 |
| | LoRA | 47.84 ± 0.28 | 18.93 ± 0.37 | 30.80 ± 0.40 | 15.54 ± 0.10 | 38.20 ± 0.41 |
| 1670 | Ours | 49.62 ± 0.48 | 20.19 ± 0.36 | 31.68 ± 0.62 | 17.08 ± 0.83 | 39.96 ± 0.98 |
| | FT Projector | 48.83 ± 0.36 | 19.43 ± 0.16 | 31.01 ± 0.21 | 15.96 ± 0.30 | 39.35 ± 0.49 |
| | Projector | 45.48 ± 1.10 | 17.37 ± 0.59 | 28.42 ± 0.99 | 13.22 ± 0.94 | 35.03 ± 1.59 |
| | LoRA | 46.83 ± 1.35 | 18.16 ± 0.75 | 29.85 ± 0.96 | 14.42 ± 0.92 | 36.84 ± 1.85 |

### K.1.5 OPENSQA DATASET

Table 17: All results and metrics for the LIMU-BERT encoder on the OpenSQA dataset. We show the mean ± standard error calculated over three random seeds.

| Sample Size | Setup \ Metric | ROUGE-1 | ROUGE-2 | ROUGE-L | BLEU | METEOR |
|---|---|---|---|---|---|---|
| 32 | Ours | 27.25 ± 0.32 | 7.43 ± 0.11 | 19.06 ± 0.12 | 3.84 ± 0.24 | 24.50 ± 0.29 |
| | FT Projector | 26.22 ± 0.10 | 7.41 ± 0.03 | 17.95 ± 0.05 | 3.33 ± 0.06 | 26.35 ± 0.21 |
| | Projector | 26.43 ± 0.35 | 7.22 ± 0.25 | 18.25 ± 0.17 | 3.26 ± 0.24 | 25.46 ± 0.42 |
| | LoRA | 26.24 ± 0.18 | 7.33 ± 0.11 | 17.88 ± 0.14 | 3.16 ± 0.06 | 26.45 ± 0.29 |
| 128 | Ours | 27.52 ± 0.64 | 7.64 ± 0.04 | 18.89 ± 0.02 | 4.06 ± 0.05 | 24.84 ± 0.89 |
| | FT Projector | 25.57 ± 0.05 | 7.14 ± 0.10 | 17.74 ± 0.11 | 3.21 ± 0.03 | 25.93 ± 0.13 |
| | Projector | 27.16 ± 0.67 | 7.48 ± 0.31 | 18.87 ± 0.68 | 3.73 ± 0.28 | 25.02 ± 0.28 |
| | LoRA | 26.21 ± 0.16 | 7.39 ± 0.07 | 18.02 ± 0.08 | 3.21 ± 0.04 | 26.55 ± 0.18 |
| 512 | Ours | 27.94 ± 1.24 | 8.65 ± 0.11 | 20.12 ± 0.36 | 4.66 ± 0.28 | 23.98 ± 1.51 |
| | FT Projector | 26.73 ± 0.18 | 7.61 ± 0.07 | 18.53 ± 0.16 | 3.51 ± 0.06 | 27.03 ± 0.09 |
| | Projector | 28.70 ± 0.54 | 8.15 ± 0.32 | 19.66 ± 0.49 | 3.99 ± 0.19 | 25.91 ± 0.58 |
| | LoRA | 27.20 ± 0.15 | 7.98 ± 0.08 | 18.99 ± 0.17 | 3.78 ± 0.19 | 26.66 ± 0.54 |
| 1624 | Ours | 30.29 ± 0.44 | 9.02 ± 0.12 | 20.99 ± 0.29 | 4.86 ± 0.20 | 27.63 ± 0.54 |
| | FT Projector | 27.10 ± 0.18 | 7.84 ± 0.08 | 18.85 ± 0.08 | 3.66 ± 0.05 | 27.13 ± 0.17 |
| | Projector | 29.60 ± 0.69 | 8.79 ± 0.25 | 20.73 ± 0.34 | 4.89 ± 0.25 | 25.48 ± 1.13 |
| | LoRA | 27.28 ± 0.18 | 8.20 ± 0.13 | 19.29 ± 0.15 | 3.80 ± 0.10 | 27.27 ± 0.25 |

### K.1.6 CHEBI-20 DATASET

Table 18: All results and metrics for the MolCA encoder on the ChEBI-20 dataset. We show the single seed results.

| Sample Size | Setup \ Metric | ROUGE-1 | ROUGE-2 | ROUGE-L | BLEU | METEOR |
|---|---|---|---|---|---|---|
| 32 | Ours | 32.30 | 13.67 | 26.64 | 8.93 | 20.62 |
| | FT Projector | 21.27 | 6.52 | 19.42 | 0.75 | 7.32 |
| | Projector | 20.97 | 6.68 | 15.97 | 3.59 | 13.93 |
| | LoRA | 22.26 | 7.38 | 18.30 | 3.26 | 12.81 |
| 128 | Ours | 33.35 | 14.05 | 26.99 | 9.31 | 22.20 |
| | FT Projector | 26.66 | 11.11 | 22.11 | 5.89 | 16.73 |
| | Projector | 18.93 | 6.13 | 15.09 | 4.13 | 12.81 |
| | LoRA | 30.11 | 12.29 | 24.23 | 7.36 | 19.90 |
| 512 | Ours | 35.49 | 17.61 | 30.26 | 11.91 | 23.85 |
| | FT Projector | 32.59 | 14.44 | 26.33 | 9.68 | 20.87 |
| | Projector | 24.88 | 10.17 | 20.46 | 5.84 | 14.72 |
| | LoRA | 32.43 | 14.13 | 26.37 | 9.46 | 20.67 |
| 2048 | Ours | 39.91 | 20.60 | 33.50 | 14.22 | 29.44 |
| | FT Projector | 37.88 | 18.17 | 30.65 | 13.36 | 26.47 |
| | Projector | 35.37 | 16.49 | 29.29 | 10.87 | 23.71 |
| | LoRA | 34.83 | 15.88 | 28.38 | 10.86 | 25.41 |
| 8192 | Ours | 42.67 | 23.57 | 36.69 | 17.19 | 31.01 |
| | FT Projector | 43.77 | 23.33 | 36.17 | 19.30 | 33.32 |
| | Projector | 32.21 | 14.34 | 26.25 | 8.42 | 22.73 |
| | LoRA | 40.17 | 21.43 | 34.44 | 15.30 | 29.01 |
| 226407 | Ours | 44.55 | 24.65 | 37.41 | 18.61 | 35.64 |
| | FT Projector | 47.24 | 26.93 | 39.53 | 22.57 | 37.40 |
| | Projector | 36.54 | 17.94 | 30.26 | 11.49 | 26.55 |
| | LoRA | 40.00 | 21.33 | 34.20 | 15.05 | 28.60 |

### K.1.7 SOUNDBIBLE DATASET

Table 19: All results and metrics for the BLAT encoder on the SoundBible dataset. We show the mean ± standard error calculated over three random seeds.

| Sample Size | Setup \ Metric | ROUGE-1 | ROUGE-2 | ROUGE-L | BLEU | METEOR |
|---|---|---|---|---|---|---|
| 32 | Ours | $27.20 \pm 0.44$ | $3.52 \pm 0.14$ | $25.49 \pm 0.49$ | $3.39 \pm 0.10$ | $23.03 \pm 0.16$ |
| | FT Projector | $27.42 \pm 0.10$ | $4.51 \pm 0.07$ | $25.36 \pm 0.15$ | $3.36 \pm 0.02$ | $24.15 \pm 0.11$ |
| | Projector | $24.60 \pm 1.24$ | $2.87 \pm 0.11$ | $22.95 \pm 1.14$ | $2.75 \pm 0.19$ | $21.76 \pm 0.70$ |
| | LoRA | $25.69 \pm 0.84$ | $2.50 \pm 0.27$ | $23.59 \pm 0.83$ | $2.69 \pm 0.18$ | $22.13 \pm 0.59$ |
| 128 | Ours | $30.32 \pm 0.47$ | $8.39 \pm 0.36$ | $28.96 \pm 0.44$ | $6.87 \pm 0.21$ | $26.74 \pm 0.31$ |
| | FT Projector | $31.52 \pm 0.40$ | $8.94 \pm 0.37$ | $29.82 \pm 0.34$ | $7.29 \pm 0.35$ | $27.92 \pm 0.45$ |
| | Projector | $29.74 \pm 0.71$ | $8.74 \pm 0.25$ | $28.02 \pm 0.63$ | $6.92 \pm 0.22$ | $26.88 \pm 0.43$ |
| | LoRA | $29.76 \pm 0.51$ | $8.28 \pm 0.40$ | $28.21 \pm 0.51$ | $7.05 \pm 0.17$ | $26.25 \pm 0.42$ |
| 512 | Ours | $32.26 \pm 0.65$ | $10.82 \pm 0.65$ | $31.10 \pm 0.66$ | $8.75 \pm 0.90$ | $29.65 \pm 0.65$ |
| | FT Projector | $31.75 \pm 0.65$ | $9.80 \pm 0.33$ | $30.41 \pm 0.72$ | $7.58 \pm 0.28$ | $28.54 \pm 0.34$ |
| | Projector | $31.75 \pm 0.89$ | $9.83 \pm 0.89$ | $30.30 \pm 0.84$ | $7.81 \pm 0.70$ | $28.64 \pm 0.69$ |
| | LoRA | $31.92 \pm 1.18$ | $9.22 \pm 0.53$ | $30.31 \pm 1.08$ | $6.85 \pm 0.53$ | $27.77 \pm 0.58$ |
| 862 | Ours | $32.86 \pm 0.56$ | $12.15 \pm 0.40$ | $31.52 \pm 0.61$ | $9.39 \pm 0.33$ | $30.27 \pm 0.41$ |
| | FT Projector | $34.26 \pm 0.33$ | $12.42 \pm 0.37$ | $32.74 \pm 0.35$ | $8.91 \pm 0.36$ | $31.56 \pm 0.38$ |
| | Projector | $33.34 \pm 0.47$ | $11.19 \pm 0.39$ | $31.83 \pm 0.52$ | $8.12 \pm 0.21$ | $29.93 \pm 0.57$ |
| | LoRA | $33.53 \pm 0.49$ | $10.14 \pm 0.41$ | $32.04 \pm 0.48$ | $7.59 \pm 0.38$ | $28.98 \pm 0.40$ |

## K.2 LLAMA 3.2 1B INSTRUCT RESULTS

### K.2.1 SYDNEYCAPTIONS DATASET

Table 20: All results and metrics for the ViT-B-32 encoder on the SydneyCaptions dataset. We show the mean ± standard error calculated over five random seeds.

| Sample Size | Setup \ Metric | ROUGE-1 | ROUGE-2 | ROUGE-L | BLEU | METEOR | CIDEr |
|---|---|---|---|---|---|---|---|
| 32 | Ours | 59.40 ± 2.15 | 43.82 ± 2.16 | 57.06 ± 2.16 | 41.34 ± 2.23 | 55.44 ± 2.46 | 166.58 ± 7.35 |
| | FT Projector | 48.91 ± 0.17 | 30.12 ± 0.55 | 45.48 ± 0.31 | 28.16 ± 0.60 | 42.32 ± 0.38 | 121.49 ± 4.14 |
| | Projector | 47.67 ± 1.02 | 30.54 ± 1.41 | 44.81 ± 1.06 | 29.82 ± 2.78 | 42.32 ± 0.98 | 128.17 ± 5.10 |
| | LoRA | 47.91 ± 0.51 | 29.65 ± 0.96 | 44.86 ± 0.63 | 26.65 ± 1.04 | 41.92 ± 0.64 | 109.91 ± 7.97 |
| 128 | Ours | 59.97 ± 2.07 | 43.80 ± 2.54 | 57.14 ± 1.96 | 37.92 ± 3.01 | 58.65 ± 1.92 | 148.99 ± 17.39 |
| | FT Projector | 60.15 ± 1.30 | 42.36 ± 1.65 | 56.87 ± 1.46 | 39.44 ± 1.71 | 55.65 ± 1.60 | 157.42 ± 7.03 |
| | Projector | 58.21 ± 1.20 | 42.07 ± 1.52 | 55.15 ± 1.43 | 38.47 ± 0.97 | 54.92 ± 1.62 | 151.02 ± 2.11 |
| | LoRA | 51.75 ± 1.52 | 33.23 ± 2.29 | 48.72 ± 1.82 | 29.01 ± 2.41 | 46.24 ± 1.78 | 84.32 ± 7.30 |
| 512 | Ours | 62.12 ± 1.38 | 45.86 ± 1.80 | 58.56 ± 1.70 | 41.72 ± 2.97 | 60.19 ± 1.41 | 164.72 ± 8.87 |
| | FT Projector | 52.67 ± 0.63 | 33.67 ± 0.69 | 48.82 ± 0.69 | 28.02 ± 1.17 | 47.46 ± 0.67 | 132.58 ± 4.79 |
| | Projector | 58.48 ± 1.10 | 41.52 ± 1.29 | 54.80 ± 1.33 | 37.70 ± 1.38 | 56.06 ± 1.70 | 149.77 ± 6.21 |
| | LoRA | 54.06 ± 2.14 | 35.10 ± 2.60 | 50.16 ± 2.55 | 30.56 ± 3.45 | 48.88 ± 1.65 | 131.85 ± 8.41 |
| 2048 | Ours | 65.61 ± 0.64 | 49.16 ± 1.04 | 62.59 ± 0.65 | 44.86 ± 1.03 | 64.06 ± 0.87 | 196.81 ± 4.96 |
| | FT Projector | 58.54 ± 0.68 | 40.19 ± 1.17 | 55.36 ± 0.73 | 37.09 ± 1.36 | 53.41 ± 0.88 | 136.64 ± 11.00 |
| | Projector | 64.30 ± 1.09 | 47.75 ± 1.14 | 61.05 ± 1.17 | 44.18 ± 1.40 | 62.18 ± 1.04 | 180.29 ± 5.82 |
| | LoRA | 61.70 ± 0.63 | 44.68 ± 1.26 | 58.99 ± 0.65 | 42.61 ± 2.18 | 56.26 ± 1.32 | 136.58 ± 15.71 |
| 2485 | Ours | 67.10 ± 0.73 | 51.08 ± 0.76 | 64.07 ± 0.61 | 48.24 ± 1.03 | 66.14 ± 0.55 | 202.00 ± 10.71 |
| | FT Projector | 60.98 ± 1.09 | 43.17 ± 1.46 | 57.30 ± 1.25 | 41.25 ± 1.70 | 56.66 ± 1.42 | 162.94 ± 14.50 |
| | Projector | 65.01 ± 1.22 | 49.05 ± 1.41 | 61.72 ± 1.26 | 44.45 ± 1.43 | 62.87 ± 1.39 | 190.30 ± 8.01 |
| | LoRA | 59.32 ± 1.41 | 40.44 ± 1.79 | 56.10 ± 1.50 | 38.83 ± 1.61 | 54.56 ± 1.61 | 128.98 ± 10.14 |

Table 21: All results and metrics for the ViT-L-14 encoder on the SydneyCaptions dataset. We show the mean ± standard error calculated over five random seeds.

| Sample Size | Setup \ Metric | ROUGE-1 | ROUGE-2 | ROUGE-L | BLEU | METEOR | CIDEr |
|---|---|---|---|---|---|---|---|
| 32 | Ours | 57.95 ± 0.83 | 41.15 ± 1.11 | 54.02 ± 0.80 | 36.37 ± 0.75 | 55.06 ± 1.02 | 133.65 ± 2.69 |
| | FT Projector | 51.19 ± 1.43 | 35.00 ± 1.49 | 48.36 ± 1.54 | 31.06 ± 1.73 | 45.31 ± 1.42 | 80.49 ± 5.57 |
| | Projector | 54.31 ± 1.76 | 38.65 ± 2.21 | 51.06 ± 1.54 | 35.90 ± 2.69 | 50.71 ± 2.22 | 113.79 ± 6.87 |
| | LoRA | 47.93 ± 1.52 | 31.50 ± 1.93 | 45.40 ± 1.65 | 28.02 ± 2.96 | 42.03 ± 1.23 | 69.21 ± 7.36 |
| 128 | Ours | 62.65 ± 0.64 | 47.53 ± 1.16 | 59.69 ± 0.83 | 45.17 ± 1.84 | 58.92 ± 0.62 | 173.37 ± 4.04 |
| | FT Projector | 55.87 ± 0.71 | 39.14 ± 0.94 | 53.30 ± 0.80 | 31.92 ± 1.16 | 52.09 ± 0.58 | 122.39 ± 2.46 |
| | Projector | 53.68 ± 2.45 | 38.27 ± 2.53 | 51.07 ± 2.56 | 34.69 ± 3.45 | 50.15 ± 2.31 | 133.22 ± 9.80 |
| | LoRA | 47.27 ± 1.10 | 27.14 ± 1.97 | 44.27 ± 1.14 | 21.17 ± 2.13 | 42.66 ± 1.52 | 61.08 ± 9.59 |
| 512 | Ours | 63.47 ± 0.55 | 46.55 ± 0.50 | 60.77 ± 0.62 | 41.75 ± 1.62 | 61.09 ± 0.26 | 167.89 ± 6.54 |
| | FT Projector | 52.92 ± 1.26 | 33.60 ± 1.39 | 49.04 ± 1.38 | 31.06 ± 1.14 | 46.91 ± 1.49 | 122.91 ± 3.31 |
| | Projector | 56.79 ± 5.23 | 39.55 ± 5.85 | 53.88 ± 5.29 | 37.23 ± 6.61 | 52.79 ± 5.27 | 128.26 ± 27.20 |
| | LoRA | 53.40 ± 1.62 | 33.39 ± 1.57 | 49.65 ± 1.57 | 29.83 ± 1.47 | 47.96 ± 2.12 | 110.08 ± 11.03 |
| 2048 | Ours | 66.07 ± 1.13 | 50.49 ± 1.92 | 62.68 ± 1.17 | 47.60 ± 2.12 | 63.80 ± 1.51 | 201.02 ± 9.60 |
| | FT Projector | 57.15 ± 0.97 | 38.63 ± 1.19 | 53.15 ± 0.86 | 35.84 ± 1.35 | 53.20 ± 1.13 | 142.95 ± 11.20 |
| | Projector | 61.93 ± 3.32 | 46.15 ± 3.00 | 59.21 ± 3.25 | 43.08 ± 2.33 | 59.54 ± 2.71 | 163.92 ± 23.81 |
| | LoRA | 58.77 ± 1.92 | 39.19 ± 2.48 | 54.79 ± 2.27 | 33.86 ± 2.26 | 54.50 ± 1.92 | 142.26 ± 6.24 |
| 2485 | Ours | 65.58 ± 1.45 | 49.75 ± 1.67 | 62.83 ± 1.57 | 47.48 ± 1.13 | 62.23 ± 1.88 | 194.78 ± 6.42 |
| | FT Projector | 59.23 ± 1.05 | 40.40 ± 0.95 | 54.94 ± 1.09 | 39.03 ± 1.25 | 54.43 ± 1.06 | 158.46 ± 5.77 |
| | Projector | 63.29 ± 1.98 | 46.60 ± 2.71 | 60.21 ± 2.23 | 44.22 ± 2.65 | 61.00 ± 1.83 | 174.95 ± 14.72 |
| | LoRA | 59.17 ± 1.32 | 40.40 ± 1.32 | 55.37 ± 1.58 | 37.16 ± 1.80 | 55.08 ± 1.43 | 146.89 ± 4.94 |

Table 22: All results and metrics for the RN-50 encoder on the SydneyCaptions dataset. We show the mean ± standard error calculated over five random seeds.

| Sample Size | Setup \ Metric | ROUGE-1 | ROUGE-2 | ROUGE-L | BLEU | METEOR | CIDEr |
|---|---|---|---|---|---|---|---|
| 32 | Ours | 58.05 ± 0.33 | 42.76 ± 1.03 | 55.17 ± 0.68 | 42.34 ± 1.37 | 53.45 ± 0.52 | 153.84 ± 5.53 |
| | FT Projector | 56.75 ± 0.85 | 39.59 ± 0.85 | 53.35 ± 0.70 | 38.67 ± 1.36 | 51.79 ± 0.95 | 130.78 ± 4.67 |
| | Projector | 51.09 ± 3.35 | 33.71 ± 4.34 | 47.76 ± 3.53 | 29.97 ± 5.36 | 46.67 ± 3.58 | 107.43 ± 22.35 |
| | LoRA | 47.91 ± 0.52 | 26.35 ± 1.43 | 44.63 ± 0.84 | 23.36 ± 1.86 | 41.03 ± 0.86 | 61.69 ± 1.26 |
| 128 | Ours | 57.78 ± 1.70 | 39.78 ± 2.26 | 54.37 ± 2.24 | 35.94 ± 3.07 | 55.09 ± 2.01 | 128.53 ± 10.50 |
| | FT Projector | 49.10 ± 0.60 | 29.71 ± 0.96 | 45.28 ± 0.41 | 28.60 ± 1.50 | 44.30 ± 0.56 | 67.96 ± 4.25 |
| | Projector | 50.07 ± 2.25 | 30.65 ± 2.90 | 46.19 ± 2.37 | 26.83 ± 2.98 | 45.41 ± 2.46 | 85.86 ± 14.36 |
| | LoRA | 41.80 ± 1.19 | 20.80 ± 0.76 | 37.82 ± 0.97 | 15.25 ± 0.53 | 37.76 ± 1.70 | 27.55 ± 2.73 |
| 512 | Ours | 63.00 ± 0.98 | 47.31 ± 0.55 | 60.13 ± 0.86 | 46.69 ± 1.30 | 60.46 ± 0.57 | 169.70 ± 5.57 |
| | FT Projector | 57.08 ± 0.45 | 37.16 ± 1.08 | 53.30 ± 0.77 | 34.82 ± 1.36 | 53.58 ± 0.72 | 121.55 ± 5.43 |
| | Projector | 58.09 ± 2.56 | 40.61 ± 3.11 | 54.85 ± 2.42 | 36.73 ± 3.55 | 54.55 ± 3.38 | 129.43 ± 21.51 |
| | LoRA | 55.80 ± 1.97 | 36.30 ± 2.93 | 51.83 ± 2.63 | 30.37 ± 3.56 | 52.74 ± 2.05 | 101.91 ± 11.97 |
| 2048 | Ours | 65.61 ± 0.69 | 49.83 ± 1.04 | 62.95 ± 0.72 | 47.83 ± 1.20 | 63.64 ± 0.92 | 202.74 ± 7.86 |
| | FT Projector | 62.28 ± 0.83 | 45.31 ± 1.06 | 59.28 ± 0.80 | 43.24 ± 1.19 | 58.33 ± 1.17 | 176.46 ± 7.88 |
| | Projector | 65.69 ± 1.16 | 50.08 ± 1.93 | 62.74 ± 1.47 | 47.98 ± 1.77 | 63.23 ± 1.36 | 191.63 ± 8.39 |
| | LoRA | 60.87 ± 1.85 | 43.42 ± 2.51 | 57.50 ± 2.26 | 39.00 ± 3.07 | 59.01 ± 1.78 | 148.69 ± 14.45 |
| 2485 | Ours | 66.44 ± 0.43 | 51.42 ± 0.40 | 63.88 ± 0.56 | 51.17 ± 0.58 | 62.90 ± 0.69 | 198.29 ± 2.26 |
| | FT Projector | 57.50 ± 0.88 | 38.17 ± 1.11 | 53.17 ± 0.93 | 35.97 ± 1.31 | 52.66 ± 1.07 | 151.78 ± 6.43 |
| | Projector | 54.30 ± 6.57 | 37.20 ± 7.50 | 51.03 ± 6.94 | 34.66 ± 7.91 | 50.82 ± 7.28 | 129.60 ± 37.51 |
| | LoRA | 57.66 ± 1.59 | 38.37 ± 1.55 | 53.53 ± 1.53 | 35.37 ± 1.60 | 53.88 ± 1.38 | 129.75 ± 10.96 |

## K.2.2 CAPDELS DATASET

Table 23: All results and metrics for the ConvNeXt-Nano encoder on the CAPDELS dataset. We show the mean ± standard error calculated over five random seeds.

| Sample Size | Setup \ Metric | ROUGE-1 | ROUGE-2 | ROUGE-L | BLEU | METEOR | CIDEr |
|---|---|---|---|---|---|---|---|
| 32 | Ours | 52.08 ± 1.22 | 35.44 ± 1.20 | 45.60 ± 1.12 | 30.29 ± 0.91 | 46.07 ± 1.37 | 135.57 ± 9.21 |
| | FT Projector | 43.76 ± 0.59 | 24.47 ± 0.62 | 36.21 ± 0.62 | 19.16 ± 0.48 | 39.47 ± 0.81 | 57.40 ± 1.52 |
| | Projector | 37.99 ± 2.35 | 20.59 ± 1.91 | 31.68 ± 2.09 | 15.49 ± 1.89 | 32.46 ± 1.74 | 34.80 ± 4.73 |
| | LoRA | 36.61 ± 1.51 | 18.49 ± 1.76 | 29.70 ± 1.66 | 14.62 ± 2.00 | 33.68 ± 1.16 | 27.40 ± 4.31 |
| 128 | Ours | 60.34 ± 0.58 | 43.06 ± 0.64 | 53.13 ± 0.58 | 37.89 ± 0.89 | 55.70 ± 0.53 | 189.89 ± 7.42 |
| | FT Projector | 53.89 ± 0.60 | 35.25 ± 0.24 | 45.96 ± 0.31 | 28.14 ± 0.25 | 49.38 ± 0.71 | 135.04 ± 2.15 |
| | Projector | 51.95 ± 1.63 | 33.71 ± 1.45 | 44.33 ± 1.63 | 28.13 ± 1.82 | 47.88 ± 1.46 | 124.49 ± 10.98 |
| | LoRA | 51.50 ± 0.80 | 32.60 ± 1.09 | 43.57 ± 0.96 | 26.77 ± 1.11 | 46.78 ± 0.97 | 107.50 ± 10.36 |
| 512 | Ours | 61.17 ± 4.01 | 44.45 ± 4.23 | 53.81 ± 3.96 | 39.38 ± 3.79 | 56.77 ± 3.81 | 208.18 ± 23.45 |
| | FT Projector | 65.69 ± 1.22 | 48.37 ± 1.52 | 57.58 ± 1.25 | 42.15 ± 1.47 | 61.66 ± 1.17 | 227.20 ± 14.22 |
| | Projector | 63.80 ± 0.69 | 47.03 ± 1.00 | 56.09 ± 0.94 | 41.00 ± 1.75 | 59.26 ± 0.99 | 199.44 ± 16.04 |
| | LoRA | 64.61 ± 1.52 | 47.04 ± 1.78 | 56.36 ± 1.58 | 41.93 ± 1.67 | 60.50 ± 1.38 | 206.65 ± 16.98 |
| 2048 | Ours | 70.10 ± 1.04 | 54.97 ± 1.21 | 63.27 ± 1.07 | 50.64 ± 1.19 | 65.90 ± 1.31 | 262.44 ± 15.99 |
| | FT Projector | 71.74 ± 0.56 | 56.08 ± 0.85 | 64.01 ± 0.70 | 50.34 ± 0.79 | 68.40 ± 0.81 | 275.99 ± 8.42 |
| | Projector | 65.43 ± 5.07 | 48.54 ± 6.59 | 57.29 ± 6.02 | 44.06 ± 6.20 | 62.36 ± 5.04 | 219.56 ± 46.77 |
| | LoRA | 71.46 ± 0.78 | 55.79 ± 1.08 | 63.92 ± 0.95 | 50.99 ± 1.03 | 68.00 ± 0.83 | 265.94 ± 11.19 |
| 4344 | Ours | 71.83 ± 0.82 | 56.83 ± 1.07 | 64.86 ± 0.98 | 53.17 ± 1.74 | 68.57 ± 0.79 | 283.46 ± 11.44 |
| | FT Projector | 70.97 ± 1.01 | 55.83 ± 0.94 | 63.40 ± 0.98 | 52.29 ± 0.87 | 67.33 ± 1.21 | 274.70 ± 5.35 |
| | Projector | 68.34 ± 2.65 | 51.87 ± 3.45 | 60.46 ± 3.00 | 47.71 ± 3.44 | 64.49 ± 3.39 | 243.58 ± 27.75 |
| | LoRA | 71.77 ± 0.47 | 56.55 ± 0.78 | 64.39 ± 0.58 | 52.27 ± 0.89 | 68.22 ± 0.51 | 266.47 ± 7.59 |

Table 24: All results and metrics for the ConvNeXt-Tiny encoder on the CAPDELS dataset. We show the mean ± standard error calculated over five random seeds.

| Sample Size | Setup \ Metric | ROUGE-1 | ROUGE-2 | ROUGE-L | BLEU | METEOR | CIDEr |
|---|---|---|---|---|---|---|---|
| 32 | Ours | 54.04 ± 1.08 | 36.09 ± 1.12 | 46.90 ± 1.02 | 31.13 ± 1.48 | 48.41 ± 1.44 | 136.77 ± 13.36 |
| | FT Projector | 45.23 ± 0.73 | 26.56 ± 0.92 | 37.83 ± 0.77 | 21.86 ± 0.81 | 39.61 ± 0.53 | 70.52 ± 4.90 |
| | Projector | 35.84 ± 3.52 | 17.22 ± 3.40 | 28.97 ± 3.25 | 13.61 ± 2.76 | 30.71 ± 3.79 | 34.94 ± 11.34 |
| | LoRA | 41.07 ± 1.18 | 21.35 ± 1.50 | 33.26 ± 1.31 | 17.53 ± 1.42 | 35.01 ± 1.22 | 33.87 ± 6.49 |
| 128 | Ours | 61.18 ± 1.06 | 44.13 ± 1.44 | 53.93 ± 1.22 | 39.95 ± 1.52 | 56.25 ± 1.21 | 178.43 ± 14.90 |
| | FT Projector | 52.64 ± 0.51 | 35.05 ± 0.71 | 45.76 ± 0.63 | 30.25 ± 0.86 | 48.16 ± 0.45 | 118.66 ± 3.97 |
| | Projector | 45.11 ± 5.37 | 26.98 ± 5.17 | 37.93 ± 5.13 | 21.92 ± 4.41 | 40.51 ± 5.37 | 77.51 ± 20.00 |
| | LoRA | 50.92 ± 1.39 | 32.95 ± 1.99 | 43.45 ± 1.64 | 27.89 ± 1.68 | 46.64 ± 1.54 | 114.08 ± 15.51 |
| 512 | Ours | 64.90 ± 3.74 | 48.39 ± 4.66 | 57.59 ± 4.17 | 44.17 ± 4.23 | 60.17 ± 4.15 | 223.53 ± 31.31 |
| | FT Projector | 63.73 ± 1.03 | 46.64 ± 1.04 | 55.82 ± 0.99 | 39.75 ± 1.09 | 60.21 ± 0.88 | 200.12 ± 8.34 |
| | Projector | 56.42 ± 6.51 | 39.92 ± 7.02 | 49.21 ± 6.54 | 33.72 ± 6.09 | 53.24 ± 7.10 | 167.93 ± 42.92 |
| | LoRA | 59.33 ± 1.95 | 40.94 ± 2.28 | 50.69 ± 2.06 | 35.68 ± 2.05 | 56.07 ± 1.83 | 148.75 ± 15.73 |
| 2048 | Ours | 71.07 ± 0.65 | 55.33 ± 1.05 | 63.70 ± 0.81 | 51.25 ± 1.43 | 68.08 ± 0.59 | 273.29 ± 9.07 |
| | FT Projector | 70.51 ± 0.79 | 54.84 ± 1.13 | 62.90 ± 0.93 | 49.58 ± 1.16 | 67.32 ± 0.80 | 257.75 ± 12.89 |
| | Projector | 60.90 ± 7.06 | 44.06 ± 8.19 | 53.32 ± 7.39 | 39.61 ± 8.08 | 56.53 ± 7.57 | 194.18 ± 53.85 |
| | LoRA | 66.10 ± 1.26 | 49.17 ± 1.35 | 58.15 ± 1.18 | 43.69 ± 1.71 | 63.42 ± 1.26 | 205.22 ± 15.54 |
| 4344 | Ours | 70.60 ± 1.92 | 54.53 ± 2.72 | 63.23 ± 2.20 | 50.99 ± 2.45 | 66.78 ± 2.62 | 254.96 ± 30.30 |
| | FT Projector | 72.00 ± 0.25 | 56.75 ± 0.47 | 64.47 ± 0.41 | 51.32 ± 0.82 | 69.15 ± 0.24 | 287.33 ± 5.02 |
| | Projector | 65.09 ± 5.95 | 48.19 ± 7.22 | 57.07 ± 6.43 | 44.38 ± 7.18 | 62.15 ± 5.85 | 212.26 ± 50.46 |
| | LoRA | 69.25 ± 0.92 | 53.19 ± 1.05 | 61.72 ± 1.00 | 48.17 ± 1.48 | 66.10 ± 0.97 | 250.90 ± 11.67 |

Table 25: All results and metrics for the ConvNeXt-Base encoder on the CAPDELS dataset. We show the mean ± standard error calculated over five random seeds.

| Sample Size | Setup \ Metric | ROUGE-1 | ROUGE-2 | ROUGE-L | BLEU | METEOR | CIDEr |
|---|---|---|---|---|---|---|---|
| 32 | Ours | 50.67 ± 3.26 | 31.75 ± 4.45 | 43.22 ± 3.81 | 26.77 ± 3.84 | 44.57 ± 3.29 | 91.85 ± 27.84 |
| | FT Projector | 45.45 ± 1.10 | 26.57 ± 1.38 | 38.17 ± 1.28 | 20.40 ± 1.39 | 40.94 ± 1.17 | 62.25 ± 5.29 |
| | Projector | 37.00 ± 2.85 | 18.31 ± 3.01 | 30.20 ± 2.63 | 14.27 ± 2.59 | 31.47 ± 2.94 | 37.62 ± 10.70 |
| | LoRA | 37.93 ± 0.80 | 18.07 ± 1.05 | 30.32 ± 0.95 | 14.13 ± 1.40 | 32.70 ± 0.69 | 18.05 ± 1.64 |
| 128 | Ours | 56.07 ± 3.56 | 39.14 ± 3.70 | 48.72 ± 3.95 | 35.83 ± 3.63 | 51.94 ± 3.35 | 151.69 ± 29.21 |
| | FT Projector | 54.70 ± 0.64 | 36.64 ± 0.87 | 47.37 ± 0.74 | 30.75 ± 0.77 | 50.36 ± 0.66 | 132.67 ± 7.65 |
| | Projector | 44.11 ± 3.46 | 25.02 ± 3.34 | 36.58 ± 3.20 | 20.51 ± 3.28 | 39.40 ± 3.37 | 67.66 ± 13.22 |
| | LoRA | 52.57 ± 1.37 | 34.97 ± 1.56 | 45.32 ± 1.38 | 30.03 ± 1.25 | 47.41 ± 1.94 | 115.87 ± 28.32 |
| 512 | Ours | 66.35 ± 1.52 | 49.89 ± 1.96 | 58.87 ± 1.59 | 43.91 ± 1.51 | 62.83 ± 1.76 | 227.16 ± 19.53 |
| | FT Projector | 61.05 ± 1.15 | 43.22 ± 1.16 | 52.22 ± 1.06 | 36.42 ± 1.09 | 56.47 ± 1.22 | 180.08 ± 10.97 |
| | Projector | 55.15 ± 5.84 | 37.00 ± 6.47 | 47.13 ± 5.87 | 31.77 ± 6.10 | 50.68 ± 6.40 | 139.86 ± 43.62 |
| | LoRA | 57.83 ± 1.18 | 38.10 ± 1.30 | 48.39 ± 1.16 | 34.44 ± 1.83 | 53.48 ± 1.26 | 127.43 ± 9.25 |
| 2048 | Ours | 70.67 ± 0.52 | 55.12 ± 0.49 | 63.77 ± 0.54 | 51.94 ± 0.78 | 67.05 ± 0.35 | 255.76 ± 7.41 |
| | FT Projector | 70.91 ± 0.51 | 55.32 ± 0.72 | 63.38 ± 0.64 | 50.18 ± 0.74 | 66.98 ± 0.54 | 271.17 ± 4.96 |
| | Projector | 67.19 ± 2.09 | 50.46 ± 2.88 | 59.20 ± 2.59 | 46.65 ± 3.05 | 63.47 ± 2.01 | 226.06 ± 25.47 |
| | LoRA | 64.89 ± 2.27 | 46.82 ± 3.22 | 55.74 ± 2.85 | 42.61 ± 3.26 | 60.25 ± 1.98 | 183.69 ± 25.57 |
| 4344 | Ours | 72.13 ± 1.36 | 56.60 ± 2.01 | 64.86 ± 1.70 | 52.44 ± 2.06 | 68.98 ± 1.37 | 270.30 ± 27.86 |
| | FT Projector | 70.28 ± 1.02 | 55.01 ± 1.24 | 62.96 ± 1.12 | 50.25 ± 0.94 | 66.37 ± 1.41 | 271.27 ± 8.59 |
| | Projector | 65.18 ± 5.60 | 48.71 ± 6.56 | 57.44 ± 5.88 | 43.98 ± 7.20 | 61.49 ± 5.78 | 223.51 ± 44.92 |
| | LoRA | 72.08 ± 0.64 | 56.14 ± 0.86 | 64.26 ± 0.76 | 50.51 ± 0.88 | 69.42 ± 0.65 | 265.31 ± 10.37 |

### K.2.3 SENSORCAPS DATASET

Table 26: All results and metrics for the LIMU-BERT encoder on the SensorCaps dataset. We show the mean ± standard error calculated over five random seeds.

| Sample Size | Setup \ Metric | ROUGE-1 | ROUGE-2 | ROUGE-L | BLEU | METEOR |
|---|---|---|---|---|---|---|
| 32 | Ours | 27.87 ± 0.69 | 6.47 ± 0.37 | 17.09 ± 0.47 | 4.00 ± 0.28 | 20.11 ± 0.24 |
| | FT Projector | 28.14 ± 0.36 | 5.86 ± 0.21 | 16.90 ± 0.22 | 3.48 ± 0.09 | 20.06 ± 0.33 |
| | Projector | 22.06 ± 3.67 | 4.59 ± 1.20 | 14.28 ± 2.27 | 3.16 ± 0.91 | 16.52 ± 2.71 |
| | LoRA | 30.26 ± 0.40 | 8.19 ± 0.31 | 17.88 ± 0.28 | 4.55 ± 0.11 | 21.53 ± 0.34 |
| 128 | Ours | 39.77 ± 1.11 | 12.93 ± 0.57 | 23.65 ± 0.63 | 9.02 ± 0.28 | 29.61 ± 0.62 |
| | FT Projector | 36.10 ± 0.46 | 10.76 ± 0.29 | 21.69 ± 0.25 | 7.89 ± 0.29 | 26.79 ± 0.42 |
| | Projector | 28.17 ± 4.72 | 7.53 ± 1.85 | 18.09 ± 2.91 | 5.21 ± 1.59 | 20.64 ± 3.92 |
| | LoRA | 37.41 ± 0.60 | 11.96 ± 0.27 | 22.43 ± 0.40 | 7.71 ± 0.32 | 26.61 ± 0.32 |
| 512 | Ours | 41.12 ± 1.38 | 14.36 ± 0.95 | 24.88 ± 1.26 | 9.52 ± 1.42 | 30.18 ± 1.99 |
| | FT Projector | 38.32 ± 0.63 | 12.69 ± 0.29 | 22.79 ± 0.53 | 8.18 ± 0.27 | 27.61 ± 0.44 |
| | Projector | 32.40 ± 5.01 | 9.34 ± 2.03 | 19.50 ± 2.79 | 6.46 ± 1.42 | 23.96 ± 3.68 |
| | LoRA | 36.92 ± 1.68 | 12.67 ± 0.32 | 22.71 ± 0.73 | 8.67 ± 0.36 | 27.40 ± 1.04 |
| 1670 | Ours | 41.66 ± 1.07 | 15.11 ± 0.57 | 25.72 ± 0.93 | 9.74 ± 0.96 | 30.15 ± 1.36 |
| | FT Projector | 41.37 ± 0.32 | 14.00 ± 0.24 | 24.79 ± 0.31 | 9.49 ± 0.34 | 29.99 ± 0.43 |
| | Projector | 29.84 ± 5.52 | 8.51 ± 2.19 | 17.84 ± 3.03 | 5.39 ± 1.48 | 21.83 ± 4.06 |
| | LoRA | 40.91 ± 1.14 | 14.23 ± 0.75 | 24.56 ± 0.94 | 9.36 ± 0.88 | 29.51 ± 1.27 |

### K.2.4 CHEBI-20 DATASET

Table 27: All results and metrics for the MolCA encoder on the ChEBI-20 dataset. We show the mean ± standard error calculated over five random seeds.

| Sample Size | Setup \ Metric | ROUGE-1 | ROUGE-2 | ROUGE-L | BLEU | METEOR |
|---|---|---|---|---|---|---|
| 32 | Ours | 27.63 ± 0.97 | 10.80 ± 0.56 | 22.20 ± 0.80 | 6.26 ± 0.49 | 17.91 ± 0.57 |
| | FT Projector | 18.56 ± 0.17 | 4.65 ± 0.08 | 14.72 ± 0.13 | 2.38 ± 0.05 | 9.89 ± 0.09 |
| | Projector | 21.99 ± 1.06 | 8.40 ± 0.67 | 18.07 ± 1.01 | 4.19 ± 0.37 | 12.81 ± 0.62 |
| | LoRA | 23.22 ± 0.40 | 6.82 ± 0.27 | 18.70 ± 0.31 | 3.83 ± 0.24 | 12.33 ± 0.44 |
| 128 | Ours | 29.60 ± 1.07 | 11.71 ± 1.26 | 23.65 ± 1.26 | 7.45 ± 0.85 | 17.86 ± 0.42 |
| | FT Projector | 24.97 ± 0.17 | 7.90 ± 0.12 | 19.88 ± 0.15 | 4.70 ± 0.11 | 14.52 ± 0.15 |
| | Projector | 25.27 ± 0.93 | 9.06 ± 1.01 | 20.47 ± 1.15 | 4.86 ± 0.52 | 14.68 ± 0.61 |
| | LoRA | 27.81 ± 0.20 | 9.78 ± 0.13 | 22.66 ± 0.12 | 5.66 ± 0.05 | 16.03 ± 0.43 |
| 512 | Ours | 33.46 ± 0.34 | 15.86 ± 0.42 | 27.79 ± 0.37 | 10.34 ± 0.35 | 21.04 ± 0.20 |
| | FT Projector | 28.29 ± 0.23 | 10.12 ± 0.19 | 22.53 ± 0.21 | 6.61 ± 0.12 | 16.60 ± 0.14 |
| | Projector | 27.74 ± 0.41 | 11.31 ± 0.31 | 23.18 ± 0.30 | 6.59 ± 0.31 | 16.46 ± 0.56 |
| | LoRA | 30.31 ± 1.43 | 13.79 ± 1.25 | 25.30 ± 1.48 | 8.81 ± 0.47 | 18.04 ± 0.53 |
| 2048 | Ours | 34.25 ± 0.72 | 16.28 ± 0.73 | 28.30 ± 0.85 | 11.02 ± 0.39 | 22.25 ± 0.51 |
| | FT Projector | 32.04 ± 0.20 | 13.61 ± 0.16 | 25.62 ± 0.28 | 9.34 ± 0.12 | 19.86 ± 0.17 |
| | Projector | 28.97 ± 0.94 | 11.92 ± 1.02 | 23.48 ± 1.15 | 7.37 ± 0.69 | 18.39 ± 0.88 |
| | LoRA | 31.67 ± 1.19 | 15.05 ± 0.66 | 26.35 ± 0.88 | 10.07 ± 0.79 | 20.81 ± 1.09 |
| 8192 | Ours | 33.95 ± 1.36 | 15.44 ± 1.21 | 27.57 ± 1.31 | 11.94 ± 0.58 | 23.16 ± 0.85 |
| | FT Projector | 35.56 ± 0.17 | 17.19 ± 0.15 | 28.96 ± 0.17 | 12.22 ± 0.25 | 25.13 ± 0.13 |
| | Projector | 32.16 ± 0.79 | 14.74 ± 0.65 | 26.60 ± 0.75 | 9.51 ± 0.31 | 20.95 ± 1.14 |
| | LoRA | 32.34 ± 0.63 | 15.38 ± 0.59 | 26.95 ± 0.66 | 10.33 ± 0.35 | 20.75 ± 0.68 |
| 26407 | Ours | 35.51 ± 0.73 | 16.51 ± 0.49 | 28.57 ± 0.70 | 11.22 ± 0.57 | 25.47 ± 0.48 |
| | FT Projector | 37.49 ± 0.17 | 19.29 ± 0.25 | 30.97 ± 0.21 | 14.24 ± 0.17 | 25.82 ± 0.20 |
| | Projector | 30.78 ± 1.74 | 13.55 ± 1.49 | 25.19 ± 1.61 | 8.61 ± 1.32 | 18.97 ± 1.84 |
| | LoRA | 34.04 ± 0.80 | 17.35 ± 0.71 | 29.01 ± 0.73 | 10.84 ± 0.53 | 23.14 ± 0.74 |

### K.2.5 SoundBible dataset

Table 28: All results and metrics for the BLAT encoder on the SoundBible dataset. We show the mean ± standard error calculated over five random seeds.

| Sample Size | Setup \ Metric | ROUGE-1 | ROUGE-2 | ROUGE-L | BLEU | METEOR |
|---|---|---|---|---|---|---|
| 32 | Ours | 26.83 ± 0.71 | 3.07 ± 0.22 | 25.28 ± 0.55 | 3.07 ± 0.18 | 22.55 ± 0.56 |
| | FT Projector | 24.63 ± 0.35 | 3.82 ± 0.60 | 23.52 ± 0.31 | 3.33 ± 0.12 | 21.97 ± 0.41 |
| | Projector | 20.50 ± 2.58 | 1.74 ± 0.48 | 18.94 ± 2.71 | 1.85 ± 0.56 | 18.30 ± 1.41 |
| | LoRA | 20.67 ± 0.93 | 2.03 ± 0.33 | 19.10 ± 0.92 | 1.44 ± 0.60 | 18.95 ± 0.57 |
| 128 | Ours | 24.66 ± 0.72 | 5.29 ± 0.42 | 23.33 ± 0.73 | 4.10 ± 0.17 | 22.08 ± 0.58 |
| | FT Projector | 24.51 ± 0.69 | 6.50 ± 0.33 | 22.83 ± 0.61 | 4.26 ± 0.28 | 23.43 ± 0.39 |
| | Projector | 19.00 ± 1.59 | 2.75 ± 0.46 | 17.48 ± 1.65 | 1.58 ± 0.29 | 18.12 ± 0.90 |
| | LoRA | 20.60 ± 0.85 | 3.56 ± 0.27 | 19.01 ± 0.85 | 1.82 ± 0.07 | 20.02 ± 0.40 |
| 512 | Ours | 29.70 ± 0.60 | 7.26 ± 0.67 | 27.88 ± 0.74 | 5.34 ± 0.45 | 26.43 ± 0.68 |
| | FT Projector | 23.88 ± 0.34 | 5.42 ± 0.32 | 21.58 ± 0.24 | 3.04 ± 0.38 | 23.03 ± 0.37 |
| | Projector | 26.26 ± 4.37 | 6.58 ± 2.10 | 21.22 ± 0.85 | 3.78 ± 1.42 | 22.83 ± 1.60 |
| | LoRA | 24.73 ± 1.08 | 3.95 ± 0.57 | 22.61 ± 1.09 | 2.62 ± 0.46 | 22.54 ± 0.59 |
| 862 | Ours | 33.26 ± 0.67 | 9.07 ± 0.91 | 31.85 ± 0.69 | 7.39 ± 0.59 | 28.58 ± 0.86 |
| | FT Projector | 25.78 ± 0.41 | 6.85 ± 0.44 | 23.99 ± 0.33 | 4.40 ± 0.29 | 24.60 ± 0.59 |
| | Projector | 26.33 ± 1.27 | 6.54 ± 0.83 | 24.52 ± 1.14 | 3.55 ± 0.71 | 24.65 ± 1.28 |
| | LoRA | 29.44 ± 0.24 | 6.45 ± 0.28 | 27.59 ± 0.36 | 4.92 ± 0.14 | 25.42 ± 0.28 |

## L    Dataset Details

The licences for datasets are provided in Table 29.

Table 29: Dataset licences.

| Dataset | Licence |
|---|---|
| COCO (Lin et al., 2014) | Captions: CC BY 4.0
Images: Mixed (CC BY 4.0 & other licences) |
| AudioCaps (Kim et al., 2019) | Unknown |
| OpenVid (Nan et al., 2024) | CC BY 4.0 |
| ShareGPT4V (Chen et al., 2023) | CC BY-NC 4.0 |
| ShareGPT4Video (Chen et al., 2024b) | CC BY-NC 4.0 |
| Clotho (Zhao et al., 2023a; Drossos et al., 2019) | Captions: CC BY-NC
Audio: Mostly CC BY (version not provided) |
| SydneyCaptions (Qu et al., 2016) | Unknown |
| SensorCaps (Imran et al., 2025) | Hippocratic License[1] |
| CAPDELS (ours) | From CANDELS (Simmons et al., 2016)
CC BY-NC-SA 4.0 |
| SoundBible (Mei et al., 2024) | Custom licence[2] |

[1] Link for the licence
[2] SoundBible website

Unless otherwise specified, all datasets are utilised in their original form, without any alterations. Splits of low-resource modality datasets are adopted according to the specifications provided by each dataset's authors.

**OpenVid**   We access the video zip files (with the template `OpenVid_part{index}.zip`) through HuggingFace (Nan, 2025). We randomly sample 9 zip files out of 185, specifically indices 1, 7, 15, 52, 80, 101, 125, 150, and 174. Moreover, only the first two sentences of the descriptions are used, determined by the `sent_tokenize` in the NLTK Python library (Bird & Loper, 2004).

**ShareGPT4V**   We access the dataset description files through HuggingFace (Chen, 2024). Rather than using the all of the dataset, we only use the images sourced from LLaVA (Liu et al., 2023a) and

Table 30: Evaluation datasets and split sizes (train / validation / test).

| Type | Dataset | Modality | Split Sizes | Task |
|------|---------|----------|-------------|------|
| Seen | SoundBible | Audio | 862 / 184 / 186 | Captioning |
| Unseen Domain | SydneyCaptions | Satellite | 2485 / 290 / 290 | Captioning |
| | RSVQA | Satellite | 2860 / 500 / 500 | Visual Reasoning / VQA |
| | CAPDELS | Astronomical | 4344 / 480 / 1311 | Captioning |
| Unseen Modality | SensorCaps | IMU | 1670 / 209 / 209 | Captioning |
| | OpenSQA | IMU | 1624 / 203 / 204 | Instruction Following |
| | ChEBI-20 | Molecule | 26407 / 3301 / 3300 | Captioning |

SegmentAnything (Kirillov et al., 2023) datasets. Furthermore, if there are multiple descriptions for an image, we consider only one of them.

**ShareGPT4Video** Rather than using full resolution videos, we use videos downscaled to 360P.

**RSVQA** We use low-resolution split and keep the comparison, presence, and rural/urban question types in our dataset. Moreover, we sample a single question from the dataset to transform the task into a low-resource challenge.

**SensorCaps** We remove the final two sentences from the description, as they typically reiterate the preceding text and offer little additional information.

**OpenSQA** OpenSQA examples need to be preprocessed to acquire test examples from teacher LLM outputs. We seperate each question-answer pair, then remove the answer prefix. Afterwards, we randomly sample one pair for each example to transform the task into a low-resource challenge.

**ChEBI-20** In addition to the molecule embedding extracted from the MolCA encoder, we augment the prompt with the SMILES string of the molecule.

## M  TRAINING DETAILS

### M.1  ENCODER DETAILS

In Table 31, we list the checkpoint descriptions of encoders used in projector pre-training or hyper-network training stages. These descriptions are either Hugging Face identifiers or specific checkpoint descriptions enabling reproduction of our results.

Table 31: The checkpoint descriptions for each encoder and LLM used during training phases.

| Encoder | Identifier | Licence |
|---------|-----------|---------|
| CLIP (Radford et al., 2021) | `openai/clip-vit-large-patch14` | MIT |
| CLAP (Elizalde et al., 2023) | `laion/clap-htsat-fused` | CC0 1.0 Universal |
| VideoCLIP-XL (Wang et al., 2024a) | `alibaba-pai/VideoCLIP-XL` | CC-BY-NC-SA-4.0 |
| SigLIP 2 (Tschannen et al., 2025) | `timm/ViT-L-16-SigLIP2-384` | Apache-2.0 |
| Cacophony (Zhu et al., 2024b) | Stage 2 checkpoint | MIT |
| ViCLIP (Wang et al., 2024c) | ViCLIP-B-16, InternVid-10M-FLT checkpoint | Apache-2.0 Licence |
| Zoobot ConvNeXt-Nano | `zoobot-encoder-convnext_nano` | Apache-2.0 |
| Zoobot ConvNeXt-Tiny | `zoobot-encoder-convnext_tiny` | Apache-2.0 |
| Zoobot ConvNeXt-Base | `zoobot-encoder-convnext_base` | Apache-2.0 |
| RemoteCLIP (Liu et al., 2024a) | `chendelong/RemoteCLIP` | Apache-2.0 |
| MolCA | Q-Former embeddings | Public, unspecified |
| BLAT | `blat_cnn14_bertm` checkpoint | CC BY 4.0 |
| Llama 3.1 8B Instruct | `meta-llama/Llama-3.1-8B-Instruct` | Llama 3.1 Community Licence |
| Llama 3.2 1B Instruct | `meta-llama/Llama-3.2-1B-Instruct` | Llama 3.2 Community Licence |

### M.2  COMPUTE RESOURCES

All training processes are done on a single 48 GB NVIDIA RTX-A6000 or an 80 GB A100 GPU with 4 CPUs. For Llama 3.1 8B Instruct, projector pre-training takes approximately 16 hours,

and hypernetwork training takes around 4 days. For Llama 3.2 1B Instruct, projector pre-training takes approximately 7 hours, and hypernetwork training takes around 11 hours. The runtimes of methods are shown in Table 32. Although the runtime of the hypernetwork seems large, this is due to the pre-processing included in the hypernetwork data loader, e.g., processing hypernetwork embeddings, interleaving text and modality embeddings, etc. and not due to more FLOPS. Therefore, this additional time can be eliminated with further code optimisation, effectively diminishing the discrepancy between our method and *FT Projector* and *Projector* baselines. Additionally, adapter generation takes an insignificant amount of time, approximately 80 milliseconds. Feature extraction and training use `float32` precision, except for LLMs which use `bfloat16` precision.

Table 32: Low-resource modality integration runtimes for full dataset sizes. The first value corresponds to Llama 3.1 8B Instruct and the second to Llama 3.2 1B Instruct.

|  | **SydneyCaptions** | **CAPDELS** | **SensorCaps** | **ChEBI-20** |
|---|---|---|---|---|
| **Ours** | 1h19m / 22m | 5h23m / 57m | 66m / 10m | 1d8h / 7h10m |
| **FT Projector** | 57m / 19m | 4h49m / 49m | 58m / 8m | 1d2h / 5h25m |
| **Projector** | 55m / 19m | 4h48m / 48m | 58m / 8m | 1d13h / 5h26m |
| **LoRA** | 48m / 20m | 4h27m / 47m | 52m / 8m | 1d1h / 4h33m |

Table 33 shows the parameter counts across methods. During test-time adaptation (Stage 3), the hypernetwork remains frozen, and SEMI fine-tunes the same number of projector parameters as the baselines. The hypernetwork is trained once on high-resource modalities and reused for all subsequent modalities. The projector parameter count varies slightly across test modalities due to encoder-specific dimensionality requirements.

Table 33: Trainable parameters (in millions) across methods and stages.

| **LLM** | **Training** | | **Adaptation (Stage 3)** | | | |
|---|---|---|---|---|---|---|
|  | **Projector** | **Hypernetwork** | **SEMI** | **FT Projector** | **Projector** | **LoRA** |
| 1B LLM | 5.7M | 73M | 5.7M | 5.7M | 5.7±0.6M | 0.2M |
| 8B LLM | 20M | 100M | 20M | 20M | 20±1M | 0.4M |

## M.3 HYPERPARAMETERS

The hypernetwork training and pre-trained projector hyperparameters are in Tables 36 and 34, respectively. The unseen modality adaptation hyperparameters for full dataset sizes can be seen in Tables 36, 37, and 38. The adaptation hyperparameters are not tuned, generally following the same values except for the learning rate scheduler. As the dataset size decreases, the epochs are multiplied to keep the number of steps (approximately) constant. Better performance for smaller dataset sizes can be achieved with less training, although we do not explicitly aim for training efficiency. Additionally, the batch sizes of SydneyCaptions, CAPDELS, SensorCaps, and SoundBible setups decrease to 16 for the dataset size of 32 to allow stochasticity during adaptation.

Table 34: Pre-trained projector hyperparameters. When two values are provided, the first corresponds to Llama 3.1 8B Instruct and the second to Llama 3.2 1B Instruct.

| | |
|---|---:|
| Optimizer | AdamW (Loshchilov & Hutter, 2019) |
| $(\beta_1, \beta_2)$ | (0.9, 0.95) |
| Weight decay | $5e$-6 |
| Learning rate | $1e$-4 |
| Learning rate scheduler | Linear warmup for 1k steps, then cosine decay to 0 towards the end |
| Warmup steps | 1000 |
| Batch size | 32 / 64 |
| Epochs | 5 |
| $\Rightarrow$ Steps | 108648 / 54325 |
| Projector | |
| Architecture | 2-layer MLP with approximate GELU non-linearity |
| Hidden dimension | 768 |
| Dropout | 0.1 |

Table 35: Hypernetwork training hyperparameters. When two values are provided, the first corresponds to Llama 3.1 8B Instruct and the second to Llama 3.2 1B Instruct.

| | |
|---|---:|
| Optimizer | AdamW |
| $(\beta_1, \beta_2)$ | (0.9, 0.95) |
| Weight decay | $5e$-6 |
| Learning rate | $1e$-4 |
| Learning rate scheduler | Linear warmup for 1k steps, then cosine decay to 0 towards the end |
| Warmup steps | 1000 |
| Batch size | 2 / 4 |
| Gradient accumulation steps | 80 / 40 |
| Subset batch size | 128 |
| Epochs | 5 |
| $\Rightarrow$ Steps | 195505 / 97952 |
| Hypernetwork | |
| Backbone | Self-attention |
| Num. heads | 1 |
| Context length | $259 \ (2 + 1 + 128 \times 2)$ |
| Hidden dimension | 768 |
| Dropout | 0.1 |
| Generated adapter | |
| Rank | 32 |
| Alpha | 32 |

Table 36: *FT Projector* baseline and hypernetwork adaptation hyperparameters for full dataset sizes. When two values are provided, the first corresponds to Llama 3.1 8B Instruct and the second to Llama 3.2 1B Instruct.

| | |
|---|---|
| Optimizer | AdamW |
| $(\beta_1, \beta_2)$ | (0.9, 0.999) |
| Weight decay | 5$e$-6 |
| Learning rate | 1$e$-4 |
| Learning rate scheduler | Constant |
| Dataset-specific hyperparameters | |
| SydneyCaptions | |
| Batch size | 16 / 64 |
| Gradient accumulation steps | 1 |
| Epochs | 60 |
| CAPDELS | |
| Batch size | 8 / 32 |
| Gradient accumulation steps | 1 |
| Epochs | 40 |
| SensorCaps | |
| Batch size | 8 / 32 |
| Gradient accumulation steps | 1 |
| Epochs | 20 |
| ChEBI-20 | |
| Batch size | 2 / 8 |
| Gradient accumulation steps | 32 / 8 |
| Epochs | 10 |
| SoundBible | |
| Batch size | 16 / 64 |
| Gradient accumulation steps | 1 |
| Epochs | 70 |
| Projector | |
| Architecture | 2-layer MLP with approximate GELU non-linearity |
| Hidden dimension | 768 (or equal to enc. dim. if enc. dim. is smaller) |
| Dropout | 0.1 |

Table 37: *Projector* baseline hyperparameters for full dataset sizes. When two values are provided, the first corresponds to Llama 3.1 8B Instruct and the second to Llama 3.2 1B Instruct.

| | |
|---|---:|
| Optimizer | AdamW |
| ($\beta_1$, $\beta_2$) | (0.9, 0.999) |
| Weight decay | $5e$-6 |
| Learning rate | $1e$-4 |
| Learning rate scheduler | Linear warmup, then cosine decay to 0 towards the end |
| Dataset-specific hyperparameters | |
| SydneyCaptions | |
| Batch size | 16 / 64 |
| Gradient accumulation steps | 1 |
| Epochs | 60 |
| Warm-up steps | 100 |
| CAPDELS | |
| Batch size | 8 / 32 |
| Gradient accumulation steps | 1 |
| Epochs | 40 |
| Warm-up steps | 100 |
| SensorCaps | |
| Batch size | 8 / 32 |
| Gradient accumulation steps | 1 |
| Epochs | 20 |
| ChEBI-20 | |
| Batch size | 2 / 8 |
| Gradient accumulation steps | 32 / 8 |
| Epochs | 10 |
| Warm-up steps | 500 |
| SoundBible | |
| Batch size | 16 / 64 |
| Gradient accumulation steps | 1 |
| Epochs | 70 |
| Projector | |
| Architecture | 2-layer MLP with approximate GELU non-linearity |
| Hidden dimension | Equal to enc. dim. |
| Dropout | 0.1 |

Table 38: *LoRA* baseline hyperparameters for full dataset sizes. When two values are provided, the first corresponds to Llama 3.1 8B Instruct and the second to Llama 3.2 1B Instruct.

| | |
|---|---:|
| Optimizer | AdamW |
| $(\beta_1, \beta_2)$ | (0.9, 0.999) |
| Weight decay | 5*e*-6 |
| Learning rate | 1*e*-4 |
| Learning rate scheduler | Constant |
| SydneyCaptions | |
| Batch size | 16 / 64 |
| Gradient accumulation steps | 1 |
| Epochs | 60 |
| CAPDELS | |
| Batch size | 8 / 32 |
| Gradient accumulation steps | 1 |
| Epochs | 40 |
| SensorCaps | |
| Batch size | 8 / 32 |
| Gradient accumulation steps | 1 |
| Epochs | 20 |
| ChEBI-20 | |
| Batch size | 2 / 8 |
| Gradient accumulation steps | 32 / 8 |
| Epochs | 10 |
| SoundBible | |
| Batch size | 16 / 64 |
| Gradient accumulation steps | 1 |
| Epochs | 70 |
| Projector | |
| Architecture | 2-layer MLP with approximate GELU non-linearity |
| Hidden dimension | 768 (or equal to enc. dim. if enc. dim. is smaller) |
| Dropout | 0.1 |
| LoRA | |
| Rank | 32 |
| Alpha | 32 |