# OpenReview forum: "Sample-efficient Integration of New Modalities into Large Language Models"
_ICLR.cc/2026/Conference — Submitted to ICLR 2026_

### Official Review · Reviewer_FcGh · 2025-10-26

**Soundness:** 2
**Presentation:** 3
**Contribution:** 2
**Rating:** 4
**Confidence:** 4

**Summary:**

This paper proposes SEMI (Sample-Efficient Modality Integration), a three-stage framework to integrate new modalities into LLMs. The method leverages a hypernetwork trained on high-resource modalities to generate low-rank adapters for a shared MLP projector, enabling adaptation to unseen low-resource modalities such as satellite images, astronomical data, IMU signals, and molecular structures. The authors introduce a new benchmark and demonstrate that SEMI achieves competitive performance with significantly fewer samples compared to baselines.

**Strengths:**

1. This paper introduces a new dataset CAPDELS, which is a new testbed for related works.

2. The experiment results outperform several baselines, providing valuable empirical reference.

**Weaknesses:**

1. The core idea of using a two-component architecture has been explored in prior works on multimodal adaptation. While SEMI introduces a hypernetwork to generate adapters, the paper does not sufficiently clarify how this differs from fine-tuning or lora-tuning LLM parameters. A more thorough discussion would strengthen the novelty claim.

2. The proposed hypernetwork introduces a non-negligible number of additional trainable parameters during the second training stage. In contrast, baselines like LoRA or direct projector fine-tuning operate with fewer or no extra parameters. This might makes an unfair comparison.

3. The evaluation is exclusively conducted on captioning tasks across all modalities. While captioning is a common benchmark, it represents only one type of multimodal integration. Given that the paper proposes a general framework for LLMs' modality integration, broader task coverage would be necessary to substantiate this claim.

**Questions:**

See `Weakness' section.

---

> ### Author Response · Authors · 2025-11-19
> **Thank your for your review! (Part 1)**
>
> We thank **FcGh** for recognizing our CAPDELS dataset as a useful benchmark and SEMI's strong experimental performance across multiple baselines.
>
> **[W1] Novelty clarification: SEMI as a training method, not architecture**
>
> Thank you for raising this important point. We think there may be a misunderstanding:
>
> **SEMI is not an alternative to two-component architectures, but rather a training method that enables these architectures to work in low-resource multimodal settings.** In standard two-component architectures, when integrating a new modality, only the projector (between encoder and LLM) is trained while the LLM remains frozen. This is the common practice we follow.
>
> **SEMI's contribution:** We introduce a hypernetwork that **generates LoRA adapters for the projector (not the LLM)**. Given minimal examples from a new modality, the hypernetwork adapts a pre-trained projector, which is then fine-tuned. Crucially, the LLM parameters remain frozen throughout; all adaptation occurs on the projector.
>
> Our baselines (FT Projector, LoRA, Projector) operate within this same framework, differing only in how they initialise/adapt the projector. SEMI's novelty lies in learning the fundamental structure of modality integration from high-resource modalities, enabling generalization to low-resource modalities with orders of magnitude less data.
>
> **[W2] Parameter count during adaptation**
>
> We respectfully disagree that the comparison is unfair. During test-time adaptation (Stage 3), the hypernetwork is frozen. SEMI fine-tunes the merged projector with identical number of parameters to Projector and FT Projector, yet substantially outperforms both (e.g., 220 vs 61 and 94 CIDEr; Table 12). The LoRA baseline has fewer trainable parameters but uses rank 32, matching the hypernetwork's adapter rank.
>
> The hypernetwork is trained once on high-resource modalities and reused for all subsequent low-resource modalities, amortizing its one-time cost. SEMI's superior performance stems from better initialization, not from parameter count advantages during adaptation.
>
> Additionally, enlarging the projector (20M) to match hypernetwork parameters (125M) would harm both sample efficiency (more parameters to fine-tune) and inference efficiency (slower forward passes). The hypernetwork's parameter count (125M) remains significantly smaller than the encoders (~200-400M) and LLM decoder (8B).
>
> We thank the reviewer for raising this important point and hope this clarifies the fairness of our comparison.

---

> > ### Author Response · Authors · 2025-11-19
> > **Thank you for your review! (Part 2)**
> >
> > **[W3] Exploring additional tasks: visual reasoning & instruction following**
> >
> > Thank you for this insightful suggestion. To demonstrate SEMI’s relevance to tasks requiring semantic understanding and reasoning, we conducted new experiments on the visual reasoning / question answering task RSVQA [1] (for satellite data) and instruction-following task OpenSQA [2] (for IMU sensor data). We again used RemoteCLIP-ViT-L [3] and LIMU-BERT [4] as encoders, as in our captioning experiments.
> >
> > **Visual Reasoning / Question Answering Results (RSVQA, 3 random seeds)**
> >
> > | **Exact Match** | **32** | **128** | **512** | **2048** | **2860** |
> > |---|---|---|---|---|---|
> > | **SEMI** | **69.87 ± 1.46** | **77.27 ± 0.79** | **82.67 ± 1.30** | **84.13 ± 0.47** | **83.13 ± 1.20** |
> > | FT Projector | 24.88 ± 1.55 | 32.12 ± 0.88 | 31.48 ± 0.86 | 34.60 ± 0.94 | 35.48 ± 0.78 |
> > | Projector | 45.96 ± 6.54 | 48.44 ± 11.02 | 63.08 ± 9.84 | 58.76 ± 10.08 | 65.48 ± 10.94 |
> > | LoRA | 24.04 ± 1.31 | 27.56 ± 0.72 | 26.12 ± 0.90 | 25.20 ± 2.06 | 27.40 ± 1.22 |
> >
> > | **F1 Score** | **32** | **128** | **512** | **2048** | **2860** |
> > |---|---|---|---|---|---|
> > | **SEMI** | **69.93 ± 1.47** | **77.27 ± 0.79** | **82.67 ± 1.30** | **84.49 ± 0.66** | **83.44 ± 1.33** |
> > | FT Projector | 34.47 ± 2.97 | 44.41 ± 1.04 | 45.13 ± 1.39 | 50.37 ± 1.51 | 50.48 ± 1.20 |
> > | Projector | 53.05 ± 5.36 | 55.53 ± 9.69 | 67.32 ± 7.45 | 65.21 ± 7.56 | 69.33 ± 9.18 |
> > | LoRA | 31.15 ± 2.00 | 36.87 ± 1.22 | 34.76 ± 1.28 | 34.57 ± 3.11 | 38.55 ± 1.73 |
> >
> > As shown, SEMI substantially outperforms all baselines across dataset sizes. Notably, SEMI never observed visual reasoning tasks during training, yet demonstrates strong cross-task generalisation while FT Projector baseline fails to adapt. For instance, at 32 samples, SEMI achieves 69.9% exact match compared to FT Projector's 24.9% – a performance gap that persists even at full dataset scale (83.1% vs 35.5%).
> >
> > **Instruction Following Results (OpenSQA, 3 random seeds)**
> >
> > | **ROUGE-L** | **32** | **128** | **512** | **1624** |
> > |---|---|---|---|---|
> > | **SEMI** | **19.06 ± 0.12** | **18.89 ± 0.02** | **20.12 ± 0.36** | **20.99 ± 0.29** |
> > | FT Projector | 17.95 ± 0.05 | 17.74 ± 0.11 | 18.53 ± 0.16 | 18.85 ± 0.08 |
> > | Projector | 18.25 ± 0.17 | 18.87 ± 0.68 | 19.66 ± 0.49 | 20.73 ± 0.34 |
> > | LoRA | 17.88 ± 0.14 | 18.02 ± 0.08 | 18.99 ± 0.17 | 19.29 ± 0.15 |
> >
> > | **ROUGE-2** | **32** | **128** | **512** | **1624** |
> > |---|---|---|---|---|
> > | **SEMI** | **7.43 ± 0.11** | **7.64 ± 0.04** | **8.65 ± 0.11** | **9.02 ± 0.12** |
> > | FT Projector | 7.41 ± 0.03 | 7.14 ± 0.10 | 7.61 ± 0.07 | 7.84 ± 0.08 |
> > | Projector | 7.22 ± 0.25 | 7.48 ± 0.31 | 8.15 ± 0.32 | 8.79 ± 0.25 |
> > | LoRA | 7.33 ± 0.11 | 7.39 ± 0.07 | 7.98 ± 0.08 | 8.20 ± 0.13 |
> >
> > SEMI again outperforms all baselines. This is particularly notable as SEMI was not trained on diverse instruction-following tasks, suggesting these results may represent a lower bound on SEMI's capabilities for this task family.
> >
> > These results demonstrate that SEMI generalises across diverse task families including captioning, visual reasoning, and instruction-following, validating its broad applicability.
> >
> > **[1]**: Lobry, et al. "RSVQA: Visual question answering for remote sensing data." IEEE Transactions on Geoscience and Remote Sensing 58.12 (2020): 8555-8566.
> > **[2]**: Imran, et al. "Llasa: A multimodal llm for human activity analysis through wearable and smartphone sensors." arXiv, 2024.
> > **[3]**: Fan, et al. "Remoteclip: A vision language foundation model for remote sensing." IEEE Transactions on Geoscience and Remote Sensing, 2024.
> > **[4]**: Xu, et al. "Limu-bert: Unleashing the potential of unlabeled data for imu sensing applications." SenSys, 2021.

---

> ### Author Response · Authors · 2025-11-28
> **Gentle Reminder: Reviewer Response to Rebuttal**
>
> Dear **FcGh**,
>
> We appreciate your review and believe we have clarified the key points. We would greatly appreciate your consideration in raising your score:
>
> **Novelty (W1):** We clarified that SEMI is a training method that enables two-component architectures to work in low-resource settings, not an alternative to them. SEMI's novelty is learning the fundamental structure of modality integration from high-resource modalities to generalise to low-resource ones with orders of magnitude less data.
>
> **Parameter fairness (W2):** We respectfully disagree that the comparison is unfair. During test-time adaptation, the hypernetwork is frozen and SEMI fine-tunes identical parameters as the baselines (Table 33), yet substantially outperforms both (220 vs 61 and 94 CIDEr). The one-time hypernetwork training cost is amortised across all future modalities, and superior performance stems from better initialisation rather than parameter advantages.
>
> **Task breadth (W3):** We conducted new experiments on satellite visual reasoning (RSVQA) and IMU instruction-following (OpenSQA). SEMI achieves 69.9% exact match vs 24.9% for FT Projector with 32 samples, demonstrating generalisation across diverse task families beyond captioning.
>
> We remain available for further discussion.

---

### Official Review · Reviewer_nnAo · 2025-10-30

**Soundness:** 2
**Presentation:** 2
**Contribution:** 2
**Rating:** 6
**Confidence:** 3

**Summary:**

This paper focuses on how to integrate new modality into pre-existing foundation models. A method for sample-efficient modality integration is proposed, namely SEMI. A hyper-network is proposed that adapt a shared projector--placed between modality-specific encoders and a LLM decoder--to any modality. SEMI achieves performance boost in sample efficiency during few-shot integration of new modalities.

**Strengths:**

1. It is important to achieve integrating new modality into a pre-existing foundation model.

2. Method is clearly illustrated throw simple figure and detailed text.

**Weaknesses:**

1. What's the differences between the proposed method against the Bind-style frameworks?

[A] Girdhar R, El-Nouby A, Liu Z, et al. Imagebind: One embedding space to bind them all[C]//Proceedings of the IEEE/CVF conference on computer vision and pattern recognition. 2023: 15180-15190.

[B] Lyu Y, Zheng X, Zhou J, et al. Unibind: Llm-augmented unified and balanced representation space to bind them all[C]//Proceedings of the IEEE/CVF Conference on Computer Vision and Pattern Recognition. 2024: 26752-26762.

[C] Zhu B, Lin B, Ning M, et al. LanguageBind: Extending Video-Language Pretraining to N-modality by Language-based Semantic Alignment[C]//The Twelfth International Conference on Learning Representations.

It would be better to clearly discuss this line of research with SEMI.

2. The experimental results in the main paper are limited. The reviewer totally understand there are more in the appendix, however, the paper needs better arrangement of layout and content distribution.

**Questions:**

## Relation to Bind-style Frameworks (ImageBind, UniBind, LanguageBind)

The current manuscript does not clearly situate the proposed method within the broader line of Bind-style multi-modal frameworks such as ImageBind [Girdhar et al., CVPR 2023], UniBind [Lyu et al., CVPR 2024], and LanguageBind [Zhu et al., ICLR 2024]. These works also target unified and sample-efficient representation learning. A deeper comparison would strengthen the contribution.


### Question 1:
How does the proposed method differ conceptually from the Bind-style frameworks that unify representations across modalities?

### Question 2:
Have the authors considered incorporating or comparing SEMI with Bind-style pretrained embeddings (e.g., using ImageBind features as state encoders)? A brief empirical or conceptual comparison could clarify whether SEMI complements or diverges from representation-binding paradigms.

### Question 3:
Can SEMI be regarded as a task-level integration framework (fusing learning paradigms) while Bind-style frameworks operate at the representation-level? If so, could combining the two approaches further enhance sample efficiency or generalization? This discussion would be beneficial to include in the related work or discussion section.

---

> ### Author Response · Authors · 2025-11-19
> **Thank you for your review!**
>
> We thank **nnAo** for acknowledging the significance of resource-efficient modality integration and our clear presentation through figures and comprehensive descriptions.
>
> **[W1] Relationship to Bind-Style frameworks (ImageBind/LanguageBind/...)**
>
> Thank you for raising this important line of work. SEMI differs from Bind-style frameworks in several ways:
>
> 1. **Sample efficiency:** SEMI integrates unseen modalities with as few as 32 modality-text pairs, while Bind-style methods require far larger datasets (e.g., LanguageBind [C] uses millions of pairs, ImageBind [A] uses thousands).
> 2. **Complementarity:** These approaches can be combined. SEMI could enable sample-efficient integration of new modalities into Bind-style anchor encoders (e.g., image in ImageBind, text in LanguageBind).
> 3. **Approach:** Bind-style frameworks explicitly align modalities into a shared embedding space, whereas SEMI focuses on modality-oriented integration into LLMs without enforcing explicit alignment.
> 4. **Practical limitations:** Some Bind methods (e.g., UniBind [B]) rely on ImageNet-1K concept centers, which do not suit modalities like astronomical images or molecules. SEMI avoids this by not requiring such correspondences.
>
> In short, Bind-style frameworks unify representations across modalities, while SEMI enables sample-efficient, modality-focused integration into LLMs.
>
> We will add this discussion to the related work section.
>
> **[D]**: Deng, Jia, et al. "Imagenet: A large-scale hierarchical image database." CVPR, 2009.
>
> **[W2] Improved content distribution in revision**
>
> We acknowledge the reviewer’s concerns. We will utilise the extra page in the camera-ready version for the additional experimental results.
>
> **[Q1] Conceptual differences between SEMI and Bind-style frameworks**
>
> We kindly refer the reviewer to W1 above.
>
> **[Q2] Incorporating Bind-style pre-trained embeddings**
>
> Yes, we discuss ImageBind-LLM in Related Work (L119–123), which uses ImageBind as an encoder and achieves data efficiency during MLLM training by relying on only image–text pairs. However, ImageBind requires thousands of low-resource modality–image pairs during its representation alignment phase, so it does not address the low-resource modality integration problem that SEMI (which integrates modalities with as few as 32 pairs) is designed to solve.
>
> **[Q3] Is SEMI a task-level integration framework?**
>
> SEMI is not a task-level integration framework but rather a task-agnostic modality integration framework. While SEMI uses instruction embeddings, these can be generic (e.g., "Complete the task given an input from the \{\<MODALITY\>\} modality"), and a single trained hypernetwork serves all downstream tasks for a given modality. This task-agnostic nature is empirically confirmed: SEMI successfully generalises to visual reasoning and instruction-following (which is inherently multi-task) despite never observing these task families during hypernetwork training (see Reviewer ZybR - W2).

---

> > ### Comment · Reviewer_nnAo · 2025-11-25
> > **Regarding: "Practical limitations: Some Bind methods (e.g., UniBind [B]) ..."**
> >
> > Thanks for the authors' responses.
> >
> > The reviewer **does not agree with the claim** that: "Practical limitations: Some Bind methods (e.g., UniBind [B]) rely on ImageNet-1K concept centers, which do not suit modalities like astronomical images or molecules. SEMI avoids this by not requiring such correspondences."
> >
> > Could the authors explain more about this? Why the authors believe that Bind models rely on ImageNet-1K's concept centers? And why can SEMI avoid this by neglecting the mentioned correspondence?
> >
> > Thanks.

---

> ### Author Response · Authors · 2025-11-25
> **Answering 'Regarding: "Practical limitations: Some Bind methods (e.g., UniBind [B]) ..."'**
>
> Thank you for this question. We would like to clarify two points:
>
> **First, regarding scope:**
>
> Our claim states "**Some Bind methods (e.g., UniBind [B])**" have this limitation – **not all Bind models generally**. We're highlighting a specific limitation of UniBind's approach as the reviewer mentioned it specifically, not characterising all Bind-style frameworks.
>
> **Second, UniBind's specific limitations:**
>
> UniBind requires: **(1)** pre-defining category sets and generating 1000 text descriptions per category via LLMs at test time, and **(2)** using MLLMs (BLIP-2, LLaMA-Adapter) to generate descriptions of modality data for building their knowledge base.
>
> For the low-resource modalities we study (IMU sensor data, molecules, astronomical images), this creates a circular dependency: UniBind requires MLLMs that can describe these modalities, but such MLLMs don't exist precisely because these are low-resource modalities.
>
> UniBind explicitly constructs category-based embedding centers: "*We generate 1,000 descriptions for each category name via LLMs (GPT-4, LLaMA) and generate multi-modal data descriptions via multi-modal LLMs (BLIP-2, LLaMA-Adapter). We construct our knowledge base with these two sets of description texts*" (Section 4.1). Their classification relies on computing similarity to these embedding centers: "*With our embedding centers, multi-modal data distributed at the boundary of the category representation space can effectively avoid interference from other categories*" (Section 3.4).
> Even if one bypassed the MLLM requirement by using only LLMs, UniBind's performance drops substantially (their Figure 8a shows ~9 percentage points lower accuracy for "Only LLM" vs "LLM & MLLM").
>
> In contrast, SEMI generates adapters directly from a few modality-text pairs without requiring pre-defined categories, MLLMs for the target modality, or task-specific constraints.

---

> ### Author Response · Authors · 2025-12-03
> **Gentle Reminder: Reviewer Response to Rebuttal**
>
> Dear **nnAo**,
>
> Thank you for your positive assessment. We believe we have provided comprehensive responses to all your questions:
>
> **Bind-style frameworks (W1, Q1-Q3):** We have thoroughly differentiated SEMI from ImageBind, UniBind, and LanguageBind across multiple dimensions. **Key differences include sample efficiency** (SEMI requires only 32 pairs versus millions for Bind-style methods), **complementarity** (SEMI could enhance Bind-style frameworks), and **practical advantages** (SEMI does not require ImageNet concept correspondences). We have also clarified that SEMI is a task-agnostic modality integration framework, demonstrated by its generalisation to visual reasoning and instruction-following tasks never seen during training.
>
> **Layout concerns (W2):** We utilised the camera-ready extra page to incorporate additional experimental results into the main paper. We greatly appreciate your recognition of our contribution.

---

### Official Review · Reviewer_MrGx · 2025-10-31

**Soundness:** 2
**Presentation:** 3
**Contribution:** 2
**Rating:** 4
**Confidence:** 3

**Summary:**

The paper studied a sample-efficient approach to integrating new modalities (with low resources) into large language models by training a hypernetwork that can adapt a shared projector and isometric transformation of encoder output. Also, the authors contributed to preparing benchmarks and creating a new dataset. The overall writing and organization of the paper are good.

**Strengths:**

1) The background, motivation, and problem statement are well described and justified. It is a meaningful and valuable problem to be addressed.
2) The proposed method is simple and straightforward, and does not require complicated theory and modules.
3) The experiment aligns with the objective of the work, supporting the claim of sample efficiency.

**Weaknesses:**

1) In line 111, modality-specific encoders are assumed. In many real-world problems, the encoders might not be available. This assumption is quite restrictive.
2) The proposed method is compared with weak baselines. While authors explained that they might require large-scale paired data, a more comprehensive comparative analysis can be provided to show the tradeoff between data efficiency and performance. To show that this work extends the SOTA in a certain way.
3) The number of new modalities studied is limited, and the generalization and scalability of the work are unclear.
4) The proposed method is only assessed on integrating one extra modality each time and does not evaluate its performance when multiple modalities are simultaneously used for cross-modal synergy.

**Questions:**

1) The proposed method just appears to be a better way of initializing the projector? Hence, better than weak baselines?
2) What is the performance of the method? If the third stage is ablated.

---

> ### Author Response · Authors · 2025-11-19
> **Thank you for your review! (Part 1)**
>
> We thank **MrGx** for noting our well-motivated problem formulation, straightforward approach without complex components, and experimental validation that confirms our sample efficiency objectives.
>
> **[W1] Encoder availability: evidence across low-resource modalities**
>
> As noted in our discussion on L477, we assume that such encoders are available. However, we do not believe this to be an unreasonable assumption. We kindly refer MrGx to Appendix B (L956-962) where we discuss the availability of low-resource modality encoders in detail. Specifically, we analyse six additional prominent low-resource modalities and found that all of them have available encoders.
>
> **[W2] Baseline strengths and experimental trade-off analysis**
>
> Thank you for raising this concern. We respectfully disagree that our baselines are weak; as they represent current state-of-the-art approaches for modality integration:
>
> 1. **Projector from scratch:** Used in leading vision LLMs (Llava 1.5 [1], InternVL [2], Qwen-VL [3], and more [4, 5, 6])
> 2. **Fine-tuning a pre-trained projector:** Adapted from seminal OneLLM's [7] incremental modality integration framework
> 3. **Adding LoRA adapters to a pre-trained projector:** Adapted from PathWeave [8], a state-of-the-art incremental modality integration technique
>
> We note that many other multimodal integration methods require orders of magnitude more paired data, making them unsuitable as baselines in our low-resource setting. Since the setting is novel, there are no established baselines. It is incumbent upon the Reviewer to indicate which one, if any, is missing.
>
> Regarding data-efficiency vs. performance: our experimental results report exactly this tradeoff. Figures 2-5 plot performance (Y-axis) against dataset size (X-axis) across all methods, explicitly showing how SEMI achieves superior performance at every data regime. This is the core contribution we claim: maintaining strong performance while dramatically reducing data requirements.
>
> **[1]**: Liu, et al. "Improved baselines with visual instruction tuning." CVPR 2024.
>
> **[2]**: Chen, et al. "Internvl: Scaling up vision foundation models and aligning for generic visual-linguistic tasks." CVPR 2024.
>
> **[3]**: Bai, et al. "Qwen technical report." arXiv, 2023.
>
> **[4]**: Wang, et al. "CogVLM: Visual expert for pretrained language models." NeurIPS 2024.
>
> **[5]**: Lu, et al. "Deepseek-VL: Towards real-world vision-language understanding." arXiv, 2024.
>
> **[6]**: Wu, et al. "Janus: Decoupling visual encoding for unified multimodal understanding and generation." CVPR, 2025.
>
> **[7]**: Han, Jiaming, et al. "OneLLM: One framework to align all modalities with language." CVPR, 2024.
>
> **[8]**: Yu, Jiazuo, et al. "LLMs Can Evolve Continually on Modality for X-Modal Reasoning." NeurIPS, 2024.
>
> **[W3] Modality coverage: from domain shift to novel structures**
>
> Thank you for raising this concern. We test SEMI on five diverse setups spanning a spectrum, from most similar to most dissimilar wrt the training data: namely audio (unseen encoder), satellite images and astronomical images (novel domains), molecules and IMU sensor data (new modalities with different structures). Most importantly, molecules and IMU sensor data are different from the training modalities structurally (graphs and multi-channel signals) and in terms of text distributions. This range demonstrates SEMI's generalisation capabilities. Additionally, following reviewer feedback, we conducted new experiments showing SEMI generalises across task families including visual reasoning and instruction-following (see W3 of Reviewer FcGh).
>
> **[W4] Measuring cross-modal synergy**
>
> This is a very interesting suggestion, but we believe that it is out of scope, as this is the first work on the topic of sample-efficient modality integration. We will list this idea as potential future work at the end of the paper.

---

> > ### Author Response · Authors · 2025-11-19
> > **Thank you for your review! (Part 2)**
> >
> > **[Q1] Learned initialisation as core sample-efficiency mechanism**
> >
> > We respectfully disagree with characterising this as "just" better initialisation. This is the very essence of all sample-efficient methods, from pre-training/fine-tuning to meta-learning: by providing a clever inductive bias (in the form of initialisation), learning can occur from a few examples. For instance, MAML focuses on providing a good initialisation that can rapidly adapt to a new task [9]. Similarly, SEMI’s hypernetwork provides a good projector initialisation that enables sample-efficient integration through fine-tuning. This fine-tuning is essential to address the extreme distribution shifts to test modalities and to learn the complex patterns that require iterative refinement beyond a forward pass.
> >
> > We address baseline strength concerns in [W2].
> >
> > **[9]**: Finn, et al. "Model-agnostic meta-learning for fast adaptation of deep networks." ICML.  2017.
> >
> > **[Q2] Ablation without fine-tuning**
> >
> > Thank you for this question. We conducted ablations evaluating SEMI and FT Projector without fine-tuning on CAPDELS and SydneyCaptions datasets.
> >
> > **Zero-shot adaptation results (No fine-tuning)**
> >
> > | **BLEU** | **Zero-shot** |
> > |---|---|
> > | SEMI - CAPDELS Base |  0.41 |
> > | FT Projector - CAPDELS Base | 0.02 |
> > | SEMI - SydneyCaptions RN-50 | 0.45 |
> > | FTProjector – SydneyCaptions RN-50 | 0.01 |
> >
> > **Few-shot adaptation results (With fine-tuning)**
> >
> > | **BLEU** | **32** | **128** | **512** | **2048** | **Full** |
> > |---|---|---|---|---|---|
> > | SEMI - CAPDELS Base | 46.52 | 53.57 | 56.38 | 59.31 | 61.27 |
> > | FT Projector - CAPDELS Base | 31.26 | 41.58 | 49.71 | 53.93 | 58.11 |
> > | SEMI - SydneyCaptions RN-50 | 43.88 | 41.87 | 47.38 | 49.18 | 48.03 |
> > | FT Projector -  SydneyCaptions RN-50 | 39.18 | 37.55 | 44.64 | 47.26 | 49.71 |
> >
> > As expected, zero-shot performance is limited for both methods (SEMI and FT Projector). The results show that fine-tuning is essential to address extreme distribution shifts to test modalities. However, the critical finding is that SEMI achieves dramatic performance gains with minimal fine-tuning (e.g., 0.41 $\rightarrow$ 46.52 with just 32 examples on CAPDELS), while FT Projector shows much smaller gains (0.02 $\rightarrow$ 31.26). This demonstrates that the hypernetwork provides an initialisation specifically suited for rapid adaptation to new modalities.
> >
> > We will add these zero-shot results and discussion to the revised manuscript.

---

> ### Author Response · Authors · 2025-11-28
> **Gentle Reminder: Reviewer Response to Rebuttal**
>
> Dear **MrGx**,
>
> We appreciate your feedback and believe we have addressed your concerns. We would greatly appreciate your consideration in raising your score:
>
> **Encoder availability (W1):** Appendix B analyses six additional low-resource modalities, confirming encoders are available for all of them.
>
> **Baseline strength (W2):** We respectfully disagree that our baselines are weak: they represent current SOTA (LLaVA 1.5, InternVL, OneLLM, PathWeave). Since this is a novel setting without established baselines, we have constructed the most appropriate comparisons. Figures 2-5 explicitly show the requested data-efficiency versus performance tradeoff.
>
> **Test modality diversity (W3):** We test five diverse setups from most similar (audio with new encoder) to most dissimilar (molecules & IMU, structurally different modalities). We also added visual reasoning and instruction-following experiments.
>
> **Better initialisation (Q1):** We respectfully disagree with "just" better initialisation. This is the essence of sample-efficient methods: providing clever inductive bias enables learning from few examples, analogous to pre-training/fine-tuning and meta-learning (MAML).
>
> **Zero-shot ablation (Q2):** We provided the ablation showing SEMI achieves much larger gains with minimal fine-tuning (0.41 $\rightarrow$ 46.52 BLEU) compared to FT Projector (0.02 $\rightarrow$ 31.26).
>
> We remain open to any additional experiments you would find valuable.

---

### Official Review · Reviewer_ZybR · 2025-10-31

**Soundness:** 2
**Presentation:** 2
**Contribution:** 2
**Rating:** 4
**Confidence:** 3

**Summary:**

This paper proposes SEMI, a sample-efficient modality integration method. Instead of retraining a new projector for each modality, SEMI employs a hypernetwork trained on multiple high-resource modalities (e.g., image, audio, video) to learn how to generate LoRA adapters for a shared projector. During adaptation to a new, low-resource modality (e.g., molecules), the trained hypernetwork observes only a few paired examples to produce a modality-specific adapter, which is then further fine-tuned on the same few-shot set to integrate the new modality into the LLM.
To enhance generalization, the authors introduce isometric transformations to artificially diversify encoder representations and apply text grounding to align modality embeddings with textual semantics. Experiments show that SEMI achieves strong sample efficiency. For instance, with only 32 examples, it matches the performance of baseline methods (e.g., training the projector or LoRA adapter from scratch) that require 16× even 64× more data.

**Strengths:**

Strengths:

1.The paper targets a practical yet underexplored problem. As AI increasingly extends into scientific and industrial domains, this work provides a solution for efficiently integrating new modalities into frozen large language models.

2.On several unseen modalities, particularly those that differ significantly from training modalities, such as IMU signals and molecular structures, SEMI demonstrates clear advantages in the low-data regime.

3.The evaluation spans a diverse set of modalities, supporting the method’s data efficiency and generalization robustness.

**Weaknesses:**

1.Stage 3 consists of two steps: (1) the hypernetwork generates a LoRA adapter ; and (2)  is then fine-tuned on the same K-shot samples. The paper lacks a critical ablation: what happens if the generated adapter is used without fine-tuning (i.e., zero-shot adaptation)? If performance remains strong, the hypernetwork truly learns to generate; if not, it may merely provide a better initialization while fine-tuning does most of the adaptation. The current setup mixes these effects.

2.Current experiments focus solely on captioning tasks, with no assessment of semantic understanding or reasoning capabilities.

3.The work lacks comparisons to cross-attention or Q-Former–style multimodal adapters, which are the most direct architectural competitors.

4.The paper discusses the hypernetwork’s training challenges only qualitatively, without reporting runtime or resource usage. Moreover, the current projector is a two-layer MLP, and the hypernetwork generates LoRA parameters for its first layer. For more complex projectors (e.g., Q-Former), the generation burden could grow rapidly, leading to scalability and optimization issues. The authors mention training challenges only qualitatively, without a deeper discussion of scalability limitations.

The claim that random orthogonal matrices “emulate numerous encoders” is overstated. Geometrically, such transformations (e.g., rotations) preserve inner products and pairwise distances, merely changing the coordinate system rather than altering the data distribution or encoder behavior. They act more as a robustness augmentation than as true encoder simulation.

**Questions:**

See weaknesses

---

> ### Author Response · Authors · 2025-11-19
> **Thank you for your review! (Part 1)**
>
> We thank **ZybR** for recognising our work tackles an important yet understudied challenge with strong few-shot performance on varied modalities (IMU sensors, molecules) and comprehensive evaluation demonstrating sample efficiency and robust generalisation.
>
> **[W1] Ablation: Zero-shot adaptation results**
>
> We agree with **ZybR** that our evaluation results compound the effects of merging the generated LoRA adapter and further fine-tuning them. Hence, to disentangle them, we run an ablation where SEMI generates adapters without any additional fine-tuning.
>
> On the other hand, we respectfully disagree on the interpretation of adapter generation as “*merely provid[ing] a better initialization*”. This is the very essence of all sample-efficient methods, from pre-training/fine-tuning to meta-learning: by providing a clever inductive bias (in the form of initialisation), learning can occur from a few examples. For instance, MAML [1] focuses on providing a good initialisation that can rapidly adapt to a new task [1]. Similarly, SEMI’s hypernetwork provides a good projector initialisation that enables sample-efficient integration through fine-tuning. This fine-tuning is essential to address the extreme distribution shifts to test modalities.
>
> **Zero-shot adaptation results (No fine-tuning)**
>
> | **BLEU** | **Zero-shot** |
> |---|---|
> | SEMI - CAPDELS Base |  0.41 |
> | FT Projector - CAPDELS Base | 0.02 |
> | SEMI - SydneyCaptions RN-50 | 0.45 |
> | FTProjector – SydneyCaptions RN-50 | 0.01 |
>
> **Few-shot adaptation results (With fine-tuning)**
>
> | **BLEU** | **32** | **128** | **512** | **2048** | **Full** |
> |---|---|---|---|---|---|
> | SEMI - CAPDELS Base | 46.52 | 53.57 | 56.38 | 59.31 | 61.27 |
> | FT Projector - CAPDELS Base | 31.26 | 41.58 | 49.71 | 53.93 | 58.11 |
> | SEMI - SydneyCaptions RN-50 | 43.88 | 41.87 | 47.38 | 49.18 | 48.03 |
> | FT Projector -  SydneyCaptions RN-50 | 39.18 | 37.55 | 44.64 | 47.26 | 49.71 |
>
> As expected, zero-shot adaptation results are significantly worse than few-shot adaptations for both SEMI and FT Projector. However, we want to emphasise that SEMI gets a huge boost with fine-tuning on as few as 32 examples (e.g. 0.39 $\rightarrow$ 46.01), while FT Projector fails to show a similar performance gain (e.g. 0.02 $\rightarrow$ 31.26).
>
> We appreciate these comments, and we will add the zero-shot adaptation results and a discussion regarding its challenges in the revised text. Thanks for the suggestion.
>
> **[1]**: Finn, et al. "Model-agnostic meta-learning for fast adaptation of deep networks." ICML.  2017.

---

> > ### Author Response · Authors · 2025-11-19
> > **Thank you for your review! (Part 2)**
> >
> > **[W2] Exploring additional tasks: visual reasoning & instruction following**
> >
> > Thank you for this insightful suggestion. To demonstrate SEMI’s relevance to tasks requiring semantic understanding and reasoning, we conducted new experiments on the *visual reasoning / question answering* task RSVQA [2] (for satellite data) and *instruction-following* task OpenSQA [3] (for IMU sensor data). We again used RemoteCLIP-ViT-L [4] and LIMU-BERT [5] as encoders, as in our captioning experiments.
> >
> > **Visual Reasoning / Question Answering Results (RSVQA, 3 random seeds)**
> >
> > | **Exact Match** | **32** | **128** | **512** | **2048** | **2860** |
> > |---|---|---|---|---|---|
> > | **SEMI** | **69.87 ± 1.46** | **77.27 ± 0.79** | **82.67 ± 1.30** | **84.13 ± 0.47** | **83.13 ± 1.20** |
> > | FT Projector | 24.88 ± 1.55 | 32.12 ± 0.88 | 31.48 ± 0.86 | 34.60 ± 0.94 | 35.48 ± 0.78 |
> > | Projector | 45.96 ± 6.54 | 48.44 ± 11.02 | 63.08 ± 9.84 | 58.76 ± 10.08 | 65.48 ± 10.94 |
> > | LoRA | 24.04 ± 1.31 | 27.56 ± 0.72 | 26.12 ± 0.90 | 25.20 ± 2.06 | 27.40 ± 1.22 |
> >
> > | **F1 Score** | **32** | **128** | **512** | **2048** | **2860** |
> > |---|---|---|---|---|---|
> > | **SEMI** | **69.93 ± 1.47** | **77.27 ± 0.79** | **82.67 ± 1.30** | **84.49 ± 0.66** | **83.44 ± 1.33** |
> > | FT Projector | 34.47 ± 2.97 | 44.41 ± 1.04 | 45.13 ± 1.39 | 50.37 ± 1.51 | 50.48 ± 1.20 |
> > | Projector | 53.05 ± 5.36 | 55.53 ± 9.69 | 67.32 ± 7.45 | 65.21 ± 7.56 | 69.33 ± 9.18 |
> > | LoRA | 31.15 ± 2.00 | 36.87 ± 1.22 | 34.76 ± 1.28 | 34.57 ± 3.11 | 38.55 ± 1.73 |
> >
> > As shown, SEMI substantially outperforms all baselines across dataset sizes. Notably, SEMI never observed visual reasoning tasks during training, yet demonstrates strong cross-task generalisation where the FT Projector baseline fails to adapt. For instance, at 32 samples, SEMI achieves 69.9% exact match compared to FT Projector's 24.9% – a performance gap that persists even at full dataset scale (83.1% vs 35.5%).
> >
> > **Instruction Following Results (OpenSQA, 3 random seeds)**
> >
> > | **ROUGE-L** | **32** | **128** | **512** | **1624** |
> > |---|---|---|---|---|
> > | **SEMI** | **19.06 ± 0.12** | **18.89 ± 0.02** | **20.12 ± 0.36** | **20.99 ± 0.29** |
> > | FT Projector | 17.95 ± 0.05 | 17.74 ± 0.11 | 18.53 ± 0.16 | 18.85 ± 0.08 |
> > | Projector | 18.25 ± 0.17 | 18.87 ± 0.68 | 19.66 ± 0.49 | 20.73 ± 0.34 |
> > | LoRA | 17.88 ± 0.14 | 18.02 ± 0.08 | 18.99 ± 0.17 | 19.29 ± 0.15 |
> >
> > | **ROUGE-2** | **32** | **128** | **512** | **1624** |
> > |---|---|---|---|---|
> > | **SEMI** | **7.43 ± 0.11** | **7.64 ± 0.04** | **8.65 ± 0.11** | **9.02 ± 0.12** |
> > | FT Projector | 7.41 ± 0.03 | 7.14 ± 0.10 | 7.61 ± 0.07 | 7.84 ± 0.08 |
> > | Projector | 7.22 ± 0.25 | 7.48 ± 0.31 | 8.15 ± 0.32 | 8.79 ± 0.25 |
> > | LoRA | 7.33 ± 0.11 | 7.39 ± 0.07 | 7.98 ± 0.08 | 8.20 ± 0.13 |
> >
> > SEMI again outperforms all baselines. This is particularly notable as SEMI was not trained on diverse instruction-following tasks, suggesting these results may represent a lower bound on SEMI's capabilities for this task family.
> >
> > These results demonstrate that SEMI generalises across diverse task families including captioning, visual reasoning, and instruction-following, validating its broad applicability.
> >
> > **[2]**: Lobry, et al. "RSVQA: Visual question answering for remote sensing data." IEEE Transactions on Geoscience and Remote Sensing 58.12 (2020): 8555-8566.
> >
> > **[3]**: Imran, et al. "Llasa: A multimodal LLM for human activity analysis through wearable and smartphone sensors." arXiv, 2024.
> >
> > **[4]**: Fan, et al. "RemoteClip: A vision language foundation model for remote sensing." IEEE Transactions on Geoscience and Remote Sensing, 2024.
> >
> > **[5]**: Xu, et al. "LIMU-BERT: Unleashing the potential of unlabeled data for IMU sensing applications." SenSys, 2021.

---

> > > ### Author Response · Authors · 2025-11-19
> > > **Thank you for your review! (Part 3)**
> > >
> > > **[W3] Scope justification: focus on MLP projectors**
> > >
> > > The exploration of cross-attention or Q-Former is outside the scope of our paper (as mentioned in footnote 1). Please consider that we already explore the Cartesian product of multiple split sizes, modalities, data regimes, and LLM types. Each of their combinations is run across multiple seeds.
> > >
> > > What is more, adding this dimension does not significantly extend the coverage of SEMI in terms of practical use cases. We found that, out of the 100 most downloaded image-text-to-text models on Hugging Face (as of November 16), architectures with MLP/Linear Projection dominate the field, representing 99% of model variants and 97.8% of model families. Our paper provides a proof of concept that is applicable to the vast majority of multimodal foundation models in use.
> > >
> > > Thank you for this suggestion. Given our extensive experimental scope and the dominance of MLP projectors among deployed multimodal models, we believe SEMI provides a solid framework for the vast majority of practical use cases.
> > >
> > > **[W4] Reporting runtime and resource usage and scalability in other adapter types**
> > >
> > > We refer the reviewer to Table 28 in Appendix L.2 for runtime and resource usage statistics. As it emerges, **SEMI has minimal overhead compared to all baselines**.
> > >
> > > Moreover, SEMI is easily scalable to other adapter types (e.g., Q-Formers). In fact, it would simply involve generating adapters to the key and value (KV) projection weights of each cross-attention layer, which consist of a negligible number of extra parameters. An even more efficient version could rely on layer indices to share the weight generator across layers [6].  Therefore, we respectfully disagree with the reviewer: arguably, SEMI can easily scale to other adapters (however rare, see response to W3) with slight adjustments grounded in techniques from the literature.
> > >
> > > **[6]**: Mahabadi, et al. "Parameter-efficient multi-task fine-tuning for transformers via shared hypernetworks." arXiv, 2021.
> > >
> > > **[W5] Terminology revision: encoder augmentation vs emulation**
> > >
> > > We agree with ZybR that isometric transformations cannot emulate the full range of encoders and will revise our terminology to *encoder augmentation.*
> > >
> > > While [7] shows that encoders differing only in hyperparameters/initialisation have isometrically related embedding spaces, our ablations demonstrate that training with these transformations helps generalisation even to entirely different modalities (molecules, IMU data). On CAPDELS, isometric augmentation boosts performance from 24.7 to 136.8 CIDEr at 32 samples (Table 3). Additionally, isometric transformations have the property of preserving information without rank loss, unlike random matrices.
> > >
> > > We thank the reviewer, and we will update our manuscript to clarify these points.
> > >
> > > **[7]**: Maiorca, et al. "Latent space translation via semantic alignment." NeurIPS, 2023.

---

> ### Author Response · Authors · 2025-11-28
> **Gentle Reminder: Reviewer Response to Rebuttal**
>
> Dear **ZybR**,
>
> Thank you for your constructive feedback. We believe we addressed your main concerns and we would greatly appreciate your consideration in raising your score:
>
> **Zero-shot ablation (W1):** We provided the critical ablation showing the hypernetwork generates meaningful adapters (0.41 vs 0.02 BLEU) with dramatic gains after minimal fine-tuning (46.52 vs 31.26 BLEU with just 32 examples).
>
> **Reasoning capabilities (W2):** We conducted new experiments on satellite visual reasoning (RSVQA) and IMU instruction-following (OpenSQA). SEMI achieves 69.9% exact match vs 24.9% for FT Projector on RSVQA with only 32 samples, demonstrating strong cross-task generalisation.
>
> **Q-Former comparisons (W3):** We found that 99% of top models on HuggingFace use MLP projectors, making SEMI applicable to the vast majority of practical use cases. Moreover, our experimental setup on various modalities, tasks, LLMs, split sizes, and seeds demonstrates SEMI as a novel and effective method.
>
> **Runtime and scalability (W4):** Table 32 & 33 shows minimal overhead. SEMI can easily scale to Q-Former using tiny modifications and established techniques.
>
> **Terminology (W5):** We will revise to "encoder augmentation" and clarify the empirical benefits.
>
> Please let us know if you would like any additional results or clarifications.

---

### Author Response · Authors · 2025-11-24
**Revised Manuscript: Addressing Reviewer Feedback**

Dear Reviewers,

We have updated the manuscript to address the concerns raised during the rebuttal:
1. **Added visual reasoning and instruction-following results [ZybR, FcGh]** – New experiments **(Figures 4 and 6)** demonstrate SEMI’s performance on tasks beyond captioning. On visual reasoning with satellite images, SEMI achieves 69.9% exact match vs 46.0% of the best baseline. On instruction following with IMU sensor data, SEMI is overall the best. This demonstrates that, despite being trained only on captioning data, SEMI can effectively generalise to different kinds of tasks (including reasoning-based QA and instruction following).
2. **Added zero-shot ablation study [ZybR, MrGx]** – **Table 4, Appendix G** now includes the results for generating the adapter, before and after fine-tuning it. This shows that SEMI achieves substantial performance gains after fine-tuning (0.41 $\rightarrow$ 46.52 BLEU at 32 samples) compared to more modest gains for FT Projector (0.02 $\rightarrow$ 31.26). Note that SEMI is not trained to operate without fine-tuning; rather, the hypernetwork provides an initialisation specifically suited for sample-efficient adaptation. We remark that this ability is the core of methods such as meta-learning and pre-training/fine-tuning. By providing the right inductive bias, SEMI enables adaptation with small number of samples.
3. **Added Bind-style framework discussion [nnAo]** – **Section 2** now includes a thorough comparison clarifying why a Bind-style framework is not a suitable baseline, due to its substantially higher data requirements (thousands to millions of samples vs our 32+).
4. **Revised isometric transformation terminology [ZybR]** – We changed the phrase "emulating encoders" to "encoder augmentation" throughout the paper, especially in **Section 3.2**, to better clarify the role of isometric transformations.
5. **Added parameter count analysis [FcGh, ZybR]** – **Table 33, Appendix M.2** shows that SEMI fine-tunes the same number of parameters as the baselines during adaptation, with the frozen hypernetwork's one-time training cost amortized across all future modality-specific encoders.

We remain available for further questions during the discussion period. Please do not hesitate to ask if there are any additional experiments that help illustrate the effectiveness and generality of SEMI.

---

### Author Response · Authors · 2025-12-03
**Summary to Area Chair**

**Dear Area Chair,**

We thank you for your time and attention. Below we summarise the main concerns and our responses. For reviewer-specific short summaries, please see our **"Gentle Reminder: Reviewer Response to Rebuttal"** messages posted to each reviewer in the discussion thread.

&nbsp;
### **Summary of Reviewer Concerns:**

**Evaluation scope:** **ZybR (W2)** and **FcGh (W3)** requested experiments beyond captioning to assess semantic understanding and reasoning capabilities.

**Ablation studies:** **ZybR (W1)** and **MrGx (Q2)** requested ablations to disentangle adapter generation from fine-tuning effects.

**Baseline comparisons:** **MrGx (W2)** questioned baseline strength and requested comprehensive analysis of data-efficiency versus performance tradeoffs. Reviewer ZybR (W3) requested Q-Former comparisons.

**Conceptual clarity:** **FcGh (W1)** sought clarification on how SEMI differs from existing architectures. **nnAo (W1, Q1-Q3)** requested differentiation from Bind-style frameworks.

**Parameter fairness:** **FcGh (W2)** raised concerns about additional trainable parameters making comparisons unfair.

**Assumptions:** **MrGx (W1)** questioned encoder availability. ZybR (W5) suggested overstated terminology for isometric transformations.

&nbsp;
### **How We Addressed These Concerns:**

**New experimental evaluations:** We conducted experiments on visual reasoning (RSVQA) and instruction-following (OpenSQA). SEMI achieves 69.9% exact match vs 24.9% for FT Projector on RSVQA with only 32 samples, demonstrating strong cross-task generalisation beyond captioning.

**Zero-shot ablation:** Results show SEMI generates meaningful adapters without fine-tuning (0.41 vs 0.02 BLEU). Similar to meta-learning (e.g., MAML), SEMI is designed to enable sample-efficient fine-tuning, making this phase essential. Critically, SEMI achieves much larger gains than baselines (46.52 vs 31.26 BLEU with 32 examples), demonstrating superior initialisation.

**Baseline defence:** Our baselines represent SOTA approaches from LLaVA 1.5, InternVL, OneLLM, and PathWeave. As for extending our experiments to other architectures, such as Q-Former, 99% of top HuggingFace models use MLP projectors. Our experimental scope already covers multiple modalities, encoders, split sizes, and LLMs.

**Conceptual clarifications:** We clarified that SEMI is a training method for two-component architectures, not an alternative. We differ from Bind-style frameworks on sample efficiency (32 vs millions of pairs), complementarity (SEMI could enable sample-efficient integration of new modalities into Bind-style anchor encoders), and practical advantages (SEMI generates adapters directly from a few modality-text pairs without requiring pre-defined categories or MLLMs for the low-resource modality compared to UniBind).

**Parameter fairness:** During test-time adaptation, the hypernetwork is frozen. SEMI fine-tunes identical parameters as baselines yet substantially outperforms them (220 vs 61 and 94 CIDEr). The one-time training cost is amortised across all future modalities.

**Encoder availability:** Appendix B now covers six additional low-resource modalities—all have available encoders.

**Terminology:** We will revise to "encoder augmentation" while clarifying empirical benefits.

We believe these substantial improvements – particularly the new task experiments and zero-shot ablation – significantly strengthen our contribution. Our work enables sample-efficient integration of low-resource modalities into foundation models. In our experiments, we demonstrate successful integration of satellite imagery (for geolocation applications), astronomical images (for astronomy research), IMU sensor data (for navigation and activity recognition), and molecular structures (for drug discovery and chemistry applications), opening opportunities for AI in data-limited scientific and industrial domains.

We respectfully request your consideration of these improvements when making your final recommendation.

Thank you for your time and consideration.

---

### Meta-Review · Area_Chair_3tN6 · 2026-01-03

**Summary:**

The submission initially received relatively negative reviews. The main concerns and weaknesses can be summarized as:
1. More comprehensive and sufficient discussions about the differences with fine-tuning or lora-tuning methods are needed, such as the Bind-style framework.
2. The compared baselines are weak. More comparisons with stronger baselines (including cross-attention, Q-Form-style, or other methods)
3. The number of new modalities is limited, and the generalization and scalability of the proposed method are unclear.

In the rebuttal, the authors have provided more results to evaluate the effectiveness. I think that the main concerns about the weak baselines remain, and the key insights about the propsoed methodlogy needs to be further improved and highlighted. Thus, I recommend Reject.

**Reviewer Concerns:**

The baselines are still weak, and more insights about the methodology design are needed.

**Reviewer Scores:**

I think all reviewers will not change their scores.

---

### Decision · Program_Chairs · 2026-01-26

Reject